



# Mid-Holocene thinning of David Glacier, Antarctica: Chronology and Controls

Jamey Stutz[1], Andrew Mackintosh[2], Kevin Norton[3], Ross Whitmore[1,2], Carlo Baroni[4], Stewart S.R. Jamieson[5], R. Selwyn Jones[2], Greg Balco[6], Maria Cristina Salvatore[4], Stefano Casale[4], Jae Il Lee[7], Yeong Bae Seong[8], Hyun Hee Rhee[8], Robert Mckay[1], Lauren J. Vargo[1], Daniel Lowry[1,9], Perry Spector[6], Marcus Christl[10], Susan Ivy Ochs[10], Luigia Di Nicola[11], Maria Iarossi[4], Finlay Stuart[11], and Tom Woodruff[12]

[1]Antarctic Research Centre, Victoria University of Wellington, PO Box 600, Wellington, New Zealand 6140
[2]School of Earth, Atmosphere and Environment, Monash University, 14 Rainforest Walk, Victoria, Australia 3800
[3]School of Geography, Earth and Environmental Sciences, Victoria University of Wellington, PO Box 600, Wellington, New Zealand 6140
[4]Dipartimento di Scienze della Terra, Università di Pisa, Via Santa Maria, 53, 56126 Pisa, Italy
[5]Department of Geography, Durham University, South Road, Durham, DH1 3LE, UK
[6]Berkeley Geochronology Center, 2455 Ridge Road, Berkeley, CA 94709, USA
[7]Korean Polar Research Institute, 26 Songdomirae-ro, Yeonsu-gu, Incheon 21990, Korea
[8]Department of Geography, Korea University,145 Anam-ro, Seongbuk-gu, Seoul, Korea
[9]GNS Science, 1 Fairway Dr. Avalon, 5010, New Zealand
[10]Department of Physics, ETH Zürich, Otto-Stern-Weg 5 8093 Zürich, Switzerland
[11]Scottish Universities Environmental Research Centre, Scottish Enterprise Technology Park/Rankine Av, Glasgow G75 0QF, United Kingdom
[12]PRIME Lab, Purdue University 525 Northwestern Avenue, West Lafayette, IN 47907, USA

**Correspondence:** Jamey Stutz (jamey.stutz@vuw.ac.nz)

**Abstract.** Quantitative satellite observations provide a comprehensive assessment of ice sheet mass loss over the last four decades, but limited insights into long-term drivers of ice sheet change. Geological records can extend the observational record and aid our understanding of ice sheet – climate interactions. Here we present the first millennial-scale reconstruction of David Glacier, the largest East Antarctic outlet glacier in Victoria Land. We use surface exposure dating of glacial erratics deposited

on nunataks to reconstruct changes in ice surface elevation through time. We then use numerical modelling experiments to determine the drivers of glacial thinning.

Thinning profiles derived from 45 [10]Be and [3]He surface exposure ages show that David Glacier experienced rapid thinning up to 2 m/yr during the mid-Holocene (∼6,500 years ago). Thinning stabilised at 6 kyr, suggesting initial formation of the Drygalski Ice Tongue at this time. Our work, along with terrestrial cosmogenic nuclide records from adjacent glaciers, shows

simultaneous glacier thinning in this sector of the Transantarctic Mountains occurred ∼3 kyr after the retreat of marine- based grounded ice in the western Ross Embayment. The timing and rapidity of the reconstructed thinning at David Glacier is similar to reconstructions in the Amundsen and Weddell embayments.

In order to identify the potential causes of these rapid changes along the David Glacier, we use a glacier flow line model designed for calving glaciers and compare modelled results against our geological data. We show that glacier thinning and



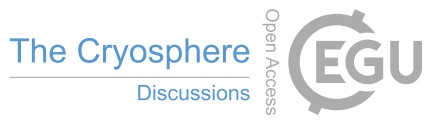

marine-based grounding line retreat is initiated by interactions between enhanced sub-ice shelf melting and reduced lateral buttressing, leading to Marine Ice Sheet Instability. Such rapid glacier thinning events are not captured in continental or sector-scale numerical modelling reconstructions for this period. Together, our chronology and modelling suggest a ∼2,000-year period of dynamic thinning in the recent geological past.

**1   Introduction**

The Earth experienced its last major period of climate warming between ∼20,000 and 11,700 years ago. During this period known as the Last Glacial Termination, ice sheets retreated in both hemispheres, causing a sea level rise of ∼130 m. Ice sheet retreat continued into the Holocene, and global sea level stabilised at near preindustrial levels by 6-7,000 years ago. During this period, the Antarctic ice sheet retreated from expanded positions on the continental shelf. Ice sheet thinning and retreat
was rapid at times, potentially contributing to periods of rapid sea level rise Weber et al. (2014). However, constraints, and particularly quantitative age control on this retreat behaviour are only available for a limited number of sites in Antarctica, and are entirely absent from David Glacier, the largest outlet glacier in Victoria Land and the focus of this study. Improved understanding of the timing and the processes that caused Antarctic Ice Sheet (AIS) retreat during this period help us to better understand the processes driving observed mass loss in parts of Antarctica today.

Geological reconstruction of the AIS through time provides critical insights into the history of ice sheet change and can narrow uncertainty on Antarctica's contribution to global sea level rise during this period. A geologic perspective on ice sheet behavior is particularly useful as:

1. Quantitative satellite observations only extend back ∼40 years (IMBIE, 2018; IPCC, 2013; Miles et al., 2013; Rignot et al., 2019) and these observations do not fully capture natural variability of ice sheet behaviour.

2. Integration of the data constraining marine extent and terrestrial thickness of an ice mass through time provides a robust data set for evaluation of numerical ice sheet model simulations, which are used to predict future ice sheet contributions to sea level rise.

3. Data from the Gravity Recovery and Climate Experiment (GRACE) satellites offer a powerful measure of present-day ice sheet mass change but require a correction for millennial to centennial-scale ice sheet load history. This is because the
gravity signal is strongly influenced by crustal uplift from glacio-isostatic adjustment (GIA) (Whitehouse et al., 2012; Whitehouse, 2018). Solutions using GRACE gravity data, corrected using a precise GIA signal, have the capability to provide a more accurate estimate of present-day ice sheet mass balance at a regional scale (King et al., 2012; Simms et al., 2019).





4. Examples of past ice sheet changes provide evidence of the climatic processes and dynamic feedbacks by which ice
sheets respond to environmental forcing. For example, the mass balance of large, marine-terminating outlets of Antarctica
is affected by decadal to interannual changes in regional climate drivers such as the Southern Annular Mode and El Niño-
Southern Oscillation(Dutrieux et al., 2014; Miles et al., 2013; Walker and Gardner, 2018; Assmann et al., 2019; Wille
et al., 2019). However, we do not currently know if these short-term drivers are important over centennial to millennial
timescales.

Reconstructions of large marine terminating outlet glaciers provide an opportunity to constrain and understand the terrestrial
and marine sectors of ice sheets. At the LGM, the David Glacier thickened and expanded as an ice stream, coalesced with
grounded marine-based ice, and extended hundreds of kilometers into the western Ross Sea (Anderson et al., 2014; Livingstone
et al., 2012; Shipp et al., 1999). During deglaciation, grounding line retreat from north of Coulman Island was initiated by ∼13
ka, with retreat near Terra Nova Bay by ∼11 ka. Subsequent grounding line retreat north of Ross Island was achieved by
∼8.6 ka and a modern configuration established by ∼2-4 ka (Anderson et al., 2014; McKay et al., 2016). Outlet and valley
glaciers along the Northern Victoria Land sector (Reeves, Priestley, and Tucker, Aviator, Campbell glaciers, respectively) began
thinning at ∼17 ka and the majority of thinning ceased by ∼7.5 ka, broadly coincident with a linear rise in sea level and ocean
temperatures throughout deglaciation (Baroni and Hall, 2004; Johnson et al., 2008; Smellie et al., 2018; Goehring et al., 2019;
Rhee et al., 2019). In contrast, outlet glaciers covering a large swath of the TAM from Southern Victoria Land to Southern
TAM, experienced episodic thinning during the early-mid Holocene, likely due to local topographic effects associated with
Marine Ice Sheet Instability (MISI) (Jones et al., 2015; Spector et al., 2017). Overall, variation in the timing of glacier thinning
suggests sea-level rise, ocean warming, and overdeepened subglacial topography as general controls for the reconstructed
glacier behaviour.

The David Glacier drains the East Antarctic Ice Sheet (EAIS), traverses and incises the Transantarctic Mountains and dis-
charges into the western Ross Sea as the floating Drygalski Ice Tongue, first discovered by the British National Antarctic
Expedition (1901-1904) (Fig. 1 and A1). Comprising an area ∼210,000 km2, the glacier is the largest in Victoria Land, repre-
senting a significant element of the Antarctic cryosphere, draining from Dome C and Talos Dome. Given its size and inferred
intimate connection with the marine-based grounded ice in the Ross sea highlights the David Glacier as a key target for further
investigation. Through surface exposure dating of glacial deposits, this study aims to constrain the LGM to present behaviour
of the David Glacier. Further, we investigate the relative role of ocean heat and lateral buttressing on glacier thinning and retreat
through idealised glacier flow-line modelling experiments.

## 2 Methods

### 2.1 Field and laboratory methods

Surface exposure dating using *in situ* terrestrial cosmogenic nuclides has transformed the ability to reconstruct the AIS through
time (e.g. Stone et al., 2003; Mackintosh et al., 2007; Todd et al., 2010; Balco, 2011; White et al., 2011; Jones et al., 2015;





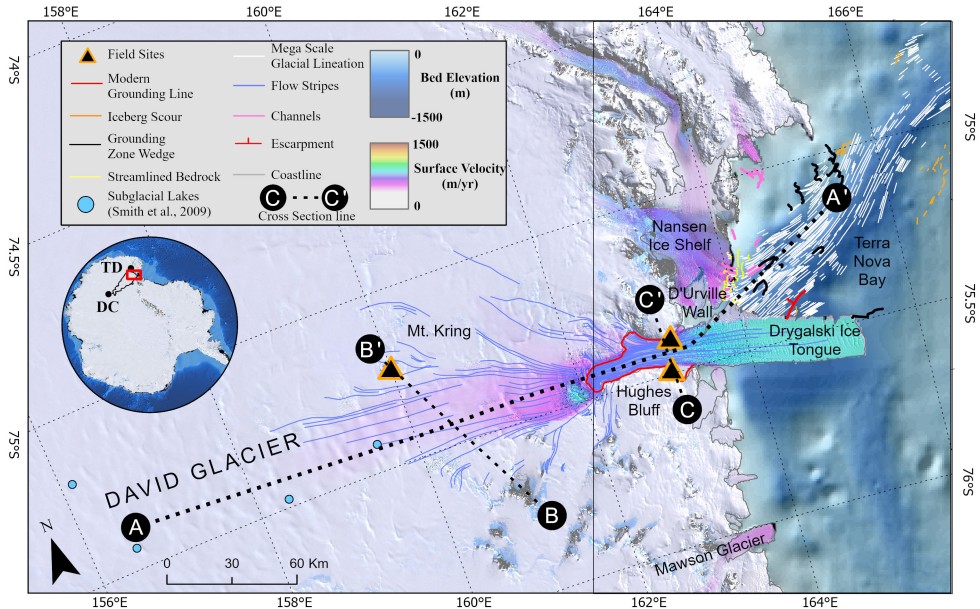

**Figure 1.** A) Map of greater David Glacier/Terra Nova Bay region including mapped sub-marine geomorphic features and geographic features mentioned in text. Geomorphic cross sections (A-A', B-B' and C-C') highlighted in Fig A1. Inset shows extent of main map along with drainage basin of David Glacier and position of two local ice domes (Dome C (DC) and Talos Dome (TD)). Data sources: Satellite imagery of (Bindschadler et al., 2008), surface velocity of (Rignot et al., 2011), surface elevation of (Howat et al., 2019), bed elevation of (Fretwell et al., 2013) and bathymetry of (Arndt et al., 2013)

Spector et al., 2017; Small et al., 2019). In the field, we sampled erratics and rocky outcrops adjacent to the David Glacier both upstream and downstream of the present-day grounding line. Downstream sites are expected to show the most recent and dramatic change while sites upstream are expected to record smaller and longer term changes in ice thickness (Anderson et al., 2004; Bockheim et al., 1989). Using structure from motion photogrammetric technique (Vargo et al., 2017), we constructed a high-resolution digital elevation model of each field site allowing integration of glacial geological field surveys with the regional and local geomorphology (Baroni et al., 2004) (Figs. A3 and A4). Sampling focused primarily on glacial erratic cobbles from local glacial deposits and glacially moulded bedrock surfaces which were collected along vertical transects perpendicular to the modern glacial flow direction. Faceted and striated glacial erratics are preferred as they indicate subglacial origins with minimal post-depositional erosion. The sampling method effectively tracks the upper ice surface through time (Stone et al., 2003; Mackintosh et al., 2007; Jones et al., 2015; Small et al., 2019).

Of the 110 samples collected in the field, 36 samples were processed for $^{10}$Be analysis and 9 samples for $^{3}$He analysis. For $^{10}$Be analysis, quartz is the target mineral and is present in granites and sandstones. In this case, sandstones are preferred as



they contain ∼90% quartz and require significantly less physical processing. In the field, we preferentially sampled on isolated, local topographic high points, distant from areas of blue ice, snow drifts or local till deposits. While it is the focus of this study to use glacial erratics to track glacier thinning, bedrock samples were also collected to understand long-term exposure and erosion.

Samples collected in the field were processed at the cosmogenic nuclide facilities at Te Herenga Waka Victoria University of Wellington. For [10]Be, we separated quartz by crushing and sieving to extract the 250-500-micron grain size fraction. Magnetic minerals were separated using a Frantz isodynamic magnetic separator. For granitic rocks, feldspars were removed by froth flotation. Multiple etchings in 1-2.5% Hydroflouric acid further purified the quartz separates. Beryllium was extracted from quartz following an established method including addition of [9]Be carrier, enhanced quartz etching and digestion followed by ion exchange column chemistry and BeOH precipitation (Norton et al., 2008). Targets of BeO were packed and sent to the PRIME lab for analysis using accelerated mass spectroscopy (AMS). For [3]He, we targeted pyroxene from the 125-250 micron grain size fraction. Pure pyroxene was obtained using an established HF etching method and [3]He/[4]He ratios were measured at the Berkeley Geochronology Center noble gas spectrometer following established methods (Bromley et al., 2014; Blard et al., 2015; Balter et al., 2020). We supplemented this data set with eight samples collected prior to the 2016-17 austral summer and those samples were processed with the methods outlined in Oberholzer et al. (2003, 2008); Di Nicola et al. (2009, 2012). Exposure ages are calculated by converting the AMS derived [10]Be/[9]Be ratio to [10]Be concentration by subtracting a known amount of [9]Be carrier added during chemical processing. Final exposure ages are calculated using [10]Be or [3]He concentration, field information (elevation, shielding, and sample thickness) and production rate scaling method (LSDn) (Balco et al., 2008; Lifton et al., 2014; Marrero et al., 2016; Jones et al., 2019). For [10]Be we employed Jones et al. (2019) systematics and for [3]He we used Balco et al. (2008) systematics. All relevant data used to calculate exposure ages are served on the ICE-D online database within the David Glacier region (antarctica.ICE-D.org)(Balco, 2020).

## 2.2 Glacier modelling approach

In order to understand potential controls on the grounding line migration and onshore thinning, we apply a glacier flow-line model to the David Glacier. Originally designed to track grounding line migration using a moving grid, the flow-line model we employ has been described fully elsewhere (e.g. Vieli and Payne (2005); Nick et al. (2009); Jamieson et al. (2012); Enderlin et al. (2013); Jamieson et al. (2014); Clason et al. (2016); Whitehouse et al. (2017)). The underlying theory and governing equations are outlined in Vieli and Payne (2005) and here we describe the components pertinent to our research questions.

The glacier modelling focuses on a ∼1,600 km long flowband covering a diverse set of flow regimes from the higher-elevation, low-velocity Dome C area to the lower-elevation, high-velocity outlet glacier (David Glacier), ice shelf (Drygalski Ice Tongue) and ice stream beyond where it converged with grounded ice in the Ross Sea. Ice flow is modelled as a mechanically coupled ice sheet, ice stream and ice shelf flow governed by fundamental equations 1 - 6.





Ice sheet flow is calculated using the shallow ice approximation which assumes ice flow only from internal ice deformation
and horizontal velocity defined as:

$$u = C \left( \frac{\delta s}{\delta x} \right) h^{n+1}$$   (1)

where n is Glens flow law exponent, s is ice surface elevation and C is a constant given by:

$$C = \frac{2A(\rho_i g)^n}{n+2}$$   (2)

As ice reaches flotation, ice is assumed to spread unidirectionally along the flowline, ice shelf flow is balanced by:

$$2\frac{\delta}{\delta x} hv \frac{\delta u}{\delta x} = \rho_i gh \frac{\delta s}{\delta x}$$   (3)

where $\rho_i$ is ice density, g is gravitational acceleration and v is the vertically average effective viscosity defined as:

$$v = A^{\frac{-1}{n}} \left[ \left( \frac{\delta u}{\delta x} \right)^2 \right]^{(1-n)(2n)}$$   (4)

Where A is the temperature dependant rate factor and $\frac{\delta u}{\delta x}$ is the effective strain rate.

For an ice stream, equation 4 is modified by including a basal friction coefficient $\beta$ which is linearly related to the horizontal
velocity $u$ and is given by:

$$2\frac{\delta}{\delta x} hv \frac{\delta u}{\delta x} - \beta^2 u = \rho_i gh \frac{\delta s}{\delta x}$$   (5)

Recent applications using this model include testing the impact of a width term to calculate lateral buttressing along a
coupled ice stream-shelf and enhanced treatment of ice shelf dynamics. Nick et al. (2010) and Jamieson et al. (2012) modelled
depth (H) and width (W) averaged ice flow ($u$) using the following equation:

$$2\frac{\delta}{\delta x} hv \frac{\delta u}{\delta x} - \beta^2 u + \frac{H}{W} \left( \frac{5u}{2AW} \right)^{\frac{1}{n}} = \rho_i gh \frac{\delta s}{\delta x}$$   (6)

The inclusion of the width term allows modelling of changing offshore trough width which is common in palaeo-ice streams
and outlet glaciers, and further supported by geomorphic interpretations using high resolution bathymetry data (Jamieson et al.,
2012; Livingstone et al., 2012, 2016).

Whitehouse et al. (2017) improved the ice shelf dynamics to incorporate the treatment of large horizontal grounding line
movements that are expected for the palaeo-David Glacier based on the long term presence of the Drygalski Ice Tongue
and evidence of extensive sub-ice shelf conditions inferred from marine sedimentary cores (Domack et al., 1999; McKay
et al., 2008). In the model, the ice shelf geometry evolves with time, and variations in ice shelf thickness and extent are fed





into lateral drag calculations (Whitehouse et al., 2017). Improved treatment of ice shelf dynamics allows exploration of the grounding line sensitivity to ice shelf feedbacks such as reduced lateral buttressing. Importantly for this study, a reduction

in lateral buttressing is expected as the expanded David Glacier and grounded ice in the Ross Sea decouple, which has been suggested from interpretation of submarine geomorphic features in the western Ross Sea (Shipp et al., 1999; Halberstadt et al., 2016).

### 2.2.1 Boundary conditions

The model domain runs along a ~1,600 km flowline from Dome C to the continental shelf break (Fig. A1 and A2). The

marine sector of the flowline follows the Drygalski Trough axis and runs parallel to mapped mega-scale glacial lineations (MSGL). Onshore, ice stream width follows the drainage basin outline derived from surface elevation and velocity. Offshore, we determine ice stream width by orientation and distribution of trough parallel MSGLs consistent with Jamieson et al. (2012).

In order to evaluate model performance and appropriate boundary conditions for deglacial experiments, we conduct sensitivity experiments designed to reproduce the modern behaviour of the David Glacier. The model allows the user to input surface

and bed geometry, ice stream width, accumulation, velocity and a basal traction parameter. Modern surface elevation, accumulation and ice velocity are well constrained from satellite and *in situ* measurements (Rignot et al., 2011; van Wessem et al., 2018; Howat et al., 2019). In order to account for confluent ice from tributary glaciers we employ an ice mass accumulation scheme used in previous modelling studies whereby tributary ice mass is added along the length of the ice stream (Jamieson et al., 2014). This tributary 'injection' from a secondary ice stream originating from Talos Dome is included in the surface

mass budget and for the David Glacier model domain.

Ongoing aerogeophysical and remote sensing techniques continue to reveal new detail of the subglacial environment of the AIS (Morlighem et al., 2019). Common amongst many TAM outlet glaciers, a prominent bedrock high immediately upstream of the modern grounding line provides a significant stabilising impact on the David Glacier (Fretwell et al., 2013; Morlighem et al., 2019). The bed geometry along the flowline is derived solely from Bedmap2 and the International Bathymetric Chart of

the Southern Ocean (Fretwell et al., 2013; Arndt et al., 2013).

Geologic data from former glaciated terrains and satellite observations can be used to approximate basal conditions, yet a general lack of *in situ* data from the ice sheet basal environment lead to enduring uncertainties in the basal stress regime (Joughin et al., 2006; Stokes et al., 2015). Based on existing knowledge of sea floor morphology and consistent with previous modelling experiments, the basal traction parameter used in this model attempts a first order approximation of two subglacial

basal environments: 1) relatively high basal traction for onshore ice flow over bedrock and 2) relatively low basal traction for offshore ice flow over soft sediments associated with basal till deposits (Dowdeswell et al., 2004; O Cofaigh et al., 2005).

Modern day sensitivity experiments tune user defined parameters such as internal ice temperature and basal traction parameters until best-fit with modern surface geometry. Overall, the modern grounding line position, ice shelf thickness and ice shelf length are stable over a 2,000 year modelled period. The model is unable to reproduce stable conditions along the steep surface

profile as the David Glacier descends from the ice sheet interior. In this zone, the modelled upper surface is steep and undergoes thinning throughout the modelled time period. In an effort to stabilise the upper surface upstream of the grounding line, we tune





the basal traction parameter, effectively stiffening the bed, to reduce the modelled instability. Resultant modelled estimates of basal shear stress approach 100 kPa in this setting. While difficult to constrain with *in situ* measurements, Zoet et al. (2012) suggest higher stresses should be expected near the modern day grounding zone which is consistent with the modelled stress
distribution in this study.

### 2.2.2    Deglaciation approach

LGM geometric boundary conditions incorporate modelled ice thickness and surface velocity from W12, a geologically con-strained continental scale glacial isostatic adjustment model (Fig.A2) (Whitehouse et al., 2012). Using the W12 modelled ice thickness and modern bed elevation (Fretwell et al., 2013) allows for an estimate of ice surface elevation without introducing
uncertainty involving variable along flow isostatic response and dynamic topography associated with the long-term evolution of the Antarctic subglacial topography (Stern et al., 2005; Whitehouse et al., 2019; Paxman et al., 2019). The resulting ice sur-face elevation is consistent with all existing geological constraints and serves as an initial ice profile for deglaciation sensitivity experiments.

    In order to account for environmental changes during deglaciation, transient changes in accumulation and internal ice tem-
perature are tuned over the modelled period to ensure a stable LGM configuration consistent with geological constraints. Knowledge of past accumulation over glacial-interglacial cycles are restricted to ice core data and internal ice sheet layer map-ping near high elevation ice domes (Siegert, 2003; Frezzotti et al., 2005; Buiron et al., 2011; Cavitte et al., 2018). Generally, modern accumulation patterns show relatively low accumulation (0.02 m/yr) over the East Antarctic interior and high accu-mulation (0.2 m/yr) near the coastline (Arthern et al., 2006; Lenaerts et al., 2012; van Wessem et al., 2018). Interpretations
from ice core records suggest LGM accumulation rates were lower than modern accumulation rates and generally correlate well with temperature proxy records throughout the Holocene (Siegert, 2003; Veres et al., 2013). We use a scaling relationship between modern accumulation patterns and suggest that accumulation at ∼15 ka was roughly 75% of modern accumulation (Veres et al., 2013).

    Sensitivity experiments designed to understand the controls on thinning and retreat are initialised with a set of user defined
parameters derived from modern sensitivity experiments. Using an optimised set of accumulation and temperature forcings, we explore transient changes in lateral buttressing and sub-ice shelf melt rate to isolate their relative influence on glacier thinning and retreat. We initiate grounding line retreat by progressively increasing sub-ice-shelf melt rate or decreasing lateral buttressing.

    All experiments run for 15 kyr with an initial spin up period lasting 7.5 kyr, at which point a forcing perturbation is applied.
When forcing the model, we linearly increase the forcing over a 500 year period. In sub-ice-shelf melt cases, we varied the sub-ice-shelf melt rate perturbation within a 0.5 m/yr window in order to simulate short-lived pulses of relatively warmer or cooler water mass changes. All idealised scenarios are presented along with results in Supplementary Table **??**.





# 3   Results - chronology

Of the 45 samples analysed in this study, 21 yield mid-Holocene exposure ages. Holocene aged samples are interpreted to have
received minimal prior exposure (i.e inheritance) and suggest a simple exposure history. Focusing on two sites, we derive a
high-resolution chronology of glacier thinning from Mt. Kring and Hughes Bluff.

## 3.1   Mt. Kring

Mt. Kring, situated in the interior of the EAIS, lies along the flank of the ice stream draining Dome C. The nunatak, composed
of layered dolerite, rises ∼300 metres above the local ice surface. Of the 24 samples processed, three are dolerite bedrock
and 21 erratics of mixed lithologies (Fig. 2 and A3). The surface at the peak of Mt. Kring lacks glacial striations or erratics
and has an apparent $^3$He bedrock exposure age of 554 ± 91 ka. A discontinuous, steep ridge line between ∼200-300 metres
above local ice was not sampled due to inaccessibility. At ∼200 metres above the local ice, the bedrock is striated parallel to
modern ice flow direction and has an apparent $^3$He exposure age of 550 ± 82 ka. The highest elevation erratic is found at ∼180
metres above the local ice with increasing erratic abundance culminating in a drift sheet covering nearly all bedrock below
∼110 metres above local ice. Striated, bulleted cobbles and boulders of dolerite, basalt and sandstone are common in the thin,
patchy drift covering the bedrock. The glacial drift is composite in age and we identify three populations of erratic exposure
ages (Fig. 3A). The younger population spans from 7.4-5.5 ka ($^{10}$Be, n=5)(Fig. 3B). The older population spans from 51-25 ka
($^{10}$Be, n=8 and $^3$He, n=4). The oldest erratic age population shows scattered evidence of glacial behavior between 123-63 ka
($^{10}$Be, n=2,$^3$He, n=2). The older populations are likely an artifact of repeated burial by overriding ice and multiple periods of
exposure (i.e. inheritance).





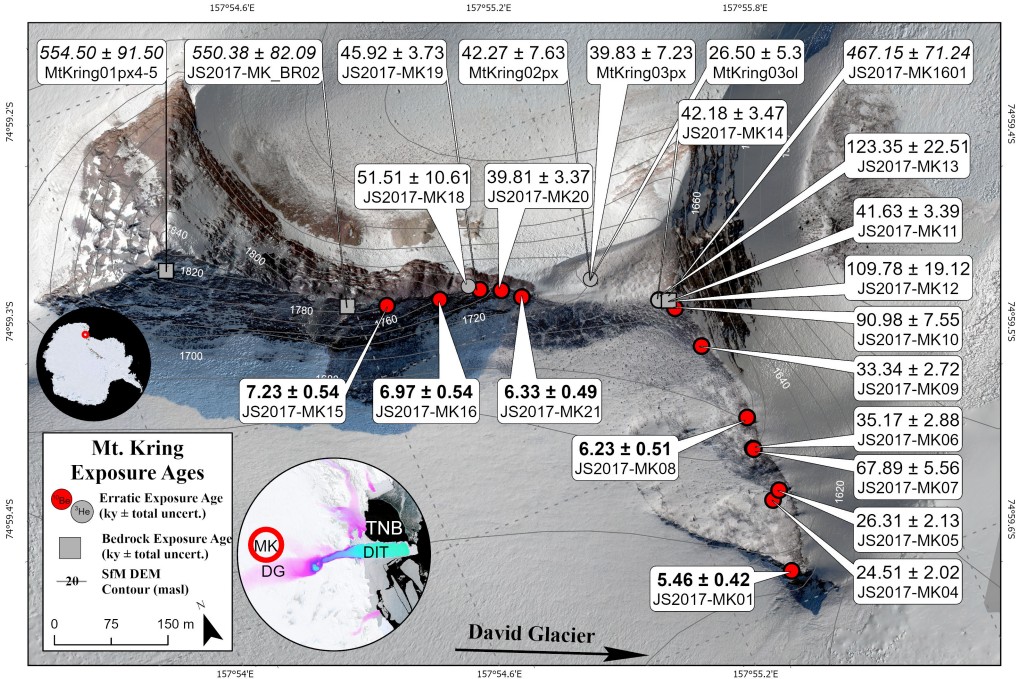

**Figure 2.** Orthomosaic map of Mt. Kring with all exposure ages ($^{10}$Be (red) and $^{3}$He (grey)) and total errors listed. Erratics and bedrock ages plotted as circles and squares, respectively. Larger inset shows surface velocity. DG=David Glacier, DIT=Drygalski Ice Tongue, TNB=Terra Nova Bay and MK=Mt. Kring





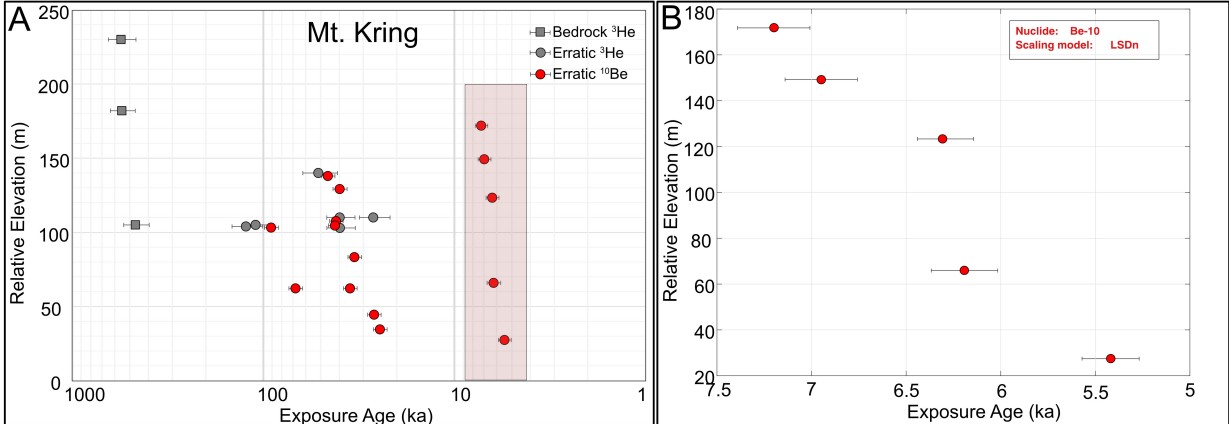

**Figure 3.** Calculated surface exposure ages for Mt. Kring. (A) All exposure ages. Erratics plotted as circles, bedrock as squares(${}^{10}$Be, red and ${}^{3}$He, grey). Errors bars show total uncertainty. (B) Shows only Holocene ages.

## 3.2 Hughes Bluff

Hughes Bluff, situated on the Scott Coast, is a granite outcrop along the southern flank of the David Glacier. The outcrop is glacially scoured exhibiting spectacular rochés moutonnées, crescentic gouge marks and striations parallel to modern flow directions (Fig. 4 and A4. Scattered glacial erratics blanket the entire outcrop. ${}^{10}$Be exposure ages from 15 erratics span from
6.7-4.3 ka (Fig. 5). The majority of glacial erratics are dated to between 6.7 and 6.2 ka. Two bedrock surface exposure ages from the highest and lowest outcrops (20.55 ka ± 2.10, 5.5 ± 0.47 ka) suggest significant wet-based glacial erosion during the LGM. At 20 metres above local ice, similar bedrock and erratic ages provide evidence of recent emergence of this lower outcrop since 5.5 ka. Given the extensive glacial erosion, the LGM surface elevation at this site was likely considerably greater than 230m, the maximum elevation at Hughes Bluff. While the onset and magnitude of thinning prior to 6.7 ka is poorly
constrained, the Hughes Bluff chronology indicates a period of rapid ice surface lowering during the mid-Holocene.





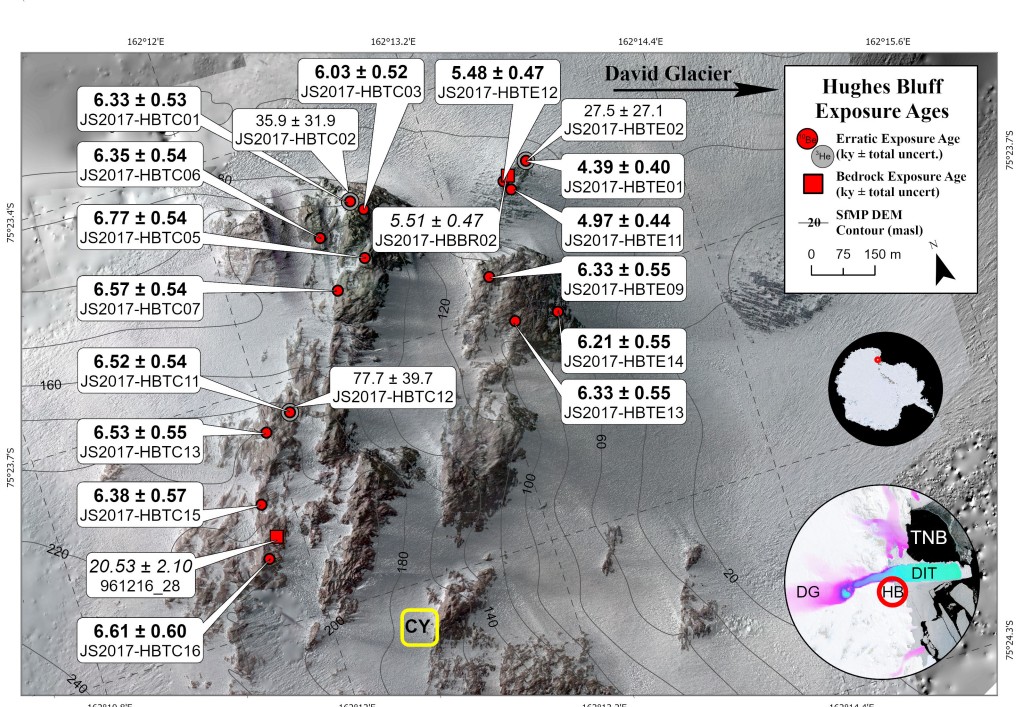

**Figure 4.** Orthomosaic map of Mt. Kring with all exposure ages ([10]Be (red) and [3]He (grey)) and total errors listed. Erratics and bedrock ages plotted as circles and squares, respectively. CY= Camp Yellow. Larger inset shows surface velocity. DG=David Glacier, DIT=Drygalski Ice Tongue, TNB=Terra Nova Bay and HB=Hughes Bluff

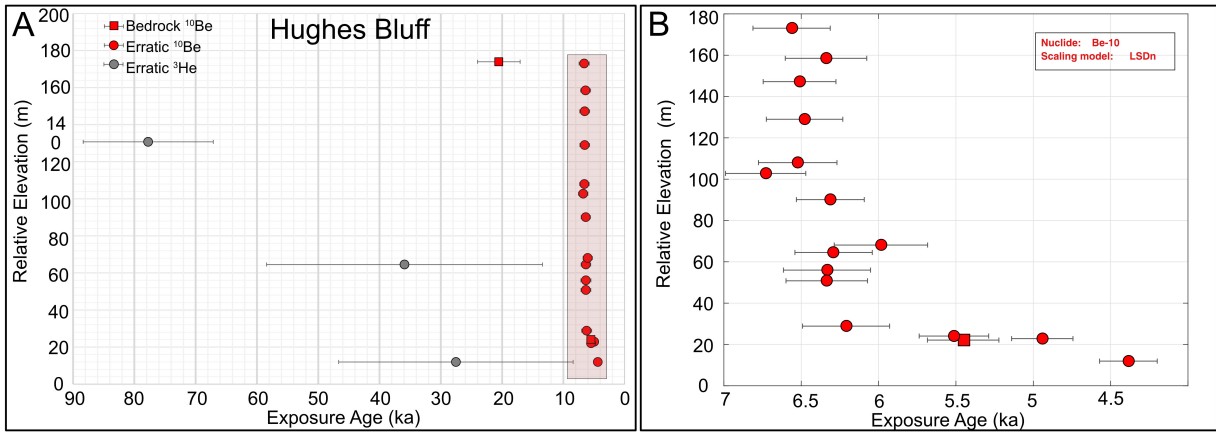

**Figure 5.** Calculated surface exposure ages for Hughes Bluff (A) All exposure ages. Erratics plotted as circles, bedrock as squares ([10]Be, red and [3]He, grey). Errors bars show total uncertainty. (B) Shows only Holocene ages.



### 3.3 High elevation constraints

In an effort to identify higher elevation glacial activity and long-term erosion history, field work was undertaken along the
D'Urville Wall and Mt. Neumayer area. The D'Urville Wall is a steep, high-elevation (>400 m) granite outcrop delimiting
the northern flank of David Fjord. The geomorphology and exposure ages from two field sites situated high above the David
Glacier (Mt. Neumayer and Cape Phillipi) limit the reconstruction of the upper glacier surface during the LGM. Mt. Neumayer
extending above the D'Urville Wall forms a rounded summit with faint striations sub-parallel to modern ice flow with few
scattered erratics. In areas above 400 m, bedrock samples contain weathering rinds and deep weathering pits filled with erratics
(Baroni et al., 2004; Giorgetti and Baroni, 2007). Bedrock exposure ages from Mt. Neumayer (649 metres above local ice, 642
$\pm$ 61 ka), a terrace on top of the D'Urville Wall (418 metres above local ice, 116 $\pm$10 ka) and Cape Phillipi ($\sim$300 metres
above local ice, 532 $\pm$52 ka, 957 $\pm$98 ka) suggest either a thin cover of cold based ice or ice free conditions through the
LGM. High elevation bedrock samples are much younger than exposure ages from nearby bedrock at similar height above
the local ice surface and suggests burial by non-erosive ice (Di Nicola et al., 2012). Together with the geomorphic evidence
from Hughes Bluff, bedrock exposure ages from the northern flank of David Glacier constrain the LGM upper surface between
300-649 metres above the local ice surface, broadly consistent with $\sim$400 masl derived from LGM age drift deposits in Terra
Nova Bay (Stuiver et al., 1981; Orombelli et al., 1990; Di Nicola et al., 2009) (Fig. A2).

### 4   Results - Glacier modelling

Given the main episode of thinning does not correlate with significant increases in atmospheric temperature or global mean
sea level (e.g. (Lambeck et al., 2014; Menviel et al., 2011; Liu et al., 2009), we ask: 1) What is the role of ocean heat in
driving the observed glacier thinning and retreat? and 2) Given the inferred intimate link between the expanded David Glacier
and grounded ice in the Ross Sea, what impact does ice sheet buttressing have on the timing and style of glacier thinning and
retreat? The style and rate of modelled thinning and retreat from all experiments are compiled in Table ??.

Focusing solely on sub-ice shelf melt rate, a set of experiments (M1-3) simulate the impacts of enhanced ocean heat on
grounding line retreat (Fig. 6). After a 7.5 kyr spin up period, a threshold sub-ice shelf melt rate of -11 m/yr achieves rapid
grounding line retreat behaviour and is consistent with the modern grounding line position. For melt rates between -2 and -10
m/yr, grounding line retreat is rapid but the final grounding line remains pinned to the prominent sill at the mouth of the David
Fjord. Modelled surface reconstructions place the upper ice surface above the Hughes Bluff site, yet agree well with Mt. Kring
data constraints. In the high melt case (M3), the grounding line position stabilises at the sill for approximately 5 kyr. The final
retreat phase from the sill to a modern position correlates with a final surface consistent with modern observations.

Focusing on lateral buttressing reduction experiments (S1-3), we simulate the impacts of glacier-ice sheet decoupling on
grounding line retreat (Fig. A3). After a 7.5 kyr model spin up period, ice shelf buttressing is incrementally reduced until
retreat is initiated. In this case, retreat occurs when lateral buttressing is reduced by 4% although it is not until further ice shelf
debuttressing to 40% that the grounding line is able to retreat over the prominent sill at the mouth of the David Fjord. In both
cases, the reduced buttressing forces rapid grounding line retreat to a prominent sill at the mouth of the David Fjord. In these




scenarios, the resulting modelled upper ice surface remains above the Hughes Bluff site when pinned to the sill. At Mt. Kring,
modelled rapid thinning is synchronous with Hughes Bluff yet results in an unrealistic final surface elevation 100s of meters
below observed modern surface elevation.

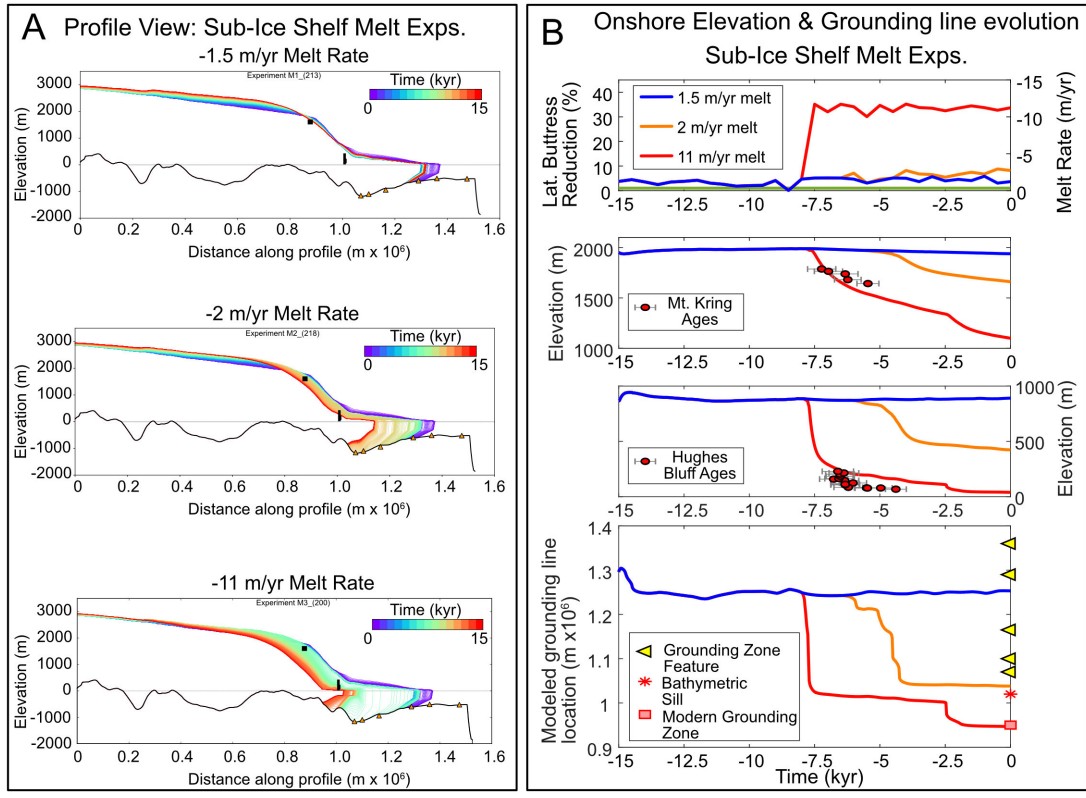

**Figure 6.** A) Profile View: 15 kyr evolution of modelled upper ice surface along flowline for sub-ice shelf melt rate experiments. Black bars
depict three sites mentioned in text. Orange triangles represent offshore features marking observed grounding zone wedges. B) Onshore eleva-
tion and grounding line evolution for sub-ice shelf melt rate experiments. Top panel: Forcings applied, Top middle panel: Time-transgressive
elevation profile for Mt. Kring with exposure ages, bottom middle panel: Time-transgressive elevation profile for Hughes Bluff surface ex-
posure ages, and lower panel: Evolution of grounding line position with modern grounding line position, bathymetric sill at mouth of David
Fjord and mapped grounding zone features.

Overall when forcings are combined, lower threshold values are required to initiate thinning and retreat (Fig. A4). Mod-
elled upper ice surface reconstruction agrees well with the Mt. Kring chronology, yet Hughes Bluff remains ice covered until
grounding line retreat from the sill approximately 5-6 kyr after the main phase of thinning and retreat.





## 5 Discussion

The thinning history of David Glacier place modern observations in a long-term perspective and allows for local, regional and continent-wide comparisons with other glacial histories and modelled ice sheet reconstructions. High-resolution, low-inheritance exposure ages obtained from Hughes Bluff and Mt. Kring overlap in time, yet reveal different thinning styles with Hughes Bluff recording rapid ice surface lowering, and Mt. Kring revealing a much slower thinning signal. The chronology from Mt. Kring implies glacier thinning reached far inland along zones of streaming ice and provides rare constraints on ice behaviour from the margins of the EAIS.

The results of flow line modelling experiments along the expanded David Glacier reveal a threshold-driven sensitivity to both sub-ice shelf melt rate and ice stream lateral buttressing. Periods of modelled grounding line retreat match periods of onshore thinning constrained by surface exposure studies at two locations along the flowline. The results show the glacier response occurs at lower threshold values when the processes act in combination, consistent with previous applications of this model (Jamieson et al., 2014).

It is not the intention of this work to reproduce the exact timing of marine-based grounding line retreat, but model results provide insights to the dominant processes responsible for glacier thinning and retreat. Although the sensitivity experiments are inconclusive as to which process or combination of processes forced the observed onshore thinning, we discuss potential explanations relating to the thinning profile at Hughes Bluff and the data-model mismatch at Mt. Kring.

### 5.1 Palaeo-thinning rates and data-model comparison

Using the high-resolution chronology from Hughes Bluff and Mt. Kring, we derive a mean estimate of past thinning rates along David Glacier using a weighted least squares regression scheme within the iceTEA plotting tools (www.ice-tea.org) (Jones et al., 2019). These reconstructed glacier thinning rates are compared to modern thinning rates derived from satellite data and continent scale ice sheet models (Small et al., 2019). At Mt. Kring, we reconstruct a longer and slower thinning event from 7.5-5.5 ka of up to 0.19 m/yr (0.06-0.19, $2\sigma$) (Fig. 7A). At Hughes Bluff, we reconstruct a period of rapid thinning 320 from 6.7-6.2 ka of up to 2 m/yr (0.19-2.06 m/yr, $2\sigma$) followed by ~4 kyr period of minimal thinning (Fig. 7B). The reconstructed palaeo-thinning along the David Glacier during the mid-Holocene is synchronous with rapid thinning reconstructed at a number of sites in Antarctica (Small et al., 2019). However, the rate of paleo thinning reconstructed at Hughes Bluff is one of the highest rates recorded in Antarctica, comparable to a reconstruction from nearby Mackay Glacier (Small et al., 2019; Jones et al., 2015).

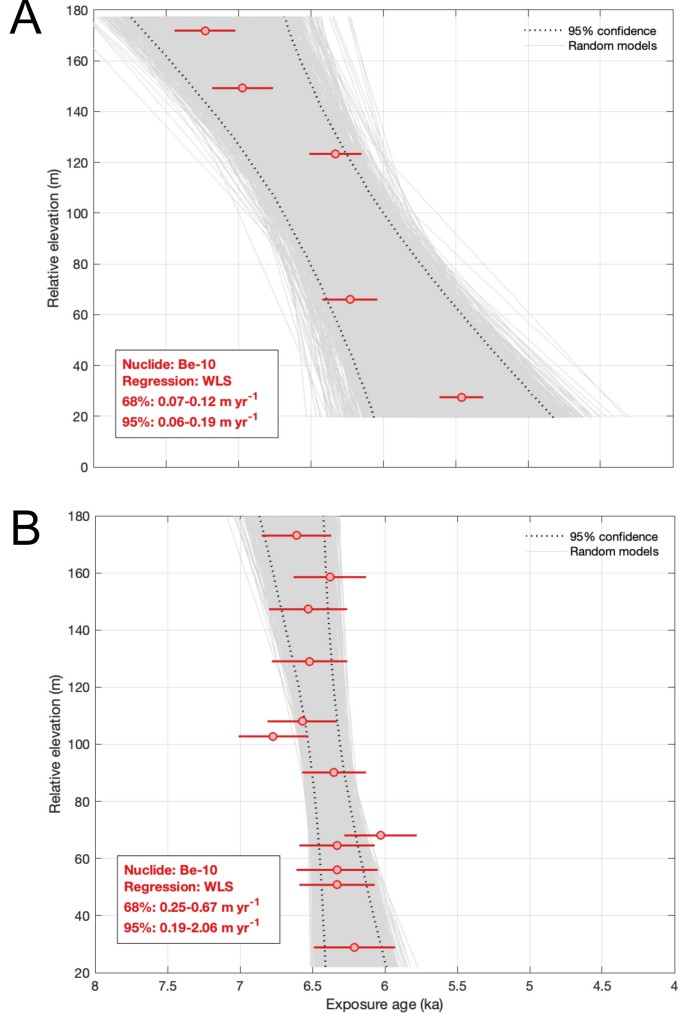

**Figure 7.** Linear thinning rates estimated for mid-Holocene records using ice-tea.org from (Jones et al., 2019) for: A) Mt. Kring Holocene samples, B) Hughes Bluff samples with outliers identified and omitted using outlier detection in ice-tea.org.

The variation in thinning rates between Mt. Kring and Hughes Bluff is suggestive of past dynamic thinning rather than accumulation-driven outlet glacier changes. Previous work from inland sites similar to Mt Kring suggested that such sites changed little or may have thickened during the Holocene due to accumulation increases (Bockheim et al., 1989; Denton et al.,

1989). In contrast, the synchronrous timing but smaller magnitude of thinning observed at Mt Kring in this study suggest that this thinning was driven by ice sheet processes at the coast that propagated inland. Such dynamic thinning, where increased ice-shelf basal melting and grounding line retreat leads to accelerated flow and inland thinning, is well documented along modern marine terminating margins of the Greenland and Antarctic ice sheets (Pritchard et al., 2009, 2012), yet identification in geologic record is rare, primarily due to a lack of exposed bedrock in the upper reaches of glacier catchments. The thinning



history from David Glacier allows for a unique comparison with the broad pattern of dynamic thinning derived from the
modern satellite record and suggests dynamic thinning can occur >100km into the interior of the EAIS and can persist over
multi-millennial timescales.

The chronologies from David Glacier provide critical insights in which to compare against modelled reconstructions from
continent-scale ice sheet models. Using the data-model comparison software (dmc.ICE-D.org), we extracted a modelled eleva-
tion history for David Glacier from five different forward ice sheet models (four continental-scale and one regional-catchment
scale) and one prescriptive post-glacial rebound model (Argus et al., 2014; Pollard et al., 2016, 2017, 2018; Kingslake et al.,
2018; Lowry et al., 2019). In order to overcome differences in spatial resolution, the scheme extracts an interpolated ice eleva-
tion history from the model grid cell containing the field site and its neighbouring model grid cell.

The resulting data-model comparison for Hughes Bluff and Mt. Kring reveal a noticeable mismatch in time during the main
phase of thinning (Fig. 8). For Mt. Kring, ice sheet models indicate a phase of thinning that precedes our thinning history
by ∼4-7 kyr. For Hughes Bluff, the timing lag is comparable to Mt. Kring with one notable exception being the post glacial
rebound model of ICE-6G. This improved match is likely because ICE-6G is constrained by multiple relative sea level curves
along the Scott Coast (Baroni and Hall, 2004; Hall, 2009; Argus et al., 2014). The style of thinning is variable between
models with noticeable short-lived pauses in thinning, mainly during the Antarctic Cold Reversal (∼15-13 345 ka) (Pedro
et al., 2016). For Mt. Kring, the amplitude of modelled elevation change matches well with the surface exposure age data. As
discussed previously, the Hughes Bluff chronology likely captures only part of the full thinning history. Overall, the data-model
mismatch may be related to (1) individual topographic features/outcrops not being spatially resolved in the models, (2) limited
constraints on ocean/atmosphere forcing which impact rate of retreat (e.g. Lowry et al., 2019) and (3) poorly constrained model
parameters that influence basal sliding, isostatic adjustment and ice flow/rheology which impact the rate of ice sheet response
to a climate forcing Lowry et al. (2020); Kingslake et al. (2018).





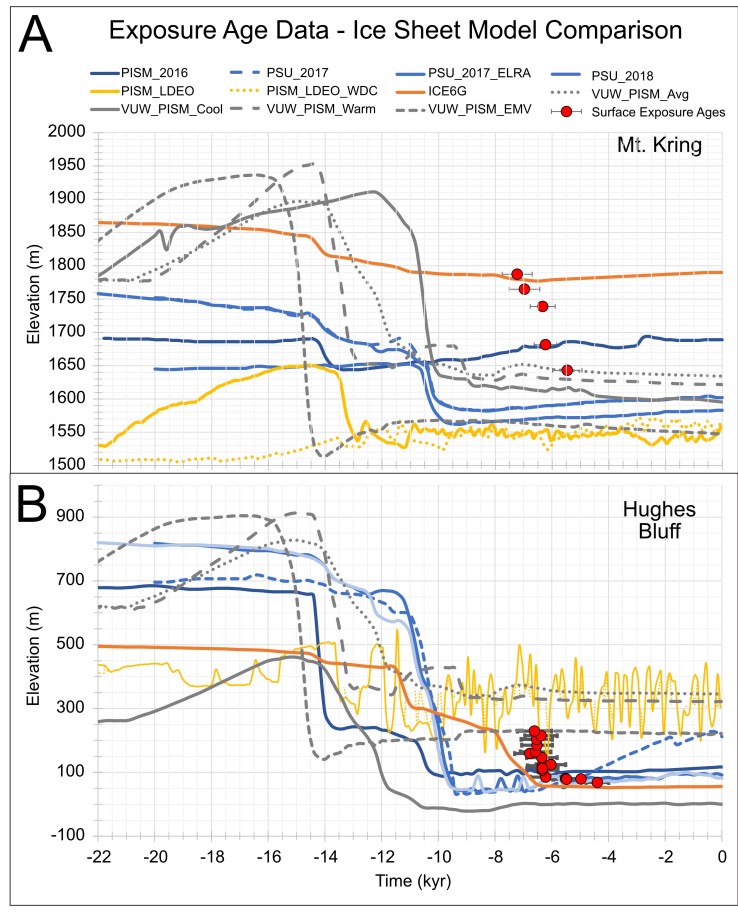

**Figure 8.** Surface exposure age data - ice sheet model comparisons for a) Mt. Kring and b) Hughes Bluff. Ice sheet modelled reconstruction data and supporting information from dmc.ice-d.org

During the LGM, ice core records and numerical model outputs suggest that the ice sheet experienced widespread interior in its interior, while coastal sites experienced extensive thickening (Verleyen et al., 2011; Mackintosh et al., 2014). Our data from Mt Kring show that the ice sheet was thicker at this site during the LGM, and recorded ∼200 metres of thinning during the Holocene. This provides a critical tie point between the high elevation, low accumulation ice domes where ice cores are drilled

and the low elevation, high accumulation coastal sites with more abundant geologic data. Mt. Kring is the first site along the high elevation margin of the EAIS to constrain LGM ice sheet behavior and represents a unique site to compare with other ice sheet reconstructions and continental-scale ice sheet modelling experiments.

## 5.2 Terrestrial-marine reconstruction

During the LGM, outlet glaciers and grounded ice thickened and expanded to the continental shelf edge surrounding Antarctica

(Livingstone et al., 2012; Anderson et al., 2014; Bentley et al., 2014). Since the LGM, the onshore thinning varied spatially, but



occurred primarily during the Holocene ((Small et al., 2019). Along the Transantarctic Mountains, Mid-Holocene outlet glacier thinning and retreat occurred during a relatively stable period of air temperature and sea level after the majority of post-glacial sea level rise and rising atmospheric temperatures occurred (Lambeck et al., 2014; Jones et al., 2015; Spector et al., 2017).

Glacial reconstructions using marine geological and geophysical data along the western Ross Sea focus on tracking a ground-
ing line that represents the ice sheet as a whole (Licht et al., 1996; Domack et al., 1999; Shipp et al., 1999; McKay et al., 2008; Anderson et al., 2014). However, there is increasing evidence that palaeo-ice streams and TAM outlet glaciers decoupled from the larger ice sheet occupying the Ross Embayment during deglaciation (Halberstadt et al., 2016; Lee et al., 2017). As the grounding line retreated southward, past TAM outlet glaciers, local subglacial topography is expected to have exerted a sig-nificant control on local glacier thinning and retreat rates (Jones et al., 2015). Marine geophysical and geological data reveal
a large grounding zone wedge (GZW) near Coulman Island, marking the probable LGM grounding line limit of ice in the Ross Sea (Shipp et al., 1999). Based on marine sedimentary analysis and radiocarbon age dating, the timing of retreat from the Coulman Island GZW initiated at ∼13 ka, punctuated by few short periods of staggered retreat through the deep Drygalski Trough (Licht et al., 1996; Domack et al., 1999; Anderson et al., 2014; Yokoyama et al., 2016). By ∼10 ka, [14]C dating of acid insoluble organic (AIO) matter from sediment cores, suggest the grounding line in the Ross Sea migrated south of the David
Glacier (Licht and Andrews, 2002). However, AIO dates in the Ross Sea are known to be unreliable due to potential input of anomalously old carbon by reworking, and this 10 ka constraint is likely to represent a maximum age of this retreat (Andrews et al., 1999; Rosenheim et al., 2013). Furthermore, short pauses in onshore thinning may temporally link to short periods of stability recorded by small grounding zone wedges deep in the Drygalski Trough and near Terra Nova Bay (Brancolini et al., 1995; Anderson et al., 2014; Lee, 2019).
Interpreted sub-ice shelf facies from marine sedimentary cores collected in the deepest parts of the inner Drygalski trough provide evidence of a lingering ice shelf, but anomalously old surface [14]C ages hinder coherent chronologies (Licht et al., 1996; Frignani et al., 1998; Domack et al., 1999; Licht and Andrews, 2002). North of Terra Nova Bay, meteoric [10]Be and compound specific radiocarbon ages capture the retreating calving line and onset of open marine conditions by 8 ka ((Yokoyama et al., 2016). This timing is consistent with the raised beach chronology of Baroni and Hall (2004), which marks the local onset of
coastal open marine conditions no later than 8 ka.

In summary, evidence from offshore David Glacier indicates that retreat of grounded ice through the Drygalski Trough and the formation of open marine conditions similar to today occurred immediately prior to the dynamic thinning of David Glacier recorded in this study. Together with the existing retreat chronology outlined above, our onshore surface exposure data records the final stages of glacial thinning and retreat along the David Glacier and wider Terra Nova Bay area.

## 5.3   Coastal thinning and impacts on local oceanography

Using continental-scale ice sheet model reconstructions and available geological data as a guide, the glacier thinning profile at Hughes Bluff likely only captures the final 200 metres of the approximately 400+ metre thinning since the LGM (Fig A2). In total, marine geological evidence, ice modelling and new results in this study suggest that at the LGM, the expanded David glacier had a surface profile above the level of Mt Kring and Hughes Bluff, and a grounding line that was pinned on the David





Fiord bathymetic sill. The subsequent modelled retreat to near modern configuration reproduces both the magnitude and rate
of onshore thinning derived from surface exposure data.

The new geological reconstruction of ice-surface elevation changes at Hughes Bluff shows a rapid lowering of the David
Glacier at 6.5 ka and a period of slow thinning from ∼6-4 ka. Minimal ice surface lowering since ∼6 ka likely records the
stability of the grounding line within the David Fjord near the modern configuration, which suggests that the Drygalski Ice

Tongue has been stable since ∼6 ka. Orombelli et al. (1990) and Baroni and Hall (2004) mapped a series of raised beaches along
the Terra Nova Bay coastline that mark beach depositional processes in an open ocean setting (e.g. no grounded ice) initiating
at 7.2 ka. A stable, thinned and retreated David Glacier since ∼6 ka is broadly consistent with a raised beach chronology
along the Terra Nova Bay coastline and suggests a long-term role for Drygalski Ice Tongue in sustaining the Terra Nova Bay
polynya as interpreted from modern paleo-oceanographic observations (Baroni and Hall, 2004; Stevens et al., 2017; Mezgec

et al., 2017).

### 5.4   Interior ice sheet thinning

Our modelling demonstrates that glacier thinning adjacent at Mt. Kring is sensitive to grounding line migration during ice
retreat from the outer to the inner continental shelf. The Mt. Kring site is part of a broad bedrock platform along the northern
flank of the major glacial trough dissecting the TAM (Fig. A1). The platform is comprised of three high elevation outcrops

protruding ∼200 metres above the local ice surface, with Mt. Kring nearest to the zone of streaming ice (e.g. surface velocities
∼100 m/yr). Geophysical characterisation of the northern TAM suggests the presence of individual tectonic blocks bounded by
faults which likely serve as zones of relatively weak rock strength allowing preferential ice flow and glacial erosion (Salvini and
Storti, 1999; Dubbini et al., 2010). Today in West Antarctica, dynamic thinning of the inland ice sheet is linked to underlying
tectonic controls (Bingham et al., 2012). At David Glacier, similar tectonic controls also conditioned the spatial pattern of

dynamic thinning during the Holocene.

Along the upper reaches of the David Fjord, there are no outcrops in the zone of highest surface velocity (>100 m/yr).
The closest outcrop to fast-flowing ice, Mt. Kring, lies ∼40 km from the modelled flowline. Modern surface velocities near
Mt. Kring are ∼15% of the surface velocities at the projected flowline position (Rignot et al., 2011). On modern ice sheets
experiencing dynamic thinning, satellite derived thinning estimates are largest in the centre of the ice streams or outlet glaciers,

and become progressively smaller at lower velocity sites further from and perpendicular to the central flowline. For example,
the central parts of Greenlands's outlet glaciers are currently thinning at rates of ∼0.84 m/yr, while marginal areas with slower
ice velocities are thinning at 0.12 m/yr (Pritchard et al., 2009). Therefore, it is likely that the Mt. Kring site likely reflects ice
stream marginal thinning (maximum of 0.17 m/yr, derived from surface exposure data) rather than the larger thinning rate that
was experienced in the centre of an ice stream (modelled maximum 0.3 m/yr). Taken together, the Hughes Bluff and Mt. Kring

chronologies suggest that ∼2 kyr of dynamic thinning occurred at David Glacier, and that this thinning propagated significantly
into the ice sheet interior.





## 5.5 Controls on thinning and grounding line migration

Overall, the style of modelled grounding line retreat appears to be controlled by MISI, a positive feedback where grounding line retreat into subglacial basins leads to progressively enhanced ice discharge and ice sheet thinning (Weertman et al., 1974; Mercer, 1978; Schoof, 2007). The topographic profile of the western Ross Sea is dominated by the deep and landward sloping Drygalski Trough and our modelling experiments show that the grounding line stabilises on a bathymetric sill at the mouth of the David Fjord. Experiments designed to simulate glacier-ice sheet decoupling show that once independent, the David Glacier grounding line rapidly retreats through the Dryglaski Trough to a temporary stable position near the TAM coastline.

This study provides insights into the processes that occurred as a large grounded section of an ice sheet retreated into discrete outlet glaciers. Previous descriptions of this retreat have focused on grounded ice in the Ross Sea as a whole (Licht et al., 1996; Shipp et al., 1999; Domack et al., 1999; McKay et al., 2008; Anderson et al., 2014). However, at the scale of David Glacier, retreat was likely influenced by local processes including interactions with adjacent glaciers and other ice bodies. For example, a recent ice sheet retreat reconstruction of Halberstadt et al. (2016) suggests that stable ice lingered on higher elevation banks as ice retreated in adjacent, large bathymetric troughs. Such lingering ice likely impacted lateral buttressing on David Glacier during retreat. Therefore, thinning and retreat histories of grounded ice resting on bank tops may influence ice stream lateral buttressing.

Knowledge of this complex lingering ice history is poorly constrained as the majority of research has focused on trough axes (Anderson et al., 2014; Halberstadt et al., 2016). Analysis of marine sediments suggests from such troughs indicate that a 'calving bay' environment formed during grounding line retreat (**?**Domack et al., 2003; Leventer et al., 2006; **?**). In this scenario, retreat is rapid along the trough axis while lingering ice remains along the lateral margins of the ice stream/glacier. In addition to sedimentary evidence, the abundant large iceberg keelmarks seaward of the Coulman Island GZW and smaller keelmarks within Terra Nova Bay provide evidence for a calving bay during deglaciation. Short-term grounding line stability of the expanded David Glacier may have been facilitated by complex interaction with other outlet glaciers (Reeves and Priestley glaciers - forming proto-Nansen Ice Sheet/Shelf) of Terra Nova Bay and/or grounded ice lingering on the banks surrounding the Drygalski Trough.

The modelled grounding line initially retreats to a location where a large GZW has been documented (Lee (2019)), thus providing an explanation for why a large GZW was deposited in this location. A lack of modelled grounding line stability at other, smaller GZWs suggests that the grounding-line may only have experienced very short stable periods in these locations. Further, in the case of the much smaller 'GZWs' observed in the deepest portion of the Drygalski Trough, the morphology may reflect a point source versus a line or zone source associated with more classically defined sheet-like GZWs. Regardless, the small mounds in the trough axis are likely to reflect Short-lived stable positions during overall grounding line retreat.

Generally, once initiated, the modelled David Glacier retreats rapidly to a stable grounding line position in ∼500 yrs, pausing at a prominent sill at the mouth of David Fjord for up to 5 kyr before subsequent grounding line migration led to its modern configuration. This simulated two-phase grounding line retreat compares well with our onshore reconstructions at Mt. Kring





and Hughes Bluff, both in terms of timing and rates of past glacier thinning. Models forced by moderate sub-ice-shelf melt rates and lateral buttressing reduction, fit best with onshore geologic constraints.

Figure 9 synthesise the results of the terrestrial thinning chronology, modelled glacier flowline behaviour and the existing regional marine retreat chronologies. This synthesis suggests that beginning at 7.5 kyr, with the grounding line pinned to the sill at the mouth of the David Fjord, the David Glacier and proto-Nansen Ice Shelf decouple and widespread onshore thinning

is initiated. Between 6-5.5 kyr, the grounding line retreats to near its modern position, thinning slows significantly and open marine conditions prevail regionally. Overall, as the grounding line retreats from the prominent bathymetric sill at the mouth of the David Fjord, David Glacier onshore thinning and decoupling from proto-Nansen Ice Shelf, and open marine conditions develop throughout the Terra Nova Bay.

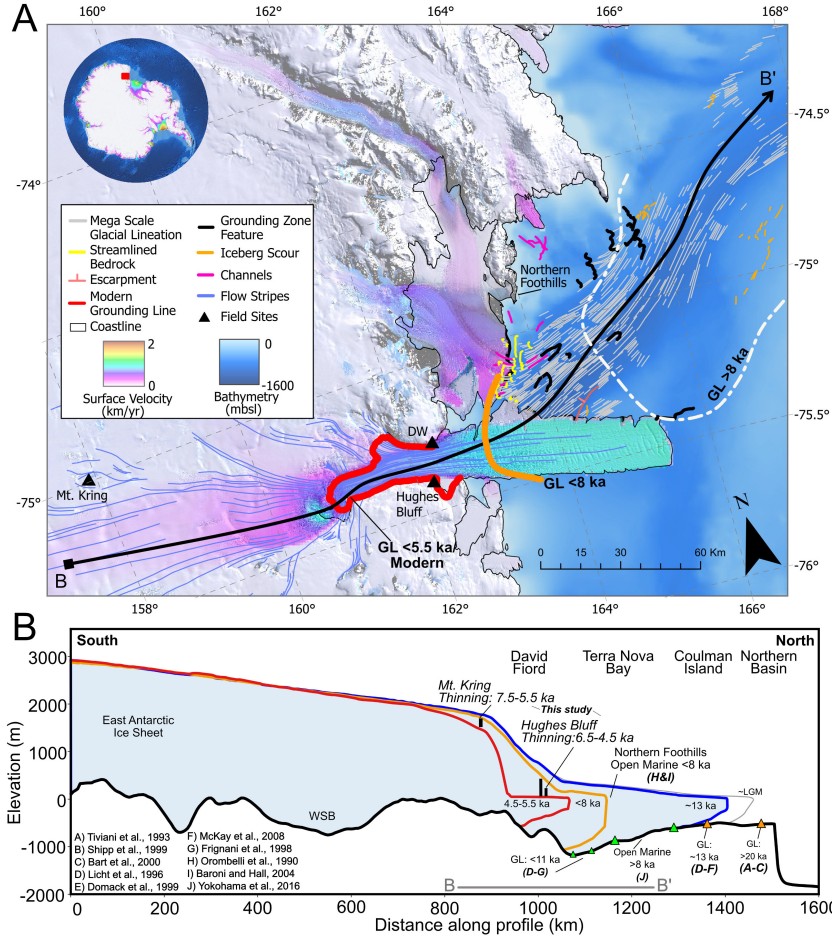

**Figure 9.** A) Map of Holocene thinning and retreat model for David Glacier, Terra Nova Bay. Synthesis map focused on two phase retreat between 11 ka and 5.5 ka, including geographic and geomorphic features mentioned in text. Inset with red box for main extent shows bathymetry and surface velocity. B) Synthesis profile focused on two phase retreat between 11 ka and 5.5 ka. Grounding zone features: well constrained (orange triangles), poorly constrained (green triangles). GL=Grounding line, SIS=Sub-Ice Shelf



## 6 Conclusions

Chronologies from the David Glacier reveal a period of glacier thinning along a large swath of the drainage basin during the Mid-Holocene (~6.5 ka). The reconstructed thinning style between two sites separated by ~130 km, reveals a dynamic thinning event that endured for two millennia. This chronology is synchronous with local, regional and continental scale geological records of ice sheet behaviour yet is not fully captured in continental scale ice sheet models. Our flowline modelling results suggest that thinning and grounding-line retreat was driven by increased sub-ice shelf melt rates and decreased ice stream lateral

buttressing, and that the combination of these two processes produces a powerful forcing that reduces the individual forcing thresholds required to initiate retreat. Modelled episodes of grounding line retreat correlate well with periods of onshore thinning, constrained by our high-resolution surface exposure ages. Data-model mismatches highlight enduring questions related to the relative role of local topographic pinning points and glacier-ice sheet decoupling on the nature of dynamic thinning, both observed and modelled.

Through careful collection of glacial deposits from numerous sites along the David Glacier, we have closed a spatial and temporal gap in the rapidly expanding onshore glacial geologic knowledge bank. Our data constrain one of the largest outlet glaciers in the world which carries regional significance for Victoria Land and the western Ross Sea, as well as offering clues about processes currently underway in rapidly changing sectors of Antarctica and Greenland. If the data and modelling presented in this study are representative of outlet glacier behaviour more generally, the ice sheet thinning observed by satellites

over the last ~40 years may be the first signs of a millennial-scale response to a warming world.

*Code availability.* Modelling code

The code used for flowline modelling is available by request from the corresponding author.

*Data availability.* Geochemical data and exposure ages

Field, lab, analytical and exposure age data are available on ICE-D online database (antarctica.ICE-D.org)

*Sample availability.* Samples collected during the 2016/17 austral summer are curated at Te Herenga Waka - Victoria University of Wellington and all metadata can be found in the Petlab database administered by GNS Science (https://pet.gns.cri.nz/). Samples collected prior to 2016/17 austral summer are curated at the University of Pisa.





# Appendix A: Appendix

## A1 Geomorphic context for field sites

For geomorphic context, we include geomorphic cross sections and field photos for Hughes Bluff and Mount Kring.

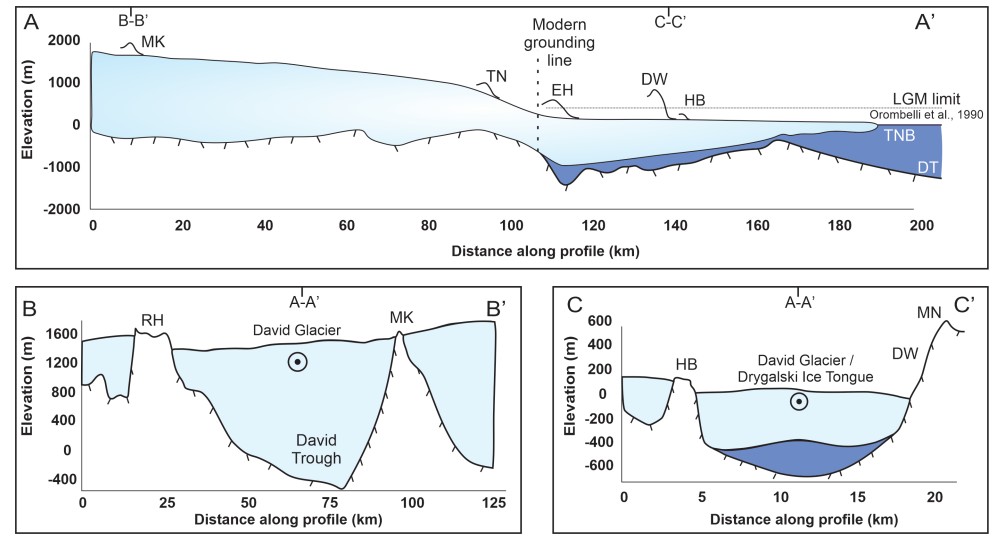

**Figure A1.** A) Flow parallel cross section showing relationship between field sites and modern surface and bed topography, B) Flow perpendicular cross section through Ricker Hills and Mt. Kring and C) Flow perpendicular cross section through Hughes Bluff and D'Urville Wall. Refer to fig 1 for profile location.



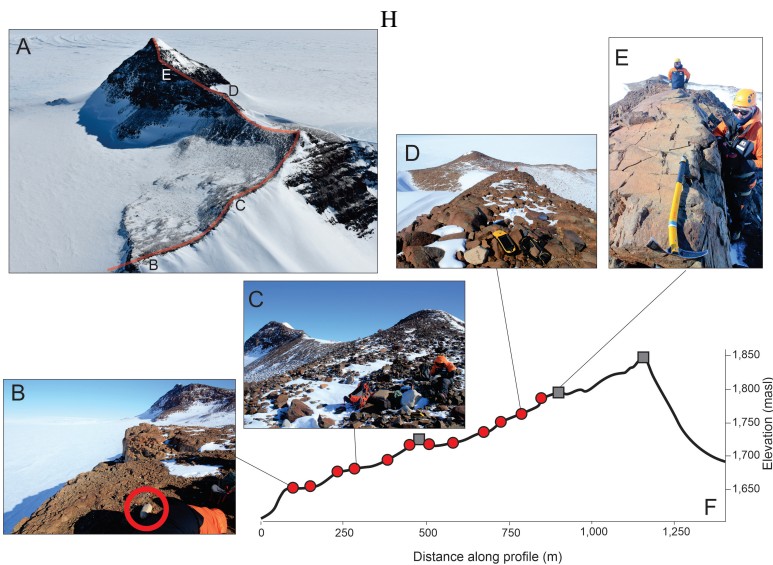

**Figure A3.** (A) Oblique aerial photo of Mt. Kring with approximate location of topographic profile (red), (B) West facing view of lowest elevation site with perched glacial erratics on bedrock, (C) West facing view of scattered glacial till draping bedrock ridge. (D) East facing view of scattered high elevation sandstone erratics perched on fractured bedrock, (E) East facing view from highest elevation striated bedrock outcrop, ice axe parallel to striation direction. (F) Topographic profile from SfMP showing representative sampling sites (orange circles).

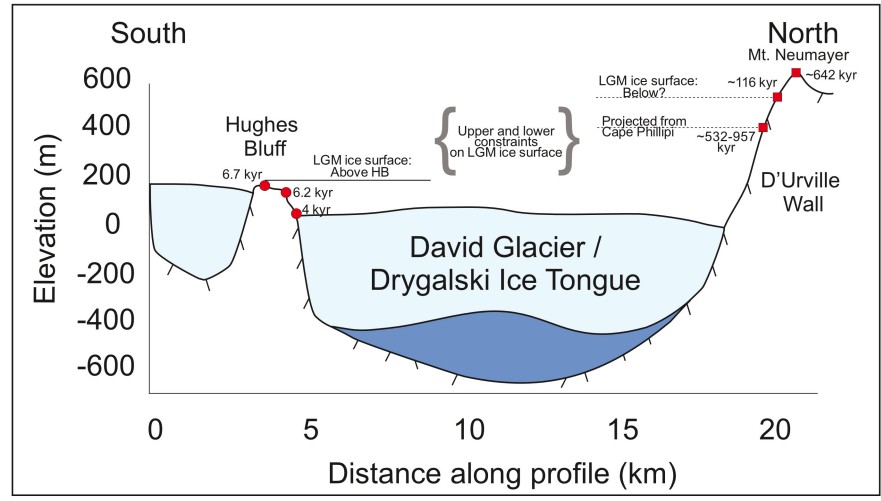

**Figure A2.** Cross section from Hughes Bluff to Mt. Neumayer combining the high-resolution chronology (red circles) from Hughes Bluff with $^{10}$Be derived exposure ages from high-elevation bedrock samples (red squares). Bedrock ages older than Holocene suggest cold-based ice cover and minimal erosion, and are used as constraints on upper ice surface elevation since the LGM. Elevation data source (Howat et al., 2019), bathymetry data source (Fretwell et al., 2013).



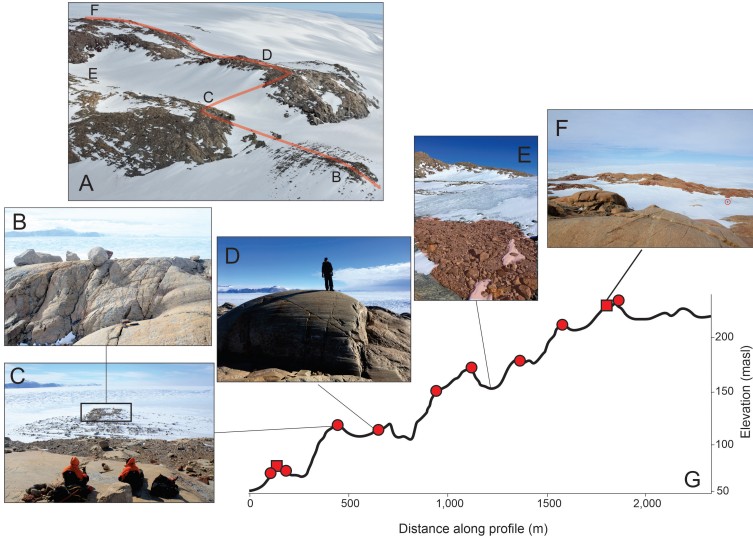

**Figure A4.** (A)Oblique aerial photo of Hughes Bluff with approximate location of topographic profile (red), (B) North facing view from low bedrock knob showing perched glacial erratics and location of lowest emergent bedrock knob, (C) Moulded, striated and grooved nature of bedrock at lowest outcrop of Hughes Bluff, (D) Example of spectacular rochés moutonnées observed throughout Hughes Bluff, (E) Example of 'cold till' observed in multiple localities along small scale channel forms (outlined in red), (F) Highest elevation outcrop of Hughes Bluff showing rounded summit morphology with glacial striations, view to east. (G) Topographic profile from SfMP showing representative sampling sites (red circles).



## A2   Model setup and Results

This section includes one table (Table A1 and three figures synthesising all modelling results.

| | Idealised Scenarios | | David Glacier Flowline Model Results | | | | |
| --- | --- | --- | --- | --- | --- | --- | --- |
| | 15 kyr model runs (spin up / forcing) | | Modelled Grounding Line | | Onshore Glacier Thinning | | |
| Exp. # | Melt Rate (m/yr) | Lat. Buttress Reduction (%) | Modelled retreat behaviour | Modelled Retreat Rate (m/yr) | Modelled thinning bahaviour | Max. thinning @ MK (m/yr) | Max. thinning @ HB (m/yr) |
| M1 | -1 / -1.5 | 0 | Stable Grounding Line | 0.439 | Minimal Thinning | 0.20 | 0.15 |
| M2 | -1 / -2 | 0 | Rapid retreat to sill | 356.5 | Moderate Thinning at HB, stable at MK | 0.20 | 0.58 |
| M3 | -1 / -11 | 0 | Two phase extreme retreat to modern GL position | 620.6 | Rapid runaway thinning at MK,Two phase rapid thinning above HB site, modern ice elevation | 1.02 | 8.73 |
| S1 | -1 | 0 / 0 | Stable Grounding Line | 0.439 | Minimal Thinning | 0.20 | 0.15 |
| S2 | -1 | 0 / 4 | Moderate retreat to sill | 191.7 | Moderate Thinning at HB, stable at MK | 0.20 | 0.44 |
| S3 | -1 | 0 / 40 | Two phase rapid retreat to modern GL position | 331.7 | Moderate runaway thinning at MK Rapid thinning above HB | 0.44 | 2.37 |
| MS1 | -1 / -5 | 0 / 40 | Two phase rapid retreat to modern GL position | 184.21 | Moderate runaway thinning at MK Rapid thinning above HB | 0.35 | 2.00 |
| MS2 | -1 / -7 | 0 / 30 | Two phase moderate retreat to modern GL position | 397.8 | Rapid runaway thinning at MK. Two phase rapid thinning above HB site, modern ice elevation by 2.5 kya | 0.36 | 2.23 |
| MS3 | -1 / -9 | 0 / 26 | Two phase moderate retreat to modern GL position | 489.3 | Rapid runaway thinning at MK Two phase rapid thinning above HB site, modern ice elevation by 2.5 kya | 0.39 | 2.58 |

**Table A1.** Experiment design for sub-ice shelf melt, lateral buttressing and combined scenarios. X / Y indicates initial conditions and forced conditions, respectively. Onshore and offshore modelled glacial behaviour, and thinning and grounding line retreat rates





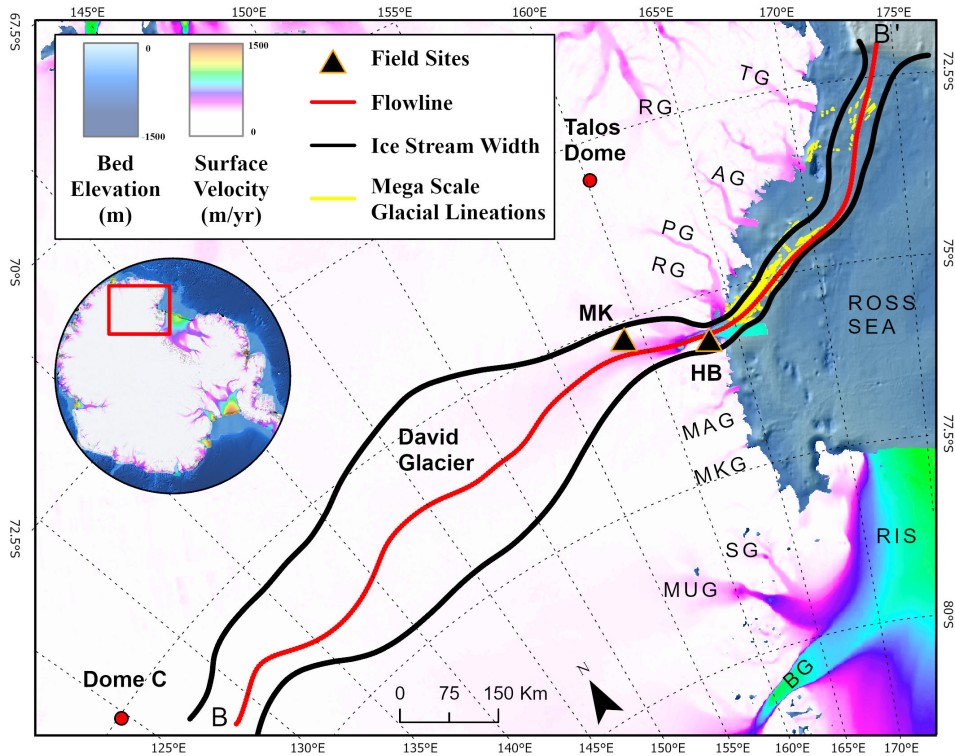

**Figure A1.** Map of model domain with modern surface velocity of (Rignot et al., 2011) and bathymetry of Arndt et al. (2013). Flowline follows red line, ice stream width in black.





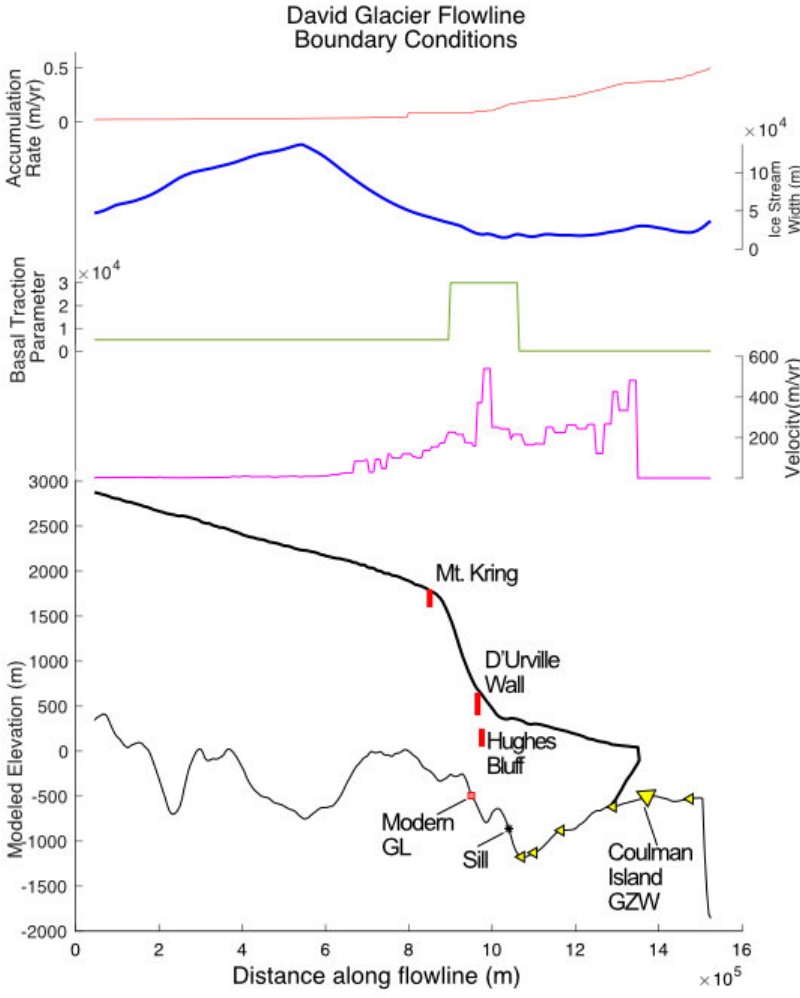

**Figure A2.** Boundary conditions along flowline distance used for deglacial modelling experiments in this study. Inputs include basal topography of Fretwell et al. (2013); Arndt et al. (2013), modelled surface and velocity of Whitehouse et al. (2012) and accumulation of van Wessem et al. (2018). Mapped grounding zone wedges, topographic features, modern day grounding line and terrestrial sites discussed in text are plotted along the upper ice surface and basal topography





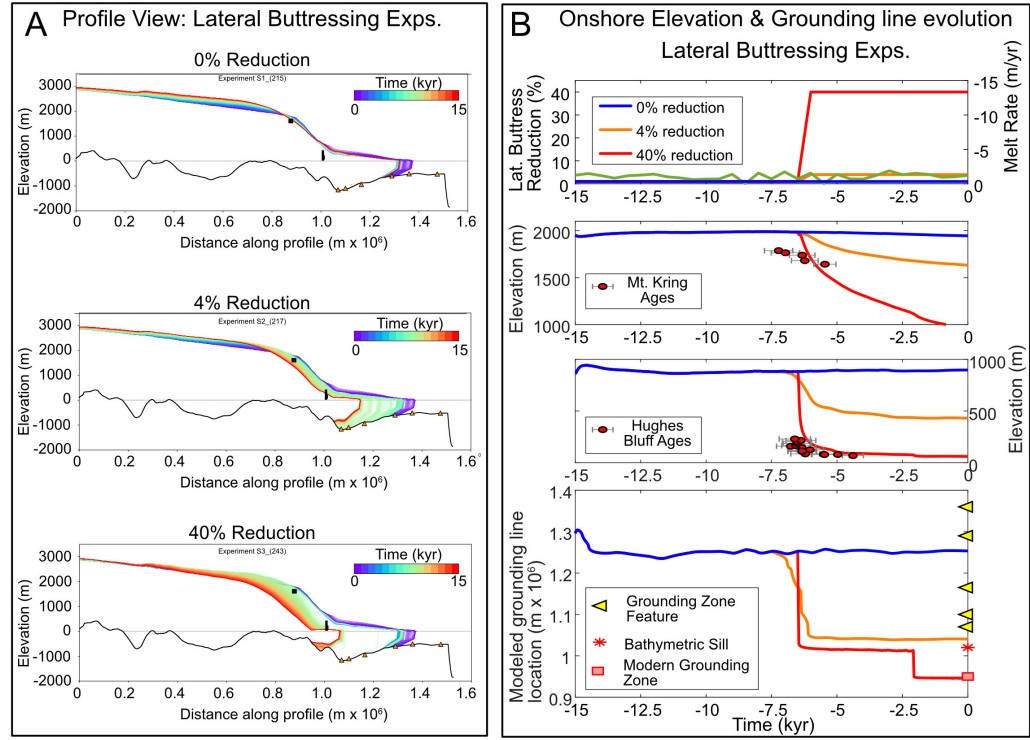

**Figure A3.** A) Profile View: 15 kyr evolution of modelled upper ice surface along flowline for lateral buttressing experiments. Black bars depict three sites mentioned in text. Orange triangles represent offshore features marking observed grounding zone wedges. B) Onshore elevation and grounding line evolution for lateral buttressing experiments. Top panel: Forcings applied, Top middle panel: Time-transgressive elevation profile for Mt. Kring with exposure ages, bottom middle panel: Time-transgressive elevation profile for Hughes Bluff surface exposure ages, and lower panel: Evolution of grounding line position with modern grounding line position, bathymetric sill at mouth of David Fjord and mapped grounding zone features.



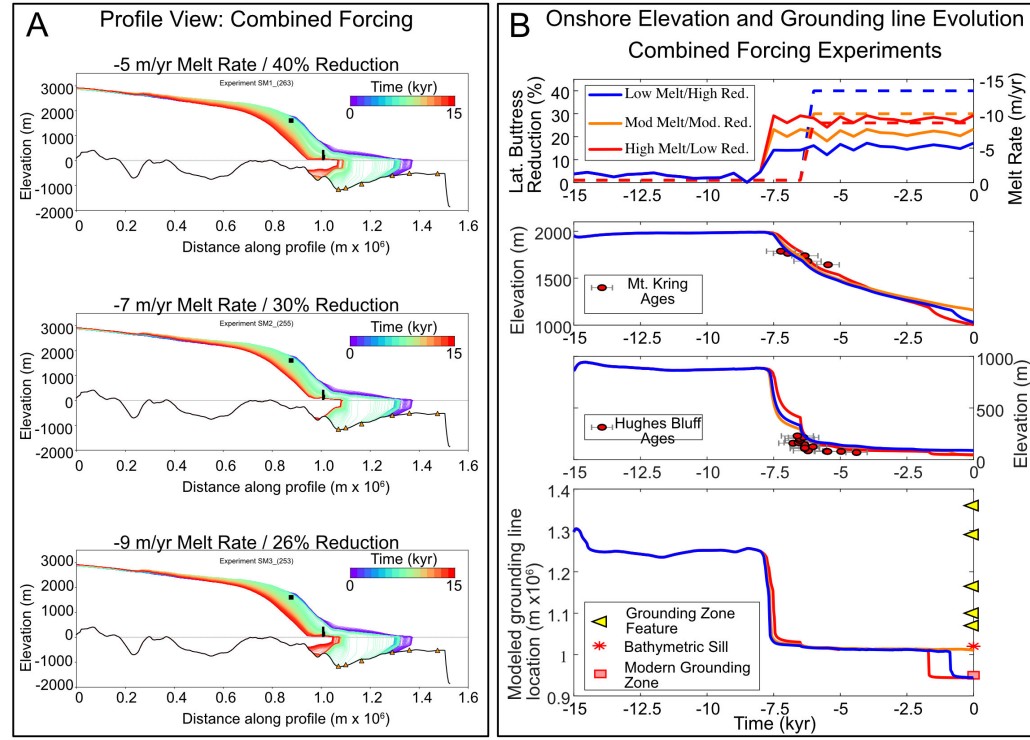

**Figure A4.** A) Profile View: 15 kyr evolution of modelled upper ice surface along flowline for combined forcing experiments. Black bars depict three sites mentioned in text. Orange triangles represent offshore features marking observed grounding zone wedges. B) Onshore elevation and grounding line evolution for lateral buttressing experiments. Top panel: Forcings applied, Top middle panel: Time-transgressive elevation profile for Mt. Kring with exposure ages, bottom middle panel: Time-transgressive elevation profile for Hughes Bluff surface exposure ages, and lower panel: Evolution of grounding line position with modern grounding line position, bathymetric sill at mouth of David Fjord and mapped grounding zone features.

*Author contributions.* JS contributed to project design, field work planning and implementation, sample analysis, modelling and preparation of manuscript. AM contributed to original project design, field work, modelling and manuscript preparation. KN contributed to original
project design, sample analysis and manuscript preparation. RW contributed to field and lab work. CB led previous regional field work, contributed data and helped prepare manuscript. MCS contributed to previous regional field work, contributed data and helped prepare manuscript. SC contributed to field and lab work. SJ led modelling work and contributed to preparation of manuscript. RSJ contributed to project design and modelling work. GB contributed to project design and BGC related lab work. PS contributed to Data-Model Comparison (DMC) work. JL contributed unpublished bathymetry data and regional marine geologic observations. YBS and HHR contributed to regional
synthesis. TW, LD, MI, FS, SIO, MC conducted AMS analyses at respective laboratories. LV contributed through preparation of SfMP models, modelling and manuscript preparation. DL conributed to DMC work. RM contributed to regional marine geology synthesis.



*Competing interests.* The authors would like to declare that no competing interests are present

*Acknowledgements.* We would like to acknowledge NZARI Grant(2017-1-3) for funding field work, AMS analyses and research visit to BGC. Support for a research visit to Durham University was provided by the Antarctic Science Bursary. We'd like to thank our trusty

mountaineer Bia Boucinhas and pilot Mark Hayes for safe passage during field work. Logistical support for field and lab work was expertly provided by Antarctica New Zealand, Italian National Antarctic Programme (PNRA), Korean Polar Research Institute (KOPRI), Southern Lakes helicopters, Kenn Borek Air, Antarctic Research Centre , School of Geography, Earth and Environmental Studies, and Te Herenga Waka - Victoria University of Wellington. We extend a great deal of gratitude for the many experienced researchers in the David Glacier area for providing field photos from past expeditions.





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
