# Peer review of "Mid-Holocene thinning of David Glacier, Antarctica: Chronology and Controls"

_The Cryosphere, 2020_

## Referee Comment (RC1) · Keir Nichols (Referee) · 16 Nov 2020

Please find below my review comments, as well as attached as a pdf (just in case this is more helpful for the authors, particularly with formatting that may not show up below)

General comments

Using 10Be and 3He exposure ages at multiple sites, Stutz et al. constrain the minimum LGM thickness and post-LGM thinning history of the David Glacier, one of the largest glaciers draining ice from the East Antarctic Ice Sheet into the Ross Sea. The paper adds to our knowledge of the past behaviour of the EAIS, filling in a large spatial gap. Through flowline modelling, the authors then explore the potential dominant mechanisms/forcings that could help explain the retreat and thinning history of the

glacier, informed by both their own constraints as well as marine evidence. The paper will be of great interest to both glacial geologists and numerical ice sheet modellers alike. I thoroughly enjoyed reading the paper and found it very informative and interesting. The paper is well written, logically structured, and the figures are of a high quality, making it easy to follow for the vast majority of it. I would not class any of my comments as "major". Most of my comments are requesting a little bit more information in a few parts of the paper, or minor technical corrections/suggestions. I recommend publication after addressing some points, listed below:

Specific comments (intermediate)

Sect. 2.1 - I think a short paragraph (either at the end of this section of the beginning of the next section, 2.2) describing how exactly the exposure ages inform the modelling approach would be helpful. I think at present it is a little unclear as to how the two are linked.

Additionally, I think a little bit more info on which parameters were varied in the sensitivity experiments, and how they were chosen, would be beneficial.

Could the authors produce a figure for the D'Urville Wall and Mt. Neumayer area similar to Figures 2 and 4? At present, section 3.3 comes as somewhat of a surprise, and is difficult to place spatially (though it is helpful that the location is shown in Figure A2. I must say that I very much like the supplementary figures).

The authors refer the reader to the online antarctica.ice-d.org database for nuclide concentrations and other information required to calculate the exposure ages reported. I think it would be beneficial to add a table to the supplement of this paper including both information that is already included in the ICE-D database (sample IDs, nuclide concentrations, samples thickness, shielding factor, etc.) as well as some information that is not. The latter would include (for Be) quartz mass, Be carrier mass, and the 10Be/9Be ratio (+ for process blank(s)).This information would be necessary if a reader were to want to redo the data reduction before recalculating exposure ages

independently.

Additionally, because the sample data is not included in a table in the paper, the only place to see which samples were analysed is in Figs 2 and 4. Because there is no figure showing the samples analysed for the D'Urville Wall and Mt. Neumayer, the reader cannot double check the exposure ages or recalculate them independently.

In the ICE-D database, there are no exposure ages or nuclide concentrations included for any of the samples from the D'Urville Wall site. Additionally, the D'Urville Wall site is named "Mt. Neumayer", whilst there is another separate section for the Mt. Neumayer samples.

Specific comments (minor/technical)

L 24 "Antarctic ice sheet" – this is the first mention of this phrase here, the authors could add "(AIS)" here rather than on line 28.

L 47 A space is needed after "Oscillation"

L 54 Are the references for the statements in this sentence the same as the next one (papers by Anderson and McKay)? If not, I think references may be needed here in line 54, otherwise please disregard.

L 59 "TAM" hasn't been defined yet. After defining it here in Line 59, you can remove "Transantarctic Mountains" in line 64 and replace with TAM.

L 76 "sampled" could be changed to "collected"

L 79 Should it be "using the structure from motion technique. . ."?

L 84 I think starting this sentence with the phrase "The aim of the sampling method is to track the upper ice surface. . ." would be more accurate.

L 90 I think some extra context would be useful at the end of this section. Why would bedrock be more useful for longer term exposure vs erratics?

L 95 How many etchings were done with the samples? A range would be useful.

L 101 (and reference list) The reference to Balter et al. (2020) can be updated from the Cryosphere Discussion paper to the final paper (Possibly Balter-Kennedy et al., 2020 now instead?).

L 101-102 Which nuclides were measured in these additional samples?

L 106-108 I think links to the online calculators, both the ice-tea one and that which has evolved from the Balco et al. (2008) paper, would be useful additions here.

Sect. 2.2.2 When the authors use the phrase "consistent with all existing geological constraints" (L 187) and "consistent with geologic constraints" (L 190), does this refer to the exposure ages produced by this study, prior geologic constraints, or both?

L 108 ( and reference list) Balco (2020) is referenced for the ICE-D database. In the reference list, the entry for Balco 2020 is for a study in the Annual Reviews journal, however, I think the paper the authors intend to reference is that in Geochronology (https://gchron.copernicus.org/articles/2/169/2020/).

L 206 Table number is missing here (also line 256).

L 246-247 "High elevation bedrock samples are much younger than exposure ages from nearby bedrock at similar height above the local ice surface" - Should the second part of the sentence read "from nearby erratics"? Otherwise, this sentence is a little confusing.

Sect. 4 L 252 I think one or two sentences briefly summarising the exposure age findings (timing and magnitude of thinning at the different sites) would make for a handy intro to this section. At present it feels like a jump to go from Sect. 3 to Sect.4, I think an additional sentence would help link them.

L 264 To help the reader follow, I would reiterate here that, as stated in L 144 – 146, "a reduction in lateral buttressing is expected as the expanded David Glacier and

grounded ice in the Ross Sea decouple"

L 295 "The reconstructed palaeo-thinning along the David Glacier during the mid-Holocene is synchronous with rapid thinning reconstructed at a number of sites in Antarctica"

In addition to citing the study by Small et al. (2019), I think it would be helpful to the reader to list and cite the sites around Antarctica which the authors have in mind here. In the abstract, the authors mention that the timing and rate of thinning at David Glacier is similar to reconstructions in the Amundsen and Weddell embayments, so I think it would be helpful to know the exact sites and records in those two regions.

L 313 – 318 I think a sentence or two on the rationale/motivation for the data model comparison may be helpful to the reader. Something on what the data model provides in the grand scheme of things (like helping to inform future modelling studies) could be useful. This may also help to link this part of the paper to the rest of the study.

On the same point, the paragraph at lines 331 to 357 covers what I think would be better suited to the start of this sub-section. I think this paragraph would be better placed prior to the data model comparison (so prior to line 313).

Additionally, I think the paper would flow better if the Palaeo-thinning rates and data-model comparison were separated into two sub sections. So 5.1 with the thinning rates, then 5.2 with the data model comparison.

L 324 "15-13 345 ka" should this be 15-13 ka?

L 331 "...widespread interior in its interior..." should possibly be widespread "thinning"?

L 387 Should it be adjacent "to" Mt. Kring, rather than adjacent "at"?

L 424 Two question marks here within the brackets – I imagine this might be two references missing due to a reference manager error?

[Figure]

Even though some of them may be obvious, I think some of the terms in equations 1-6 are not defined.

Coulman Island is mentioned a few times but is not included in any of the location figures (though the Coulman Island GZW is mentioned in Figure A2). If possible, labelling it in one of the earlier figures would be helpful – though I do not think this is a problem worth making an entirely new figure for. If it cannot be easily labelled in an existing figure, at the first mention in the text, the location could be described in a little more detail (e.g., XX km in XX direction from the DIT) to save from making a new figure just to add a label for one location.

Figures

Figure 1 caption – there is an "A)" at the start of the caption, but it appears to be the only part of the figure (i.e. no Figure 1B, C etc.).

I may have missed it in the text, but what is the source of the bathymetric features? The iceberg scour, grounding zone wedge etc. locations? If not mentioned in the text, I think this could be added to the caption (my apologies if I missed this in the text, though).

Figure 2 and 4: Changing the colour of the 3He exposure ages from grey to something else may help them stand out – at present they blend in with the colour of the ice

Figure 2 – It is not clear which samples in ICE-D match those with the sample IDs MtKring01px4-5, MtKring02px, 03px, and 03ol in Figure 2. MK04 is in Figure 2, but there are no ages or nuclide concentrations for this sample in ICE-D. Additionally, MK14 is a 10Be age but is grey, should it be red?

Figure 3 B "20" on the y axis, and "7.5" on the x axis are overlapping

Figure 9 My apologies if I have missed it, but SIS is defined in the figure caption, but I don't see SIS labelled in the figure.

[Figure]

I was a little confused by the appendix – is it meant to be split into two parts (the latter with the model setup and results)? At present there seems to be two Figs A1, 2, 3, and 4.

Figure A3 (first one) Orange circles – do the authors mean red circles? Also, the grey squares are not mentioned in the caption.

Figure A4 (first one) - The red squares are not mentioned in the caption.

Figure A1 (second one) This is not of huge importance, but I think Figure A1 would be more useful within the main text given the importance of the flowline model to the overall study. Also, location name abbreviations in the figure caption need to be defined (my apologies if they have been defined elsewhere and I missed them).

Please also note the supplement to this comment:
https://tc.copernicus.org/preprints/tc-2020-284/tc-2020-284-RC1-supplement.pdf

---

## Referee Comment (RC2) · James Lea (Referee) · 25 Nov 2020

In this paper Stutz et al. present a combination of geochronological and numerical model evidence for the glacial history of David Glacier and the potential drivers of its retreat.

I really like data/model comparison investigations like this study, and the paper includes some interesting results regarding the dynamics of the largest outlet glacier in Victoria Land. I have included detailed points for consideration by the authors below. In addition to these, as a general point, I think the findings of the paper would come through better if there was a clear separation between background/results/discussion in section 5. This may require some restructuring/rewording of the paper, but would really allow a

more concise discussion of the key results of the paper and their implications while communicating its overall findings more clearly.

L25 – (Weber et al., 2014)

L53-64 – there's a few names of locations mentioned that I'm unfamiliar with – if names of locations are mentioned they should be labelled on location figures

All figures – I would encourage the authors to ensure that all figures and their labels are at the very least red/green colour blind friendly to improve accessibility and interpretability

L79 – should state whether this from ground based photos or drones

L81 – There are two sets of figures A3 and A4 (p 24/25 and 30/31)

L86-90 – should include a supplementary table indicating location, type and (if available) geomorphological setting of samples that were collected, those that were analysed and information about results of analysis.

L119 – should make clear that by ice sheet flow, you're referring to the ice sheet interior rather than the entire domain

L149 – figure A1 (p28) – it would be worth having a panel showing a zoomed in view of the region around the grounding line so the transition from stream to shelf flow can be resolved in detail. A map of subglacial topography would be valuable in this area too to show how representative the ice stream width is of the trough where flow is most rapid.

Section 2.2.1 – the authors should expand on how width is defined in the model, especially in the regions where the grounding line is observed to be dynamic. Upstream definition of width is also important as defining the accumulation area and hence balance flux velocities. These are always tricky to define, but a bit of information about how they have been arrived at would be useful. Also, a table of key model parameters (e.g. grid size, ice T, ice density, proglacial water density etc) would be informative

L162-3 – this is where a zoomed in view around the modern grounding line would be useful for the reader

L174/Section 2.2.2 – some more info about the model spin up to LGM would be useful, i.e. is it tuned to the W12 configuration or is there a relaxation period from this? Also given that you're using W12 which was derived using the shallow ice approximation based GLIMMER model, are any mismatches between spun up configurations/velocities and the W12 configuration observed/expected. Given W12 was simulated on a 20 km grid this may be tricky to identify, depending on the along flow grid size that is being used in the flowline model. Are there reasons why W12 was chosen over other model simulations? If the model is struggling to replicate the steep descent from the interior, my gut feeling is that it may be due to a combination of too wide ice width and the SSA nature of the model that include longitudinal stresses. Without a map of the subglacial topography in this area however, it's tricky to say. It may also be a product of how bed/surface topography values have been input into the model and how the real world data have been summarised (i.e. whether they are a simple transect, or if they are width averaged). These points should be addressed if it is thought that they impact/have impacted the tuning of the model, and/or if it will impact the delivery of ice to the grounding line or significantly impact downstream ice thickness (i.e. have implications for the comparison of modelled results to observations).

L207 – Table number needs filling in

Section 3.3 – as earlier, place names referred to need to be labelled

L252 – this sentence dives straight into the detail, and would benefit from clarification as to whether the ice thinning is the observed or modelled thinning.

L256 – Table number

L257 – why were melt rates of -1.5, 2 and 11 m/yr chosen? If they were part of a larger ensemble of simulations (as indicated by the end of L259?) this is worth reporting. At

present the values chosen to be reported in the paper appear a bit arbitrary

L261 – how much above the Hughes Bluff site is the modelled ice surface?

L261/262 – are there criteria for what represents good agreement? If not, the difference between the reconstructed and simulated elevation should be included.

L264 – again, a bit of justification for the range of simulations presented would be good to have, in addition to the forcing value choices for the combined forcing simulations

L266-269 – check this sentence for grammar

Fig 6, A3, A4 (model simulations) – on the right hand panels, is the time axis appropriate in that I don't think the model is being forced by any date specific reconstructions?

L282-4 – need to be clear what exactly you mean by "match periods of onshore thinning" (linked to above comment). Although retreat occurs approx. -6.5kyr in model simulation time, it should be explained why it is anticipated/expected that this matches to "real world" years.

L287-8 – this should probably be referred to up front in the methods

L291-99 – I think these would go better in the results section, with any methods employed described there.

Figure 7 – the plots don't really give much of an impression as to the variability within the line cloud – is it possible to replot the lines but set a transparency on each so can get an impression of the distribution of the modelled uncertainty?

L313-324 – again, a clearer separation of results from the discussion would help

L313 – I would be very cautious of attempting to read too much into straight data/model comparisons without accounting for model grid size, flow approximations/model physics used, forcing and boundary conditions in the interpretation

L325 – magnitude instead of amplitude?

Section 5.1 – this would benefit from a sentence or so on what the motivation for undertaking the data/model comparison is. As it's not mentioned in the paper before it appears a bit out of the blue currently.

Section 5.2 – data presented in the paper are only written about in the last paragraph of this section, and otherwise is background info about the site.

L383 – if the ice tongue is grounded then definitely, however if it isn't then it could be that the upstream ice thickness is maintained in a scenario where the Drygalski Ice Tongue is lost (as its removal would not change the amount of buttressing). To demonstrate this for certain though would require a separate set of model experiments. Unless there is other evidence for the Drygalski Ice Tongue being a permanent feature since 6ka BP I would still be cautious about linking it to the Terra Nova Bay polynya.

L389-403 – most of this is site description rather than discussion

L408 – write out full abbreviation of MISI

L416-7 – if this is the case it should be acknowledged/alluded to when the definition of the model domain is described.

L422-30 – more site description than discussion of results

L463-465 – this is quite a bold statement, and it is a bit of a leap to say that the results of this study show this conclusively.

---

## Author Comment (AC1) · 9 Feb 2021

**Keir Nichols (Referee)** knichol3@tulane.edu

Please find below my review comments, as well as attached as a pdf (just in case this is more helpful for the authors, particularly with formatting that may not show up below)

General comments

Using 10Be and 3He exposure ages at multiple sites, Stutz et al. constrain the minimum LGM thickness and post-LGM thinning history of the David Glacier, one of the largest glaciers draining ice from the East Antarctic Ice Sheet into the Ross Sea. The paper adds to our knowledge of the past behaviour of the EAIS, filling in a large spatial gap. Through flowline modelling, the authors then explore the potential dominant mechanisms/forcings that could help explain the retreat and thinning history of the

glacier, informed by both their own constraints as well as marine evidence. The paper will be of great interest to both glacial geologists and numerical ice sheet modellers alike. I thoroughly enjoyed reading the paper and found it very informative and interesting. The paper is well written, logically structured, and the figures are of a high quality, making it easy to follow for the vast majority of it. I would not class any of my comments as "major". Most of my comments are requesting a little bit more information in a few parts of the paper, or minor technical corrections/suggestions. I recommend publication after addressing some points, listed below:

Specific comments (intermediate)

*We thank Dr. Nichols for their clear and thoughtful review, and offer our responses below each comment in italics.*

Sect. 2.1 - I think a short paragraph (either at the end of this section of the beginning of the next section, 2.2) describing how exactly the exposure ages inform the modelling approach would be helpful. I think at present it is a little unclear as to how the two are linked.

*We tried to highlight this in the first sentence in section 4 'Results: Glacier Modelling' but we agree this can be expanded and emphasised in Section 2. We will highlight the conditions required of a geometric fit (i.e. the initial ice surface covers the site of interest prior to thinning). However, this is quite difficult to do without first presenting the mid-Holocene exposure ages. For this final reason, we highlight how the exposure ages focus our modelling*

[Figure]

*during the thinning period identified in the exposure ages and data model comparison figures (Section 3 Results-chronology).*

Additionally, I think a little bit more info on which parameters were varied in the sensitivity experiments, and how they were chosen, would be beneficial.

*We expect this comment refers to modern sensitivity experiments. We use the published in situ and satellite data (e.g. ice sheet upper surface, bed, accumulation and velocity) to support our modern sensitivity experiments, those were held constant, helping to hone in on a suitable basal traction condition that best matched the modern configuration (upper ice surface and grounding line position). We did not vary sub-ice shelf melt rate or lateral buttressing parameters for these modern experiments. If this comment applies to deglacial sensitivity experiments we primarily focused on sub ice shelf melt rate and lateral buttressing, as regional proxies for internal ice temperature and accumulation show relatively minor variation, therefore we did not focus on these parameters. We will add these details near L200 around at "Using an optimised set of accumulation and temperature forcings…*

*As with the previous comment, we will highlight this further in section 4 as we first should present the chronology which then focuses us in the mid-Holocene. We will also include a comment here regarding limitations of this model and the parameters it includes. Additionally, we intend to supply a table in the supplement showing exactly how we vary parameters in sensitivity experiments.*

[Figure]

Could the authors produce a figure for the D'Urville Wall and Mt. Neumayer area similar to Figures 2 and 4? At present, section 3.3 comes as somewhat of a surprise, and is difficult to place spatially (though it is helpful that the location is shown in Figure A2. I must say that I very much like the supplementary figures).

*Yes, we are happy to do this.*

The authors refer the reader to the online antarctica.ice-d.org database for nuclide concentrations and other information required to calculate the exposure ages reported. I think it would be beneficial to add a table to the supplement of this paper including both information that is already included in the ICE-D database (sample IDs, nuclide concentrations, samples thickness, shielding factor, etc.) as well as some information that is not. The latter would include (for Be) quartz mass, Be carrier mass, and the 10Be/9Be ratio (+ for process blank(s)).This information would be necessary if a reader were to want to redo the data reduction before recalculating exposure ages.

*Yes, we will add supplemental tables (in .xls format) for sample information as well as sample analytical data.*

[Figure]

Additionally, because the sample data is not included in a table in the paper, the only place to see which samples were analysed is in Figs 2 and 4. Because there is no figure showing the samples analysed for the D'Urville Wall and Mt. Neumayer, the reader cannot double check the exposure ages or recalculate them independently.

In the ICE-D database, there are no exposure ages or nuclide concentrations included for any of the samples from the D'Urville Wall site. Additionally, the D'Urville Wall site is named "Mt. Neumayer", whilst there is another separate section for the Mt. Neumayer samples.

*This is mislabelled will be fixed for clarity.*

Specific comments (minor/technical)

L 24 "Antarctic ice sheet" – this is the first mention of this phrase here, the authors could add "(AIS)" here rather than on line 28. *Thank you, noted.*

L 47 A space is needed after "Oscillation" *Agree, noted*

L 54 Are the references for the statements in this sentence the same as the next one (papers by Anderson and McKay)? If not, I think references may be needed here in line 54, otherwise please disregard.

*The references are different. The sentences on L53-54 needs updated references and we will add these (Licht et al., 1996, Domack et al., 1999 and McKay et al., 2008)*

L 59 "TAM" hasn't been defined yet. After defining it here in Line 59, you can remove "Transantarctic Mountains" in line 64 and replace with TAM. *Agree, noted.*

[Figure]

L 76 "sampled" could be changed to "collected". *Agree, noted.*

L 79 Should it be "using the structure from motion technique…"? *Agree, noted.*

L 84 I think starting this sentence with the phrase "The aim of the sampling method is to track the upper ice surface…" would be more accurate. *Agree, noted.*

L 90 I think some extra context would be useful at the end of this section. Why would bedrock be more useful for longer term exposure vs erratics?

*Noted. While bedrock is not the focus of this study, we will add "Exposure ages from bedrock is useful for understanding longer term exposure histories and duration due to recognition of non-erosive burial by cold-based ice (e.g. Atkins et al., 2013; Joy et al., 2014).*

L 95 How many etchings were done with the samples? A range would be useful.

*We will expand this to include "Two etchings in total: One day etching at 2.5% HF and a multi-day etching at 1% HF"*

L 101 (and reference list) The reference to Balter et al. (2020) can be updated from the Cryosphere Discussion paper to the final paper (Possibly Balter-Kennedy et al., 2020 now instead?). *Agree, noted.*

[Figure]

L 101-102 Which nuclides were measured in these additional samples?

*Be and He as indicated. We will clarify this for the additional samples.*

L 106-108 I think links to the online calculators, both the ice-tea one and that which has evolved from the Balco et al. (2008) paper, would be useful additions here. *Agree, noted.*

Sect. 2.2.2 When the authors use the phrase "consistent with all existing geological constraints" (L 187) and "consistent with geologic constraints" (L 190), does this refer to the exposure ages produced by this study, prior geologic constraints, or both?

*It refers to both. W12 fits well with all geological constraints (prior to 2012 publication) and the modelled initial ice surface lies above our highest elevation Holocene aged erratics. We will clarify this in the text by including after L182: "W12 is chosen as, at the time of its publication, fits well with all existing geological constraints" Further on L190, we will include "the modelled initial ice surface lies above our highest elevation Holocene aged erratics"*

L 108 ( and reference list) Balco (2020) is referenced for the ICE-D database. In the reference list, the entry for Balco 2020 is for a study in the Annual Reviews journal, however, I think the paper the authors intend to reference is that in Geochronology (https://gchron.copernicus.org/articles/2/169/2020/). *Agree, noted.*

L 206 Table number is missing here (also line 256). *Agree, noted.*

[Figure]

L 246-247 "High elevation bedrock samples are much younger than exposure ages from nearby bedrock at similar height above the local ice surface" - Should the second part of the sentence read "from nearby erratics"? Otherwise, this sentence is a little confusing.

*We will clarify this and include a plot of bedrock samples measured in this study and their position on the landscape relative to other bedrock samples from previous studies (e.g. Ricker Hills, Strasky et al., 2007,2009 and NVL, Di Nicola et al., 2012) as a way to contextualise the bedrock data without further nuclide measurements. This is not the focus of the study but a comparison of local bedrock data will provide some context from higher elevation sites along the David Glacier.*

Sect. 4 L 252 I think one or two sentences briefly summarising the exposure age findings (timing and magnitude of thinning at the different sites) would make for a handy intro to this section. At present it feels like a jump to go from Sect. 3 to Sect.4, I think an additional sentence would help link them.
*Agree, noted.*

[Figure]

L 264 To help the reader follow, I would reiterate here that, as stated in L 144 – 146, "a reduction in lateral buttressing is expected as the expanded David Glacier and

grounded ice in the Ross Sea decouple" *Agree, noted.*

L 295 "The reconstructed palaeo-thinning along the David Glacier during the mid-Holocene is synchronous with rapid thinning reconstructed at a number of sites in Antarctica"

[Figure]

In addition to citing the study by Small et al. (2019), I think it would be helpful to the reader to list and cite the sites around Antarctica which the authors have in mind here. In the abstract, the authors mention that the timing and rate of thinning at David Glacier is similar to reconstructions in the Amundsen and Weddell embayments, so I think it would be helpful to know the exact sites and records in those two regions.

*We agree and it will also give better credit to those studies that preceded ours.*

L 313 – 318 I think a sentence or two on the rationale/motivation for the data model comparison may be helpful to the reader. Something on what the data model provides in the grand scheme of things (like helping to inform future modelling studies) could be useful. This may also help to link this part of the paper to the rest of the study.

*Agree. In fact, the exposure ages and the DMC all setup the rationale for our flowline modelling. For this, we propose to move the DMC section from Discussion to Results: specifically, the Chronology section starting on L250.*

On the same point, the paragraph at lines 331 to 357 covers what I think would be better suited to the start of this sub-section. I think this paragraph would be better placed prior to the data model comparison (so prior to line 313). *Agree. We will move L331-357 to follow L318.*

Additionally, I think the paper would flow better if the Palaeo-thinning rates and data-model comparison were separated into two sub sections. So 5.1 with the thinning rates, then 5.2 with the data model comparison.

*Thank you, we agree. As mentioned before, we will move this section into results after L250.*

L 324 "15-13 345 ka" should this be 15-13 ka? *Agree, noted.*

[Figure]

L 331 "…widespread interior in its interior…" should possibly be widespread "thinning"? Agree, noted.

L 387 Should it be adjacent "to" Mt. Kring, rather than adjacent "at"? *Agree, noted. We will remove adjacent and keep 'at'*

L 424 Two question marks here within the brackets – I imagine this might be two references missing due to a reference manager error? *Yes this is an error. Thanks.*

[Figure]

Coulman Island is mentioned a few times but is not included in any of the location figures (though the Coulman Island GZW is mentioned in Figure A2). If possible, labelling it in one of the earlier figures would be helpful – though I do not think this is a problem worth making an entirely new figure for. If it cannot be easily labelled in an existing figure, at the first mention in the text, the location could be described in a little more detail (e.g., XX km in XX direction from the DIT) to save from making a new figure just to add a label for one location.

*We will label it on the model map (second fig A1-which will be relabelled as A5).*

Figures

Figure 1 caption – there is an "A)" at the start of the caption, but it appears to be the only part of the figure (i.e. no Figure 1B, C etc.). *Noted, will remove 'A)'*

I may have missed it in the text, but what is the source of the bathymetric features? The iceberg scour, grounding zone wedge etc. locations? If not mentioned in the text, I think this could be added to the caption (my apologies if I missed this in the text, though). *The author mapped these features using GeoMapApp based on analogs and experience from MSc research. We will indicate the mapping method, type of data, spatial resolution and include link to GeoMapApp GMRT dataset as well as analogs (after Dowdeswell et al., 2016 10.1144/M46.171) in L138.*

Figure 2 and 4: Changing the colour of the 3He exposure ages from grey to something else may help them stand out – at present they blend in with the colour of the ice.

[Figure]

*We'd prefer to keep standard colors for nuclides (Grey for $^3$He as suggested here: https://cosmognosis.wordpress.com/2018/10/08/what-color-is-beryllium-10/). Also to emphasise Holocene aged samples, the focus of the study. We will ensure the grey boxes stand out better.*

Figure 2 – It is not clear which samples in ICE-D match those with the sample IDs MtKring01px4-5, MtKring02px, 03px, and 03ol in Figure 2. MK04 in Figure 2, but there are no ages or nuclide concentrations for this sample in ICE-D. Additionally, MK14 is a 10Be age but is grey, should it be red?

*We will fix the labels in fig 2 to match ICE-D. MK04 is mislabelled on the map and should be MK03. The samples are from the same location and only MK03 has been measured for 10Be. MK14 is plotted beneath MK13 (again, same location). We will ensure the Be derived age is visible.*

Figure 3 B "20" on the y axis, and "7.5" on the x axis are overlapping. *Agree, noted.*

[Figure]

Even though some of them may be obvious, I think some of the terms in equations 1-6 are not defined. *Noted. We will ensure all terms are appropriately defined.*

Figure 9 My apologies if I have missed it, but SIS is defined in the figure caption, but I don't see SIS labelled in the figure.
*We will remove SIS as an earlier version of this figure included SIS.*

I was a little confused by the appendix – is it meant to be split into two parts (the latter with the model setup and results)? At present there seems to be two Figs A1, 2, 3, and 4.

*Appendix should probably be a supplement. There are two sets of appendix figures labelled A1-3. This is a latex derived plotting error and will be fixed.*

Figure A3 (first one) Orange circles – do the authors mean red circles? Also, the grey squares are not mentioned in the caption. *Agree, noted*

Figure A4 (first one) - The red squares are not mentioned in the caption. *Agree, noted*
Figure A1 (second one) This is not of huge importance, but I think Figure A1 would be more useful within the main text given the importance of the flowline model to the overall study. Also, location name abbreviations in the figure caption need to be defined (my apologies if they have been defined elsewhere and I missed them).
*We agree and will include it in main text as well as define the abbreviations in the caption.*

[Figure]

Please also note the supplement to this comment: https://tc.copernicus.org/preprints/tc-2020-284/tc-2020-284-RC1-supplement.pdf.

---

## Author Comment (AC2) · 9 Feb 2021

In this paper Stutz et al. present a combination of geochronological and numerical model evidence for the glacial history of David Glacier and the potential drivers of its retreat.

I really like data/model comparison investigations like this study, and the paper includes some interesting results regarding the dynamics of the largest outlet glacier in Victoria Land. I have included detailed points for consideration by the authors below. In addition to these, as a general point, I think the findings of the paper would come through better if there was a clear separation between background/results/discussion in section 5. This may require some restructuring/rewording of the paper, but would really allow a

more concise discussion of the key results of the paper and their implications while communicating its overall findings more clearly.

[Figure]

*We sincerely thank referee James Lea for their thoughtful and detailed review of this manuscript. We offer our responses below each comment in italics. We agree with this general comment and will provide a clearer separation between the background, results and discussion in the revised paper. We acknowledge that the discussion does contain suitable material for the background section but we feel is better suited in its current place in line with the major discussion points. We propose to move Discussion section 5.1 to the results section 3 (after L250).*

L25 – (Weber et al., 2014). *Agree, noted*

[Figure]

L53-64 – there's a few names of locations mentioned that I'm unfamiliar with – if names of locations are mentioned they should be labelled on location figures

*This is highlighted by the other referee and we agree. We will include all appropriate place names on maps and map insets.*

All figures – I would encourage the authors to ensure that all figures and their labels are at the very least red/green colour blind friendly to improve accessibility and interpretability

*We agree in principle but we prefer to keep the existing colours for samples/data because they follow an effort to standardise colours in the surface exposure dating community. In our maps, surface ice velocity is typically in a rainbow colourmap, but this will be changed in the figures. For modelling results, the rainbow pattern does help to highlight the various phases during retreat but we will ensure red/green colour blind friendly where applicable by ensuring that colours are not superimposed.*

L79 – should state whether this from ground based photos or drones.

*Agree. Will indicate this is photography from a helicopter.*

L81 – There are two sets of figures A3 and A4 (p 24/25 and 30/31). *Agree, noted.*

L86-90 – should include a supplementary table indicating location, type and (if available) geomorphological setting of samples that were collected, those that were analysed and information about results of analysis. *Agree, noted. Will include data tables in .xls format as supplementary data.*

[Figure]

L119 – should make clear that by ice sheet flow, you're referring to the ice sheet interior rather than the entire domain. *Agree, noted.*

L149 – figure A1 (p28) – it would be worth having a panel showing a zoomed in view of the region around the grounding line so the transition from stream to shelf flow can be resolved in detail. A map of subglacial topography would be valuable in this area too to show how representative the ice stream width is of the trough where flow is most rapid.

*We agree this is an important area to show detail. Fig. A1 (p24) is meant to convey both the transition from stream to shelf flow as well as provide along and cross flow cross sections of topography/bathymetry. We will highlight this and reference this figure.*

Section 2.2.1 – the authors should expand on how width is defined in the model, especially in the regions where the grounding line is observed to be dynamic. Upstream definition of width is also important as defining the accumulation area and hence balance flux velocities. These are always tricky to define, but a bit of information about how they have been arrived at would be useful.
*Agree, we will expand on the methodology for determining basin dimensions in this section.*

Also, a table of key model parameters (e.g. grid size, ice T, ice density, proglacial water density etc) would be informative.
*Agree. We will include this information (in table form) alongside equations 1-6*

[Figure]

L162-3 – this is where a zoomed in view around the modern grounding line would be useful for the reader.

*We agree that a zoomed in view around the modern grounding line is useful. We think that Fig. A1 (p28) provides a reasonable scale view and context for the modern grounding line and surrounding regions, and would prefer not to generate another figure unless strictly necessary.*

L174/Section 2.2.2 – some more info about the model spin up to LGM would be useful, i.e. is it tuned to the W12 configuration or is there a relaxation period from this?

Also given that you're using W12 which was derived using the shallow ice approximation based GLIMMER model, are any mismatches between spun up configurations/velocities and the W12 configuration observed/expected. Given W12 was simulated on a 20 km grid this may be tricky to identify, depending on the along flow grid size that is being used in the flowline model. Are there reasons why W12 was chosen over other model simulations? If the model is struggling to replicate the steep descent from the interior, my gut feeling is that it may be due to a combination of too wide ice width and the SSA nature of the model that include longitudinal stresses. Without a map of the subglacial topography in this area however, it's tricky to say. It may also be a product of how bed/surface topography values have been input into the model and how the real world data have been summarised (i.e. whether they are a simple transect, or if they are width averaged). These points should be addressed if it is thought that they impact/have impacted the tuning of the model, and/or if

it will impact the delivery of ice to the grounding line or significantly impact downstream ice thickness (i.e. have implications for the comparison of modelled results to observations).

*Most Antarctic deglacial simulations do not attempt to fit to all available geological constraints, and other alternatives that did fit to constraints are coarse resolution (e.g. Briggs et al., 2014, 40 km). We mainly were interested in a model that fit to all geological constraints and thus provided a reasonable starting point in which to model the upper ice surface. W12 is on an old bed topography, has a lower spatial resolution and is solved using the shallow ice approximation, so we should not expect it to match our surface profiles – it's purely a starting point for the model from which our model equilibrates as it adjusts to the boundary conditions, parameters and physics of our flowline model. We will expand and clarify this in section 2.2.2.*

L207 – Table number needs filling in. *Agree, noted.*

Section 3.3 – as earlier, place names referred to need to be labelled. *Agree, noted.*

L252 – this sentence dives straight into the detail, and would benefit from clarification as to whether the ice thinning is the observed or modelled thinning. *Agree, noted. We feel that by moving discussion section 5.1 to ~L250 will help us explain our motivation for undertaking the modelling work.*

L256 – Table number *Agree, noted.*

[Figure]

L257 – why were melt rates of -1.5, 2 and 11 m/yr chosen? If they were part of a larger ensemble of simulations (as indicated by the end of L259?) this is worth reporting. At present the values chosen to be reported in the paper appear a bit arbitrary

*We agree that we should include an explanation that we progressively increased melt rate until partial to full retreat is initiated. Further, we will add a table of parameter values and experiments in the supplement.*

L261 – how much above the Hughes Bluff site is the modelled ice surface?

*300 m above modelled ice surface, Fig A2 (pg29).*

[Figure]

L261/262 – are there criteria for what represents good agreement? If not, the difference between the reconstructed and simulated elevation should be included.

*Good agreement means the modelled ice surface at the end of the simulation lies slightly below lowest collected erratic. We will include the difference in the text but do not see much value in this as we do have a discussion of final modelled upper ice surface (particularly for Mt. Kring) in Section 5.4.*

L264 – again, a bit of justification for the range of simulations presented would be good to have, in addition to the forcing value choices for the combined forcing simulations

*We agree. A table of parameters and listing the different experiments will be included in the supplement*

L266-269 – check this sentence for grammar. *Noted*

Fig 6, A3, A4 (model simulations) – on the right hand panels, is the time axis appropriate in that I don't think the model is being forced by any date specific reconstructions?

*The model is not forced by a date-specific reconstruction, but it is plotted in model years to allow general comparison with cosmogenic ages.*

[Figure]

L282-4 – need to be clear what exactly you mean by "match periods of onshore thinning" (linked to above comment). Although retreat occurs approx. -6.5kyr in model simulation time, it should be explained why it is anticipated/expected that this matches to "real world" years.

*We agree that this can be clearer. We will highlight the geometric fit and improve on what appears to be a chronological fit. The modelled period is 15,000 years with spin up during the first 7,500 years. This approach approximates the timescale for change following the Antarctic Cold Reversal and main phase of deglaciation in Antarctica. It is not meant to reflect 'real world' years but simply serves as a common timescale in which to compare against our thinning chronology. "match periods of onshore thinning" refers to the simultaneous upper ice surface elevation and grounding line location being consistent with onshore thinning (e.g. upper ice surface is below the lowest/youngest erratic at each site). This is a geometric fit and we will highlight this point. The fundamental take-home point is that the upper ice surface lies above the Hughes Bluff site when the grounding line is pinned to the sill at the outlet and the resulting modelled retreat over this sill is responsible for the thinning history deduced from our chronologies.*

L287-8 – this should probably be referred to up front in the methods. *Noted*

L291-99 – I think these would go better in the results section, with any methods employed described there. *We agree and will change this.*

[Figure]

Figure 7 – the plots don't really give much of an impression as to the variability within the line cloud – is it possible to replot the lines but set a transparency on each so can get an impression of the distribution of the modelled uncertainty?

*The uncertainty bounds represent a quantitative assessment, and we do not agree that simply changing the transparency would provide any relevant insight*

L313-324 – again, a clearer separation of results from the discussion would help

*Agree, noted.*

L313 – I would be very cautious of attempting to read too much into straight data/model comparisons without accounting for model grid size, flow approximations/model physics used, forcing and boundary conditions in the interpretation.

*This is a reasonable point which we agree with, but the multiple ice sheet models that we compare against have a range of different resolutions, boundary conditions, parameter choices and flow physics considered. This is the point. The purpose of the comparison is to highlight the differences between models as well as between the model suite and geological data – to illustrate that few models perform well and that there is still important work to do in this space.*

[Figure]

L325 – magnitude instead of amplitude?
*Agree, noted.*

Section 5.1 – this would benefit from a sentence or so on what the motivation for undertaking the data/model comparison is. As it's not mentioned in the paper before it appears a bit out of the blue currently.

*This is a fair point that was also noted by referee 1. We will move this text to the results section to help contextualise the modelling results.*

Section 5.2 – data presented in the paper are only written about in the last paragraph of this section, and otherwise is background info about the site.
*We agree and will include more discussion of our data. We argue that the 'site information' is critical here to highlight the offshore ice constraints as well as gaps in understanding.*

L383 – if the ice tongue is grounded then definitely, however if it isn't then it could be that the upstream ice thickness is maintained in a scenario where the Drygalski Ice Tongue is lost (as its removal would not change the amount of buttressing). To demonstrate this for certain though would require a separate set of model experiments. Unless there is other evidence for the Drygalski Ice Tongue being a permanent feature since 6ka BP I would still be cautious about linking it to the Terra Nova Bay polynya.

[Figure]

*We will clarify this point by better explaining existing geological constraints from around TNB e.g. The raised beach chronology suggests open marine conditions are established in TNB...our chronology from Hughes Bluff is the 'other evidence' for the persistence of the DIT. We include references to modern observations and paleo-oceanographic studies that suggest the intimate link with DIT and TNB polyna.*

L389-403 – most of this is site description rather than discussion.

*Agree. Happy to remove lines 391-394 but in our view L 396-403 remain a powerful comparison with modern understanding from satellite data as well as highlighting complexities in projecting Mt. Kring data over 10's of km to the flowline location in the middle of the ice stream / glacier.*

L408 – write out full abbreviation of MISI. *Agree, noted.*

L416-7 – if this is the case it should be acknowledged/alluded to when the definition of the model domain is described. *Agree, noted*

L422-30 – more site description than discussion of results.

*Agree, we will move some of this to background section but we argue some of this is relevant to the discussion topic: controls on thinning and retreat, particularly the potential for lingering ice on bathymetric highs and it's impact on lateral (drag) buttressing.*

[Figure]

L463-465 – this is quite a bold statement, and it is a bit of a leap to say that the results of this study show this conclusively.

*We respectfully disagree. Our modelling and chronology highlight a well-known process – dynamic thinning which has been observed in modern satellite data as well as in models. This process has been poorly documented over geologic timescales and we argue that our unique chronology documents this process and provides a first glimpse at how long dynamic thinning can persist. Given the unique nature of our chronology, we do not agree that this does not apply elsewhere in Antarctica, particularly those areas that have been shown to be undergoing dynamic thinning currently. Further, we state, 'if the data and modelling presented in this study is representative of outlet glacier behaviour more generally' and 'may' suggests the potential inconclusivity of our results. Essentially, this is the first paleo-documented case, otherwise we've only observed dynamic ice sheet thinning during the satellite era (last 40 years).*
* * *

---

## Referee Report (RR1)

The authors have carefully addressed all of the comments/suggestions brought up in the review. I thoroughly enjoyed reading the paper again and think it is in great shape. I recommend publishing in the current form, pending a few **very minor** things I noticed:

In Figure 6, the label for 140 m on the y axis is spread over two lines.

In Table S5 (the Be table), I think sample IDs need to be added in place of "JS1", "JS2" etc.

In the reference list, a DOI is needed for Frezzotti et al. (2005), Johnson et al. (2014), and Morlighem et al. (2020) – this latter reference should be changed from 2019 to 2020 (cited in lines 175 and 176).

---

## Referee Report (RR2)

**Review – Stutz et al., TCD –** *Mid-Holocene thinning of David Glacier: Chronology and Controls*

This is a review for the revised version of the above. All line numbers refer to the author tracked changes version of the manuscript. Below are a few minor comments on the new draft that mostly refer to style and clarity rather than anything substantive…

L178 – need to clarify if the bed geometry is taken from the transect or is width integrated (see Sergienko, 2012)

L193 – a sentence describing the success/otherwise of the modern day experiments would be informative

Figure 7 – though I note the author's response to my previous comment on this figure, I still feel that panel A especially would benefit from some level of transparency being applied to the grey age model lines. The potential age distributions for each elevation will not be normally distributed, and given the solid block of grey on the plot currently, coupled with the 95% confidence intervals this may give a false impression of this to the reader. Depending on level of confidence in the dates, by applying a transparency to each grey age model line it would also help the reader to pin down whether the apparent acceleration in thinning between 120 and 60 m is reflected in the age modelling. Whether this needs addressing in part depends whether the authors assume thinning rates to be linear/there is insufficient confidence in the dates at each elevation to capture variations in thinning rates, and if this is the case it should be stated in section 3.4. As the authors have already responded once to this comment I defer to the editor on whether this should be changed.

Section 5.1 – again noting the previous response of authors (comment on section 5.2 in previous review), while I agree that the information in here is important, all but the last paragraph should be placed in the background section of the paper. This will allow the last paragraph in this section where results of the study are referred to, to be expanded upon and placed more into direct context with previous work. I acknowledge this may be more a stylistic comment on my part, though think it would allow for a more complete discussion.

L447-454 – should try to make clear from the start of the paragraph that the authors are linking interactions with adjacent glaciers to changes in buttressing (thus making the link to the model experiments conducted more clear). Initially reading the first part of this paragraph was left thinking why experiments in changing the flux from tributaries wasn't tested, and at the moment it's only at the end of the paragraph that the reasoning for mentioning this becomes apparent.

Figure 12A – from the legend there's a similarity between the bold orange line  and the symbol for iceberg scour. This should be clarified.

L487 – suggest change "periods of onshore thinning" to "rates and magnitudes of onshore thinning" given that the model does not match up with real years.

L494-6 – actually think this sentence goes too far the other way and the first clause underplays the findings of the paper slightly. I agree it could be a template for future studies in some cases, though would suggest concisely flagging in the final paragraph that melt and buttressing changes appear to trigger retreat (i.e. on L493, instead of saying "offers clues" could say "as well as highlighting the role of moderate buttressing and submarine melt changes being potentially able to trigger rapid change in sectors of Antarctica and Greenland", or something roughly to that effect)

---

## Editor Decision (ED1)

I would like to thank both reviewers for their constructive comments on this manuscript and also the authors for posting their responses to the reviewers' comments.

The manuscript investigates the chronology and controls on ice thickness change along David Glacier, East Antarctica, during the last deglaciation via a combination of surface exposure dating and glacier flowline modelling.

The reviewers highlight that the study is a useful contribution to our understanding of the controls on outlet glacier behaviour which will be of interest to both glacial geologists and numerical modellers. However, both reviewers request additional detail on the modelling setup and justification of parameter choices. They also request a clearer explanation of how the dating and modelling components of the study inform each other and I recommend that you address this latter point within the opening section of the article to provide motivation for your two-pronged approach. It may also be useful to review previous studies that have carried out data-model comparisons in a similar setting.

A second aspect that is raised by both reviewers is the approach used to compare the field data and modelling results. Comparisons with previously published model output will be moved to earlier in the manuscript, which will improve the logical flow of the article. However, the reviewers also request that you state more clearly that the modelling carried out here comprises a sensitivity study, and that the experiments are not designed to replicate the chronological details of glacier retreat. When comparing results it would also be useful to acknowledge any factors that may explain differences between model output and data (e.g. unmodelled processes) or between modelling studies.

One reviewer queries the validity of your final conclusions. I encourage you to focus on the specific findings of your study, i.e. your identification of evidence for prolonged, widespread Holocene thinning along David Glacier, and the likely controls on that thinning. Any extrapolation to a different setting should be carefully justified. In general, please ensure that statements are supported by evidence.

Individually, the reviewers raise a number of additional points that require clarification. The authors provide a response to each of these points and indicate that edits will be made to the manuscript to address them. In addition to the issues highlighted by the reviewers, I note the following:

- the phrase 'ice surface elevation' usually refers to 'elevation above sea level', but of course, sea level changes through time. Referring to 'ice thickness' can navigate this tricky issue.

- when you refer to the W12 and ICE-6G models I think that you are referring to the ice sheet reconstructions created for the purposes of glacial isostatic adjustment (GIA) modelling, not the actual GIA models (which include predictions of bed deformation, sea-level change etc.)

- regarding the top panel of Fig. 6B: (i) if the melt rate is prescribed why does it vary over time, and (ii) how should the left y-axis be interpreted; is lateral buttressing prescribed or inferred?

- I encourage you to consider including Figures A3 and A4 (modelling results) in the main text to support statements in the Discussion and Conclusions relating to the role of buttressing or the combined role of buttressing and ice shelf thinning in controlling glacier dynamics

Overall, both reviews are positive, and I invite the authors to submit a revised version of the manuscript that addresses all points raised during this review process. When responding to more substantial points, please document the details of any changes implemented.

Kind regards,

Pippa Whitehouse

---

## Author Response (AR2)

**Response to Editor comments on "Mid-Holocene thinning of David Glacier, Antarctica: Chronology and Controls" by Stutz et al.**

I would like to thank the reviewers for reviewing the re-submitted version of this article. The reviewers identify a number of minor outstanding issues but confirm that their previous comments have largely been addressed. However, on reading the revised version of the manuscript I feel that there are still a large number of points that require clarification. To this end, I have carried out my own review of the current version, my comments are below. I hope that you find my feedback constructive.

Pippa Whitehouse (Editor)

We would like to thank the editor for their comprehensive review of this article. We are pleased to hear the reviewers confirm their previous comments have largely been addressed. We hope our responses to comments improve the clarity of our study.

**Major points**

**Model Setup**: there appear to be two sets of sensitivity experiments carried out; in the first, the aim is to reproduce the modern behaviour of David Glacier (line 167), while the second investigates the controls on glacier retreat from an LGM state (more on this in the next main point).

With regard to the experiments that seek to replicate modern conditions: what climate forcing was used?

The climate forcing used in these experiments that seek to replicate modern conditions is the surface mass balance from RACMO2 of van Wessem et al., 2018. We include the statement "modelled surface mass balance" with citation (van Wessem et al., 2018), on line 214.

What parameters were tuned and what values do these parameters take? I could not see any results documented for this set of experiments.

We agree that our model tuning approach could be better documented. Ultimately, we are using these modern sensitivity experiments as a starting point in which to conduct deglaciation experiments.

We include on lines 214-217: We use modelled surface mass balance (van Wessem et al., 2018) and modern geometry (Howat et al., 2019; Fretwell et al.,1852013) to set the model up so that it replicates modern flow conditions and geometry at a steady state. We then tune the basal traction parameter so that we end up with appropriate ice surface thickness and velocities.

We include a table of parameters Table 1 and values used in this model tuning approach.

With regard to the experiments that investigate controls on glacier retreat: lines 215-216 suggest that the deglacial experiments are initialised using parameters tuned to replicate modern conditions – state clearly what these parameters are, what values they take, and provide justification that they are relevant for initialising the model under glacial conditions.

- Lines 208-213 provide justification that basal traction values and understanding of variation through time is poorly understood.

o We include on line 216-217: 'We note here that as the ice surface evolves, basal traction evolves naturally - e. g. as the ice profile gets steeper or thicker, the basal traction evolves."
- Internal ice temperatures values and justification are provided on lines 244-248
- Accumulation values and justification are provided on lines 239-240

Separately, state what values are used for accumulation and temperature (ice? atmospheric? oceanic?) forcing in the deglacial experiments and whether they vary over the course of the experiments. I have tried to ascertain this information from the text but in places it is very confusing. E.g. text on lines 205-206 states that "transient changes in accumulation and internal ice temperature are tuned over the modelled period…" (what period?) "…to ensure a stable LGM configuration". How does this statement relate to the forcing applied in the deglacial experiments?

We scale RACMO2 accumulation values by 75%. Line 242-243.

We include on line 244-248: "For internal ice temperature, we increase this value through time to represent the relative increase in temperature though deglaciation. Knowledge of the internal ice temperature through deglaciation is poorly understood and for this study, we used values of -25∘C for the first 7,500 model years to -20∘C during the remaining 7,500 model years.

The modelled period is 15 kyr (line 264). We include on line 265-267: We run our experiments for 15 kyr to provide ample spin up time for glacial conditions. We force our model for 7.5 kyr to explore scenarios that, by the end of the modelled period, result in a configuration consistent with modern observations.

The statement on lines 235-236 relates to the forcing applied in the deglacial experiments by generally stating what approach we used to ensure a stable grounding line during the first 7,500 model years. We include on line 246-248: 'By accounting for changes in accumulation rate and internal ice temperatures during deglaciation, we are able to demonstrate these were accounted for and not responsible for driving modelled grounding line retreat.'

Further, we separate table S1 into two parts and include in main text. Now table 2: Experiment forcings and Table 3: Experiment results.

**Deglacial sensitivity experiments**: text on lines 112-114 clearly links the two main components of this study but justification for, and details of, the precise modelling experiments carried out is lacking or hard to track down within the text. Consideration of the following questions within the methods section would help clarify the study:

Based on previous reviewer suggestions, we have moved the results of our data – ice sheet model comparison (Section 3.5 and Fig. 8) within the results-chronology section in order to provide logical support for our modelling approach. We also introduce this logic on line 91 in the introduction.

- Why did you decide to carry out your own modelling experiments?

    o We provide justification for this in the opening paragraph of section 4-Results-Deglacial sensitivity experiments. We include that no prior detailed modelling has been undertaken for the David Glacier catchment (line 94).

- Are you seeking to address specific research questions?

- o We provide these two questions on lines 378-380.

- Specifically, why do you seek to investigate the role of basal melt?

    - o We now clarify the justification in the text.

    - o We state our general justification (without including reference to our chronology or data-ice sheet model comparison) on line 253-255

        - ▪ Further justification included on lines 255-256

        - ▪ Lines 334-340 suggest a paleo-thinning mechanism linked to a process driven from the ocean (e.g. elevated sub-ice shelf melt from relatively warm ocean water) rather than accumulation changes.

- and lateral buttressing?

    - o In the introduction (from line 66), we direct the reader to many studies that support this amalgamated nature of the ice sheet in the western Ross Sea.   As this grounded ice retreated, buttressing conditions would have changed and hence we investigate buttressing and basal melt as the two likely processes driving the dynamic outlet glacier response we see in our chronologies.

- Why do you think they are the most important factors controlling ice stream dynamics in this setting?

    - o Changes in ocean-driven basal melt and buttressing in the seward extension of the glacier, as explained above.

- What experiments were carried out, what values were used for each parameter, what is your justification for using these parameter choices / combinations? (Include a table in the main text)

    - o We split table S1, include more detail on experiments, values used for each parameter, and justification. This table 2 will be moved into the main text.

- You mention that grounding line retreat is initiated by progressively increasing sub-ice-shelf melt or decreasing lateral buttressing (lines 221, 331) – is the progressive change carried out within the course of a single numerical experiment or do you carry out many experiments with constant forcing to identify the threshold for triggering grounding line retreat?

    - o We clarify this on line 381-82

**Site information**: Field sites are first mentioned in the results – consider introducing them in methods section 2.1: Field and laboratory methods. The location of some sites is not clear from the main text, e.g. Cape Phillipi, and it is not clear when fieldwork was carried out.

Sites are now introduced in section 2.1. Location of Cape Phillipi is marked in figure S5.

**Interpretation**: for readers whose expertise does not lie in cosmogenic exposure dating, some aspects of the interpretation of the field data (sections 3.2 and 3.3) warrant additional explanation or supporting references. Check statements/text on lines 241, 243, 249, 250, 251-252, 263.

We include additional explanation and supporting references line in the methods section (Lines 105-115).

Lines 298-299: we clarify and include an additional reference to strengthen this statement.

**Data-model comparisons**: some comparisons made in the text are not supported by evidence in the figures. E.g., "the magnitude of modelled elevation change matches well with the surface exposure data" (line 315) – this statement is not supported by the information shown in figure 8.

We quantify the comparison to strengthen the statements on line 365-370:

Similarly, "at the LGM, the expanded David glacier had… a grounding line that was pinned on the David Fjord bathymetric sill" (lines 405-407) – modelling profiles in figures 9-11 have the glacial grounding line location well offshore, not pinned on the David Fjord bathymetric sill (assuming I have correctly identified the location of the sill).

We fix this error: "…and a stable grounding line near Coulman Island (Licht et al., 1996, Domack et al., 1999, Shipp et al., 1999, Mckay et al., 2008).

In addition, more care is needed to avoid making comparisons between the model predictions and the timing of ice sheet change recorded by the exposure data (see lines 348, 350, 364-365, 472-473, 487). The flowline sensitivity experiments simply apply a step change in forcing, they do not seek to replicate the evolution of forcing during the Holocene. A better approach may be to compare observed and modelled *rates* of thinning; the latter are not reported in the main text, but various statements claim 'good agreement' (lines 407-408, 473). You could also discuss what the modelling reveals about the *relative timing* of grounding line retreat and upstream thinning.

We agree that a better approach may be to compare observed and modelled rates of thinning. These comparisons are compiled in Table 3. Further, we expand on the thinning rate comparison for Mt. Kring and Hughes Bluff in section 5.2, lines 449-453.

**Combined forcing**: several statements imply that interactions between enhanced sub-ice shelf melt and reduced lateral buttressing are required to initiate grounding line retreat and/or retreat is triggered at lower threshold values when forcings are combined (see lines 15, 347, 366, 486). These results are not rigorously demonstrated by the results shown in the current manuscript.

We respectfully disagree that the results do not demonstrate that, compared with individual forcings, a better geometric fit and closer comparison with thinning rates is achieved in the combined forcing experiments. We do agree that the text can be expanded to clarify our combined forcing approach. We include clarifying text at lines 259-261 & 403-406.

**Section 5.1**: I agree with reviewer 2 that the information in this section provides useful motivation for your study and hence, with the exception of the final paragraph, material should be moved to the opening section of the manuscript.

We incorporate this useful motivation material to the introduction. We incorporate the final paragraph of section 5.1 into section 5.4 to draw stronger connections to summary figure 12.

**Writing style**: there are several places where the text is implicit rather than explicit, leaving the reader to guess what you are talking about. E.g. Instead of saying "Given the inferred intimate link between the expanded David Glacier and grounded ice in the Ross Sea…" (lines 325-326), this could read, "Given that ice from David Glacier coalesced with grounded ice in the Ross Sea…". Look out for other instances.

We agree and include this suggestion.

**Minor Comments**

**Comments on the text**

Many statements in the first paragraph of the introduction require supporting references

We now include further supporting references in first paragraph.

Line 39: suggest "…require a correction for *the ongoing response to* millennial…"

We agree with and include this suggestion.

Lines 38-43: it is not clear how "a geological perspective on ice sheet behaviour" (lines 31-32) addresses the issues raised in this point

We agree and re-write this point for clarity. Lines 41-48.

Lines 44-49: your study does not address the role of short-term climate variability

We agree and remove this text.

Lines 85-87: additional explanation is needed to explain how samples with "subglacial origins" are used "to track the upper ice surface through time"

We clarify this on lines 104-115

Line 100: "the PRIME lab" – what does PRIME stand for?

We include 'Purdue Rare Isotope Measurement Laboratory (PRIME lab)' on line 128-129.

Lies 125-126: please justify your use of a flowline model to represent ice *sheet* flow and clarify the implications of the assumptions stated on line 126

We remove 'sheet' and replace ice sheet interior with 'grounded portion the model domain'. Further, we remove 'shelf'

Line 132: incorrect units for gravitational acceleration

We now include '$m/s^2$' for gravitational acceleration.

Line 134: the exponent in equation 4 should be (1-n)/(2n)

We now include (1-n)/(2n) for the exponent in equation 4.

Lines 153-154: when you talk about 'lateral buttressing' I think you are referring to buttressing by ice rather than buttressing by, e.g., fjord margins – make this clear

On line 180, we replace 'such as lateral buttressing' with 'associated with MISI'. The following statement on lines 181 provides the context for lateral buttressing and we add '…a reduction in lateral buttressing *from adjacent, coalesced ice* is expected…'

Line 164: clarify what you mean by an "ice stream-parallel width"

We include clarification on line 190 and lines 193-194

Line 172: indicate the location where tributary mass flux is injected on relevant figures

We include the location of tributary ice on Figure 2.

Line 196: W12 is not a GIA model, it is an ice sheet reconstruction (that was created with the purpose of using it to drive a GIA model). A similar comment applies to the "prescriptive post-glacial rebound model" mentioned on line 303

We make the correction to line 226-7 and 352-3.

Lines 197-202: throughout this section you refer to 'ice surface elevation', but elevation is usually defined in terms of a height above sea level – you do not clarify what assumptions you make about sea level. I suggest referring to 'ice thickness' rather than 'ice surface elevation' throughout this paragraph. Ice thickness is defined relative to the bed (regardless of the absolute elevation of the ice surface or the bed) and ice thickness change is what is recorded by the exposure data.

We agree and the model requires an estimate/calculated upper ice surface elevation to run. We include the following statement to provide justification for our approach to model input. Line 229: 'introduce uncertainty involving variable along flow isostatic response and dynamic topography associated with the long-term evolution of the Antarctic subglacial topography.' Further, we include "calculated" when referring to the model input s, surface elevation 229 and 232.

Lines 202-204: text on the timing/processes responsible for retreat should be in the next paragraph

We agree and have moved this statement to line 255-264.

Line 222: is the 7.5 kyr spin-up prior to, or included within, the 15 kyr model run?

We clarify this on line 266-7.

Line 224: what is the temporal frequency of the sub-ice-shelf melt rate perturbations?

We clarify this on line 268-9.

Lines 266-267: text is unclear

We improve clarity on lines 313-315

Line 275: what does '320' refer to?

Typo, is removed.

Line 277: "at a number of sites in Antarctica" – summarise the geographic distribution

We summarise the geographic distribution on lines 325-330

Line 293: specify that you are talking about the East Antarctic Ice Sheet

We include EAIS on line 341.

Line 302: "forward ice sheet models" – explain, or simplify the terminology

We simplify by removing 'forward' on line 351.

Line 308: "its neighbouring grid cell" – surely each cell has several neighbours?

We include 'cells' on line 355.

Line 318: "which impact rate of retreat" – retreat of what? Suggest delete

We agree and delete 'which impact rate of retreat'

Line 335: "modelled surface reconstructions place the upper ice surface ~300 metres above…" – what experiment does this text relate to, what time does it relate to?

We clarify on line 389.

Line 345: please clarify what is pinned to the sill

We clarify on line 400.

Line 352: what are you referring to by the phrase "meltwater parameterization"?

We clarify on line 412

Lines 353-354: "a general fit to modern sub-ice shelf melt rates and basal stress" – are these known for David Glacier, if so, please report values and confirm the fit with your modelling

We report modern sub ice shelf melt rates (with citation) on lines 385-6 and justify our use of sub-ice shelf melt rates prescribed in our modelling approach.

While basal stress is poorly constrained/understood in general, the resultant modelled basal stress approaches 100 kPa. The general discussion on this item is found on lines 208-213 and the support for appropriate use of basal traction parameter and resultant modelled basal stress values is supported by the findings of Zoet et al., 2012 (lines 222-225).

Lines 411-412: please provide evidence to support your statement that stability of the grounding line implies stability of the Drygalski Ice Tongue

We provide support to this statement on line 443-447.

Lines 415-416: text on the Terra Nova Bay polynya is not relevant to this study

We respectfully disagree and include further justification on lines 441-446

Lines 445-446: is the "stable position" mentioned here related to the bathymetric sill mentioned on line 444? How does mention of these stable positions link to opening text in this paragraph, which is about unstable retreat?

We remove the use of stable and replace it with pinning point. We make similar changes elsewhere to ensure the proper use of 'stable'.

Line 447-454: as suggested by one of the reviewers, link this paragraph more closely to the modelling that you have carried out

We now provide a clearer link on line 487-489

Line 464: "The modelled grounding line initially retreats to a location where a large GZW has been documented by Lee (2019)" – be more specific about when this happens and the location of the GZW; there is no reference to a GZW identified by Lee (2019) in fig. 12

We provide more information on lines 500-501

In fig 12. caption, we now include grounding zone features: triangles with age constraints (orange) and without age constraints (green) of Lee (2019)

Lines 488-489: "Data-model mismatches highlight enduring questions…" – be more specific/explicit about what questions you are referring to

We remove "enduring questions" and clarify on line 370.

Line 492: "Our data constrain…" – what aspect of the glacier do your data constrain?

We clarify on line 532.

**Comments on Figures**

Figures 3 and 5: why are some ages in bold, why are some italicised?

We include in the caption: Bold ages indicate Holocene age and italicised ages indicate bedrock age.

Figures 4, 6 and 7: please explain the y-axis caption 'Relative Elevation'

On line 279, we establish use of 'above the ice' throughout text and link with 'relative elevation' usage in captions for Figs 4,6,7.

Fig. 8: it is difficult to differentiate between some lines; the pale blue line in the lower plot does not appear to be represented in the legend; it is not clear how the 12 models listed in the legend are related to the six studies referenced on lines 303-304

The model output for PSU_2018 was coloured incorrectly in the legend and Fig 8A. For consistency, we change the colour of PSU_2018 to match the pale blue colour of Fig 8B.

We agree that we can link the 11 models listed in the legend to the referenced studies on lines 353. We synchronise the names in the text and add qualifiers to differentiate model within each reference. 11 legend items appear as Pollard2016, Pollard2017, Pollard2017-ELRA, Pollard2018, Kingslake2018, Kingslake2018-WDC, Argus2016, Lowry2019-Cool, Lowry2019-Warm, Lowry-EMV, Lowry2019-Avg.

Figures 9-11: three sites are mentioned in the caption, but I only see two on the figures in (A)

We separate them for clarity (as in Fig S6).

Figures 9-11: the top plots in (B) are very confusing. The two sets of y-axes labels imply that a particular melt rate translates into a specific % reduction in lateral buttressing – is this correct? In figure 9, the legend suggests that melt rates are kept constant, but the plotted lines suggest that both melt rate and % of lateral buttressing vary with time. Similar comments can be made for figures 10 and 11; it is not clear what is actually represented by the coloured lines plotted in these figures.

We offer two y-axes, not to imply that one impacts another, rather to co-view forcings in one plot, relative to model years. The legend in figure 9B, incorrectly implies that melt rates are kept constant, whereas '% reducing' in lateral buttressing is held constant at 0%. We include discrete model names (M1, S1, MS1, etc) in the legend for clarity between figures, tables and text. For Figure 11B (top panel), we include dashed lines in the legend.

Figures 9-11: it is not appropriate to plot the exposure ages on the same figure as the output from the sensitivity experiments without careful caveat. The implication of plotting the exposure ages is that the x-axis represents some specific time in the past. However, the model output does not represent a specific time, it simply documents the response of a flowline to a step change in forcing.

We offer 382-383 as our careful caveat. We include a shorter version in each caption

Fig. 9: what does '0% reduction' refer to – no reduction in buttressing or no reduction in melt rate?

0% reduction refers to no reduction in lateral buttressing. Through our changes proposed above, both in the legend and caption, we believe this will add clarity.

Fig. 10: ice profiles in the top plot of Fig. 10A ('0% reduction') do not agree with plots of ice surface elevation change or grounding line migration in Fig. 10B. Fig. 10A also disagrees with text on line 342 which says that retreat occurs when buttressing is reduced by 4%

Thanks for pointing this out. It was a plotting error, has been fixed, and is now consistent with the text.

Fig. 11B: not clear how to interpret the six experiments shown by the six different lines; why are there only three lines in subsequent plots, and why are the two types of forcing seemingly applied at different times?

We hope that the changes proposed above, both in the legend and caption, have improved clarity. The forcings are applied at different times. We include a description of this 255-263.

Fig. 12: the upper plot is not a map of "Holocene thinning". It would be useful to include the position of B' in plot (A) to help relate the two figures. The caption to (B) refers to retreat between 11 ka and 5.5 ka, but plotted ice profiles relate to 13 ka and 4.5-5.5 ka.

We change the caption to more accurately represent the figure: "Map of David Glacier, Terra Nova Bay and surrounding areas. Synthesis map…"

We note that B' is indicated in the top right of figure 12A.

We include: A) Map of David Glacier, Terra Nova Bay and surrounding areas. Synthesis map focused on two phase retreat between 13 ka and 5.5 ka…'. We remove '4.5 ka' beneath red line depicting near modern position of grounding line.

**General Comments**

Check that all acronyms are defined, especially those used in figures

In all cases check it is clear whether 'surface' refers to bedrock or ice

Check the grammar of all text and figure captions. There are several instances of singular/plural errors and in a number of places the text does not make sense or is ambiguous

Check the use of brackets in conjunction with references in the text and figure captions

Use a consistent number of significant figures throughout the text, figures, and tables

We thank the editor for these general comments and have made careful checks to ensure consistency and accuracy.

**Comments on Supplementary Material**

Fig. S3/S4: ensure that the orientation of the photo is stated in all cases

In fig S3, we add: 'West facing oblique…'. In fig S4, we add: A) West facing oblique…, C) North facing view of moulded…, D) North facing view of…, E) West facing view of…

Figure S6: does the accumulation profile in the top plot relate to modern or palaeo conditions? Do the ice velocity and ice sheet profiles in the lowest two plots relate to modern or palaeo conditions? What is the implication of the different size yellow triangles?

The accumulation profile in the top plot relates to modern conditions. Ice velocity and ice sheet profiles in the lowest two plots relate to palaeo conditions. We include: '…modelled palaeo-ice surface and velocity of…' in caption.

We include the same colour and description for the triangles as in figure 12: triangles with age constraints (orange) and poorly constrained without age data (green) of Lee (2019)

Table S1: there are some inconsistencies in the text description of the modelling results. For example, why is the grounding line retreat rate for experiment MS1 described as 'rapid' when the value listed is less than half the value listed for experiments MS2 and MS3, where the retreat is described as 'moderate'?

We have fixed this error by including: 'moderate' modelled retreat behaviour for experiment MS1 and 'rapid' modelled retreat behaviour for experiment MS2 and MS3.

---

## Editor Decision (ED2)

**Editor comments on "Mid-Holocene thinning of David Glacier, Antarctica: Chronology and Controls" by Stutz et al.**

I would like to thank the reviewers for reviewing the re-submitted version of this article. The reviewers identify a number of minor outstanding issues but confirm that their previous comments have largely been addressed. However, on reading the revised version of the manuscript I feel that there are still a large number of points that require clarification. To this end, I have carried out my own review of the current version, my comments are below. I hope that you find my feedback constructive.

Pippa Whitehouse (Editor)

**Major points**

**Model Setup**: there appear to be two sets of sensitivity experiments carried out; in the first, the aim is to reproduce the modern behaviour of David Glacier (line 167), while the second investigates the controls on glacier retreat from an LGM state (more on this in the next main point).

With regard to the experiments that seek to replicate modern conditions: what climate forcing was used? What parameters were tuned and what values do these parameters take? I could not see any results documented for this set of experiments. With regard to the experiments that investigate controls on glacier retreat: lines 215-216 suggest that the deglacial experiments are initialised using parameters tuned to replicate modern conditions – state clearly what these parameters are, what values they take, and provide justification that they are relevant for initialising the model under glacial conditions. Separately, state what values are used for accumulation and temperature (ice? atmospheric? oceanic?) forcing in the deglacial experiments and whether they vary over the course of the experiments. I have tried to ascertain this information from the text but in places it is very confusing. E.g. text on lines 205-206 states that "transient changes in accumulation and internal ice temperature are tuned over the modelled period…" (what period?) "…to ensure a stable LGM configuration". How does this statement relate to the forcing applied in the deglacial experiments?

**Deglacial sensitivity experiments**: text on lines 112-114 clearly links the two main components of this study but justification for, and details of, the precise modelling experiments carried out is lacking or hard to track down within the text. Consideration of the following questions within the methods section would help clarify the study:

- Why did you decide to carry out your own modelling experiments?

- Are you seeking to address specific research questions?

- Specifically, why do you seek to investigate the role of basal melt and lateral buttressing? Why do you think they are the most important factors controlling ice stream dynamics in this setting?

- What experiments were carried out, what values were used for each parameter, what is your justification for using these parameter choices / combinations? (Include a table in the main text)

- You mention that grounding line retreat is initiated by progressively increasing sub-ice-shelf melt or decreasing lateral buttressing (lines 221, 331) – is the progressive change carried out within the course of a single numerical experiment or do you carry out many experiments with constant forcing to identify the threshold for triggering grounding line retreat?

**Site information**: Field sites are first mentioned in the results – consider introducing them in methods section 2.1: Field and laboratory methods. The location of some sites is not clear from the main text, e.g. Cape Phillipi, and it is not clear when fieldwork was carried out.

**Interpretation**: for readers whose expertise does not lie in cosmogenic exposure dating, some aspects of the interpretation of the field data (sections 3.2 and 3.3) warrant additional explanation or supporting references. Check statements/text on lines 241, 243, 249, 250, 251-252, 263.

**Data-model comparisons**: some comparisons made in the text are not supported by evidence in the figures. E.g., "the magnitude of modelled elevation change matches well with the surface exposure data" (line 315) – this statement is not supported by the information shown in figure 8. Similarly, "at the LGM, the expanded David glacier had… a grounding line that was pinned on the David Fjord bathymetric sill" (lines 405-407) – modelling profiles in figures 9-11 have the glacial grounding line location well offshore, not pinned on the David Fjord bathymetric sill (assuming I have correctly identified the location of the sill).

In addition, more care is needed to avoid making comparisons between the model predictions and the timing of ice sheet change recorded by the exposure data (see lines 348, 350, 364-365, 472-473, 487). The flowline sensitivity experiments simply apply a step change in forcing, they do not seek to replicate the evolution of forcing during the Holocene. A better approach may be to compare observed and modelled *rates* of thinning; the latter are not reported in the main text, but various statements claim 'good agreement' (lines 407-408, 473). You could also discuss what the modelling reveals about the *relative timing* of grounding line retreat and upstream thinning.

**Combined forcing**: several statements imply that interactions between enhanced sub-ice shelf melt and reduced lateral buttressing are required to initiate grounding line retreat and/or retreat is triggered at lower threshold values when forcings are combined (see lines 15, 347, 366, 486). These results are not rigorously demonstrated by the results shown in the current manuscript.

**Section 5.1**: I agree with reviewer 2 that the information in this section provides useful motivation for your study and hence, with the exception of the final paragraph, material should be moved to the opening section of the manuscript.

**Writing style**: there are several places where the text is implicit rather than explicit, leaving the reader to guess what you are talking about. E.g. Instead of saying "Given the inferred intimate link between the expanded David Glacier and grounded ice in the Ross Sea…" (lines 325-326), this could read, "Given that ice from David Glacier coalesced with grounded ice in the Ross Sea…". Look out for other instances.

**Minor Comments**

**Comments on the text**

Many statements in the first paragraph of the introduction require supporting references

Line 39: suggest "…require a correction for *the ongoing response to* millennial…"

Lines 38-43: it is not clear how "a geological perspective on ice sheet behaviour" (lines 31-32) addresses the issues raised in this point

Lines 44-49: your study does not address the role of short-term climate variability

Lines 85-87: additional explanation is needed to explain how samples with "subglacial origins" are used "to track the upper ice surface through time"

Line 100: "the PRIME lab" – what does PRIME stand for?

Lies 125-126: please justify your use of a flowline model to represent ice *sheet* flow and clarify the implications of the assumptions stated on line 126

Line 132: incorrect units for gravitational acceleration

Line 134: the exponent in equation 4 should be (1-n)/(2n)

Lines 153-154: when you talk about 'lateral buttressing' I think you are referring to buttressing by ice rather than buttressing by, e.g., fjord margins – make this clear

Line 164: clarify what you mean by an "ice stream-parallel width"

Line 172: indicate the location where tributary mass flux is injected on relevant figures

Line 196: W12 is not a GIA model, it is an ice sheet reconstruction (that was created with the purpose of using it to drive a GIA model). A similar comment applies to the "prescriptive post-glacial rebound model" mentioned on line 303

Lines 197-202: throughout this section you refer to 'ice surface elevation', but elevation is usually defined in terms of a height above sea level – you do not clarify what assumptions you make about sea level. I suggest referring to 'ice thickness' rather than 'ice surface elevation' throughout this paragraph. Ice thickness is defined relative to the bed (regardless of the absolute elevation of the ice surface or the bed) and ice thickness change is what is recorded by the exposure data.

Lines 202-204: text on the timing/processes responsible for retreat should be in the next paragraph

Line 222: is the 7.5 kyr spin-up prior to, or included within, the 15 kyr model run?

Line 224: what is the temporal frequency of the sub-ice-shelf melt rate perturbations?

Lines 266-267: text is unclear

Line 275: what does '320' refer to?

Line 277: "at a number of sites in Antarctica" – summarise the geographic distribution

Line 293: specify that you are talking about the East Antarctic Ice Sheet

Line 302: "forward ice sheet models" – explain, or simplify the terminology

Line 308: "its neighbouring grid cell" – surely each cell has several neighbours?

Line 318: "which impact rate of retreat" – retreat of what? Suggest delete

Line 335: "modelled surface reconstructions place the upper ice surface ~300 metres above…" – what experiment does this text relate to, what time does it relate to?

Line 345: please clarify what is pinned to the sill

Line 352: what are you referring to by the phrase "meltwater parameterization"?

Lines 353-354: "a general fit to modern sub-ice shelf melt rates and basal stress" – are these known for David Glacier, if so, please report values and confirm the fit with your modelling

Lines 411-412: please provide evidence to support your statement that stability of the grounding line implies stability of the Drygalski Ice Tongue

Lines 415-416: text on the Terra Nova Bay polynya is not relevant to this study

Lines 445-446: is the "stable position" mentioned here related to the bathymetric sill mentioned on line 444? How does mention of these stable positions link to opening text in this paragraph, which is about unstable retreat?

Line 447-454: as suggested by one of the reviewers, link this paragraph more closely to the modelling that you have carried out

Line 464: "The modelled grounding line initially retreats to a location where a large GZW has been documented by Lee (2019)" – be more specific about when this happens and the location of the GZW; there is no reference to a GZW identified by Lee (2019) in fig. 12

Lines 488-489: "Data-model mismatches highlight enduring questions…" – be more specific/explicit about what questions you are referring to

Line 492: "Our data constrain…" – what aspect of the glacier do your data constrain?

**Comments on Figures**

Figures 3 and 5: why are some ages in bold, why are some italicised?

Figures 4, 6 and 7: please explain the y-axis caption 'Relative Elevation'

Fig. 8: it is difficult to differentiate between some lines; the pale blue line in the lower plot does not appear to be represented in the legend; it is not clear how the 12 models listed in the legend are related to the six studies referenced on lines 303-304

Figures 9-11: three sites are mentioned in the caption, but I only see two on the figures in (A)

Figures 9-11: the top plots in (B) are very confusing. The two sets of y-axes labels imply that a particular melt rate translates into a specific % reduction in lateral buttressing – is this correct? In figure 9, the legend suggests that melt rates are kept constant, but the plotted lines suggest that both melt rate and % of lateral buttressing vary with time. Similar comments can be made for figures 10 and 11; it is not clear what is actually represented by the coloured lines plotted in these figures.

Figures 9-11: it is not appropriate to plot the exposure ages on the same figure as the output from the sensitivity experiments without careful caveat. The implication of plotting the exposure ages is that the x-axis represents some specific time in the past. However, the model output does not represent a specific time, it simply documents the response of a flowline to a step change in forcing.

Fig. 9: what does '0% reduction' refer to – no reduction in buttressing or no reduction in melt rate?

Fig. 10: ice profiles in the top plot of Fig. 10A ('0% reduction') do not agree with plots of ice surface elevation change or grounding line migration in Fig. 10B. Fig. 10A also disagrees with text on line 342 which says that retreat occurs when buttressing is reduced by 4%

Fig. 11B: not clear how to interpret the six experiments shown by the six different lines; why are there only three lines in subsequent plots, and why are the two types of forcing seemingly applied at different times?

Fig. 12: the upper plot is not a map of "Holocene thinning". It would be useful to include the position of B' in plot (A) to help relate the two figures. The caption to (B) refers to retreat between 11 ka and 5.5 ka, but plotted ice profiles relate to 13 ka and 4.5-5.5 ka.

**General Comments**

Check that all acronyms are defined, especially those used in figures

In all cases check it is clear whether 'surface' refers to bedrock or ice

Check the grammar of all text and figure captions. There are several instances of singular/plural errors and in a number of places the text does not make sense or is ambiguous

Check the use of brackets in conjunction with references in the text and figure captions

Use a consistent number of significant figures throughout the text, figures, and tables

**Comments on Supplementary Material**

Fig. S3/S4: ensure that the orientation of the photo is stated in all cases

Figure S6: does the accumulation profile in the top plot relate to modern or palaeo conditions? Do the ice velocity and ice sheet profiles in the lowest two plots relate to modern or palaeo conditions? What is the implication of the different size yellow triangles?

Table S1: there are some inconsistencies in the text description of the modelling results. For example, why is the grounding line retreat rate for experiment MS1 described as 'rapid' when the value listed is less than half the value listed for experiments MS2 and MS3, where the retreat is described as 'moderate'?

---

## Author Response (AR3)

**Dear authors,**

Thank you for addressing the points raised in the previous round of reviews and for submitting a revised version of your manuscript.

This is an extensive study that addresses several important research questions using complementary information from the analysis of field data and numerical modelling. The manuscript is significantly improved from the previous version and the two strands of the study are now clearly linked. The inclusion of additional tables is very useful and the restructuring that has taken place in several sections has improved the flow and coherence of the text. You have included additional detail in the methods sections, but many points remain unclear, and you will see a large number of queries in relation to this section below. The results are generally clearly summarised but justification for a couple of the key points mentioned in the abstract require a little more care when interpreting and communicating the results – these are addressed in my major points below.

I know that you will be frustrated to receive another detailed review. None of the points should require additional work to be carried out and therefore my decision is 'publish subject to minor revisions', but addressing the points below will require some careful edits in terms of explaining your methods and interpreting and reporting your findings.

Pippa Whitehouse (Editor)

**Major points**

**Drygalski Ice Tongue** (abstract line 7, section 5.1): as far as I can tell, presence or absence of the ice tongue cannot be inferred from the flowline modelling because the ice tongue does not play a role in controlling the dynamics/thickness of the glacier, it does not provide any buttressing. To make statements about the history of the ice tongue you would need to demonstrate that the observed thinning history cannot be replicated unless the ice tongue was present from a specific time. Since the data/modelling presented in this study does not provide any insight into the history of the ice tongue, please review the relevance of any related text about local ocean conditions (e.g. lines 441-447).

The evidence for the presence/absence of the ice tongue does not come from the modelling, but from the lowest exposure ages at Hughes Bluff. Thus, this is an interpretation from field data and not a modelling interpretation. We have already cited the relevant studies from Terra Nova Bay that support our conclusion. We have clarified our interpretation and discussion point in the text

Text changed to: The new geological reconstruction of ice surface elevation changes at Hughes Bluff shows a rapid lowering of the David Glacier at 6.5 ka and a period of slow thinning from ~6-4 ka. Given the marked slowdown in thinning rate from ~6 ka at Hughes Bluff (Figs 6B and 7B), we suggest that stabilisation of the Drygalski Ice Tongue occurred after ~6 ka. This finding is consistent with Orombelli et al. (1990) and Baroni and Hall (2004), who mapped a series of raised beaches along the TNB coastline that mark beach depositional processes in an open ocean setting (i.e. no grounded ice) initiating at 7.2 ka. Stevens et al. (2017a) show that the modern Drygalski Ice Tongue is essential for the development of the modern TNB Polynya. Thus, If the Drygalski Ice Tongue formed at ~6 ka as our exposure chronology from Hughes Bluff and the raised beach chronology from TNB suggests, it is likely that the TNB polynya has also existed since this time.

**Relative timing of retreat and thinning**: the abstract (lines 8-9) states that 'simultaneous thinning along the Transantarctic Mountains occurred ~3 ka after the retreat of marine-based grounded ice'. I could not find any explicit support for this statement in the results, discussion, or conclusions. A good place to address this issue

might be in section 5.3 (e.g. around lines 506-510), drawing on information presented in the results. See also comments relating to lines 512-514.

We now provide the range of data-ice sheet model mismatch of  $\sim$ 4-7 kyr. We now unify this in the abstract and give it another mention in section 3.5.

Abstract text changed to say: Our work, along with ice thinning records from adjacent glaciers, shows simultaneous glacier thinning in this sector of the Transantarctic Mountains occurred 4-7 ka after the peak period of ice thinning indicated in a suite of published ice sheet models.

**Modelling methodology**: despite useful edits to the methods section, many aspects of the modelling remain unclear, as evidenced by the large number of minor queries below. As you address these queries, please review the order that information is provided to the reader and make sure that the following are clearly stated: which parameters or variables are fixed, which are tuned to fit modern/LGM constraints, which vary over time/space, what is the experiment duration, how/when are changes in model forcing implemented?

**Addressed in 'minor' points below.**

**Data-model comparisons at Mt Kring**: Section 3.1 states that bedrock at the summit of Mt Kring (~300 m above the ice) has an exposure age of ~550 ka and that the highest erratics are ~180m above the ice (exposure age ~7.2 ka). Text on lines 344-345 implies that you assume the site has experienced ~200 m of Holocene thinning, but text on line 434 implies that you assume the ice surface was above the summit of Mt Kring at the LGM. Please clarify upper and lower bounds on the magnitude of Holocene thinning at this site to enable comparisons to be made with the modelling.

Text in Section 3.5 changed to 'at least 171 m of thinning'. Lower bound is modern ice surface.

We remove 'the level of Mt. Kring' on line 433 as section 5.1 discusses coastal thinning and impacts on local oceanography.

In various places (e.g. lines 391, 407) you state that the modelling agrees well with the data constraints at Mt Kring or modern observations (e.g. line 393). However, the final modelled ice surface at Mt Kring is several hundred metres below present in many of the experiments (this mismatch is acknowledged in relation to one of the experiments on line 401). I suspect the statements about a good fit are based on the fact that the modelled ice surface passes through the data constraints for some experiments (e.g. M3, fig. 9B; MS1-3, fig. 11B), but the continued thinning predicted by these experiments is in stark contrast with the fact that the present surface of the ice sheet is ~30 m below the lowest data point (fig. 4). Please review your assessment of the model fit to Holocene changes at Mt Kring.

We agree we need to be more careful here. We have now made table 3 and the text in section 4 consistent. For LBR experiments to indicate '...results in an unrealistic final ice surface elevation 100s of metres below observed modern ice surface'. For combined experiments, we include: 'For experiments MS1-MS3, the modelled upper ice surface reconstruction at Mt. Kring lies below the modern ice surface.' Further, we have referenced this effect in our model set up to highlight that during basal traction tuning, this effect is present but minimised.

**Minor points**

**Abstract and Introduction**

Line 14: text states that retreat and thinning is 'initiated by *interactions* between enhanced subiceshelf melting and reduced lateral buttressing'. This is not supported by the results of experiments M2/M3/S2/S3 (see figures 9 and 10) which show that retreat and thinning can be initiated solely by enhanced basal melt or a reduction in buttressing. Edit text to remove the implication that interactions between processes are necessary to initiate retreat.

Text changed to say: 'We show that glacier thinning, and marine-based grounding line retreat is controlled by either enhanced sub-ice shelf melting, reduced lateral buttressing, or a combination of the two, leading to Marine Ice Sheet Instability.'

Lines 15-16: Figure 8 demonstrates that rapid thinning *is* captured in previous large-scale modelling efforts; perhaps clarify what you mean by 'this period' (line 16)

Text changed to say: 'Such rapid glacier thinning events during the Mid- Holocene are not fully captured in continental or sector- scale numerical modelling reconstructions for this period.'

Lines 41-48: simplify this bullet point. Suggest starting at "Ice load reconstructions constrained by cosmogenic dating..." and say that the reconstructions are needed to model GIA, which (i) may play an important role in controlling ice sheet grounding line dynamics, and (ii) is needed to interpret gravity-based estimates of contemporary ice mass balance

We prefer to keep the text as it is – we are leading in from the general geological perspective and then going into detail about the processes and then how cosmogenic nuclides can help. We think this flows better than the alternative suggestion.

Line 72: text in this paragraph repeats and expands on earlier material. Consolidate the text and review the overall structure and flow of the Introduction.

We have consolidated the text. We believe this improves the overall structure and flow of the introduction.

Line 82: 'evidence of a lingering ice shelf' – when?

The point of this statement is to highlight the lack of coherent chronologies in the Drygalski Trough. Thus, we cannot say when. Text changed to: 'Marine sediment cores from...provide evidence of a lingering ice shelf...14C ages mean that the timing of this ice shelf presence is uncertain.'

**Methods**

Line 109: 'sufficient erosion' – sufficient for what? Text changed to: 'significant erosion'.

Line 114: 'would have been elevated to the former ice margin prior to deposition' – text is a little ambiguous, please clarify the process described here

Text changed to 'would have been brought to the glacier surface by upward-flowing ice before being deposited at the ice margin.

Line 151: equations 5 and 6 are revised versions of equation 3 – which version is used in this study? In general, try to be more specific about how the equations presented here are used in your study.

We add text: We also incorporate a width term, including  $f_{lat}$ , a lateral buttressing factor, to account for lateral buttressing along a coupled ice stream-shelf. Thus, as in previous applications of this model (Nick et al., (2010), Jamieson et al., 2012 and Whitehouse et al., 2017) the final modelled depth (*H*) and width (*W*) averaged ice flow (*u*) is computed using the following equation:

We also modify equation 6 and include corresponding description of  $f_{\text{lat.}}$

Equation 2: define all terms as soon as they are used

All terms in equation 2 defined directly after being used.

Line 163: '...equation 3 is modified...'

Changed to: 'For an ice stream, equation 3 is modified by including...'

Line 172: how is the statement about mapping geomorphic features related to the equations?

The offshore glacial geomorphology (for example the distribution of mega-scale glacial lineations) provides evidence of the former width of the ice stream, and provides the justification for having a width term in the equations. In the text, we clearly say that the width term (e.g. the equation) allows modelling of changing width as observed in the geomorphology.

We change text: 'To determine offshore trough width, we map all glacial geomorphic...' and draw attention to a more detailed description of the flowline and width follows on in section 2.2.1.

Line 180: MISI does not consider the role of the ice shelf, so it is not clear what sort of feedbacks are described here. Do you seek to understand how ice shelf buttressing can modify the MISI process?

Changed: we delete 'associated with marine ice sheet instability'. This is because we are looking at grounding line changes overall and how ice shelf changes may impact it and thus as you note, MISI is not the key here.

Line 180-181: how is a reduction in lateral buttressing implemented within your model?

See above regarding inclusion of flat term and corresponding description.

Line 190: method used to define the onshore flowline-parallel width is unclear and not replicable

Text changed to: 'We then construct a flowline that follows the centre of the ice stream and thereafter calculate ice stream width perpendicular to this flowline (W) and reaching to a defined lateral ice stream margin across the entire domain. Following Jamieson et al., 2012), the lateral ice stream margins in the offshore part of the domain are determined using geophysical data to map the distribution of trough-parallel mega scale glacial lineations (MSGLs) that are indicators of past ice stream flow, and thus show the width of the ice stream. Onshore, the lateral margins are defined by the valley width. The ice stream width perpendicular to the flowline is thereafter used to control the lateral stress applied by coalesced ice along the flowline (Eq. 6).'

Line 195: somewhere, it would be useful to explicitly state which parameters are tuned in the modern experiments. Some information is in Table 1, but this could be better signposted in the text

We modify text: '...we tune basal friction (B), accumulation, sub-ice shelf melt rate and the rate factor (A) in order to reproduce, as closely as possible, the modern geometry and flow speed of the David Glacier.

Line 214: should basal melt rate be listed as a user-defined parameter on line 197? How is basal melt implemented within your model? (no mention of this in section 2.2)

We have listed basal melt rate now – see above.

We also add new text immediately before section 2.2.1 which indicates how accumulation (due to a later comment) and sub-ice shelf melt rate are applied thus: 'Accumulation in the model is applied using modern rates measured at the central flowline and multiplied to account for the ice stream width (*W*). Accumulation is then scaled to represent warmer or cooler conditions in our experiments (see section 2.2.2). Where an ice shelf is present, sub-ice shelf melt rates are applied as a linear function of the depth of the ice shelf draught. From a minimum rate of 0.1 m yr-1 at 0 m (the ocean surface) SIMR increases to a maximum value at a depth of 500 m, and where ice shelf draught is deeper than 500 m, the maximum value is applied.'

Line 219: what is the modelled time period? Stated later, but reader needs to know before this point

Changed to: '...throughout a 2,000 year initiation period.'

Line 222-224: you state that you model basal stresses of 100 kPa, but then go on to say that Zoet et al. (2012) predict higher values which are consistent with your modelled values. The logic does not hang together.

We clarify the sentence thus: 'Resultant modelled estimates of basal shear stress approach 100 kPa in this setting. While difficult to constrain with in situ measurements, Zoet et al. (2012) suggest higher stresses such as these should be expected near the modern day grounding zone which is consistent with the modelled stress distribution in this study.'

Table 1: (i) units are missing for temperature and sub-ice-shelf melt rates. (ii) What is implied by a negative melt rate? (iii) Please expand on the fact that you only list 'maximum' basal melt rates. (iv)

Clarify whether the values listed in the final column relate to glacial or deglacial conditions – it may be useful to document in the caption how/when the different conditions are applied in the model.

(i) Added degrees C, m/yr. (ii) Removed negative signs throughout manuscript, (iii) This should now be clear from the newly added section indicated how sub ice shelf melt rates are applied. (iv) We change deglacial to glacial.

Change table 1 caption to say: 'Parameter values used for sensitivity experiments indicating choices for model tuning to modern conditions over a spinup period and subsequently applied deglacial conditions. SIMR=sub-ice shelf melt rate. Accumulation values reported as percentage of RACMO2 (van Wessem et al., 2018).'

Line 230: need to explicitly state that the model does not account for isostatic deformation (along with other model limitations that only become apparent on line 412

Changed to: '...without introducing uncertainty involving variable along-flow isostatic response and dynamic topography associated with the long-term evolution of the Antarctic subglacial topography (Stern et al., 2005; Whitehouse et al., 2019; Paxman et al., 2019) because our ice flow model does not adjust for isostatic deformation as it evolves.'

**Note that we do not indicate the other limitations at this stage and prefer to leave them in-situ.**

Lines 235-236: In the authors' rebuttal, it is explained that this text describes the approach used to ensure a stable grounding line during the first 7,500 model years. However, that is not what is implied by the text, which includes confusing information about tuning transient changes to ensure a stable LGM (how does this relate to model time?) configuration. How can applying transient changes in temperature and accumulation result in a stable configuration? How is the tuning carried out? What is 'the modelled period'?

'Transient' was not the correct word choice, and neither was 'stable'. Text changed to: 'To account for environmental changes during deglaciation, accumulation and internal ice temperature are tuned over the modelled period to ensure an LGM configuration which is consistent with geological constraints and which has a grounding line that is not moving significantly.'

The detail of the tuning and the period are further down in the paragraph, so we do not clarify them further.

Line 243: why are we told that accumulation was 75% of modern at 15 ka BP? How does this information relate to the forcing applied in the numerical experiments?

Text changed to: 'We use a scaling relationship between modern accumulation patterns and estimate that accumulation at the start of the model run was roughly 75% of modern accumulation (Veres et al., 2013).'

Line 244: 'we increase this value' - what value?

Text changed to: 'Internal ice temperature is increased through time to represent the increase in temperature that occurred during deglaciation. The internal ice temperature during deglaciation is not known and for this study, so we used values of -25C for the first 7,500 model years and -20C during the remaining 7,500 model years as a way to represent an appropriate amount of warming in the ice column.'

We remove the final sentence of this paragraph indicating we robustly tested accumulation and temperature, which we did not.

Line 246-247: it is not clear how accumulation changes over time are applied within the model

We have added some detail immediately prior to section 2.2.1 which indicates: 'Accumulation in the model is applied using modern rates measured at the central flowline and multiplied to account for the ice stream width (*W*). Accumulation is then scaled to represent warmer or cooler conditions in our experiments (see 2.2.2).'

Line 247-248: 'we are able to demonstrate...' – this result (which perhaps belongs in the results) is not robust because you do not investigate the impact of varying the temperature or accumulation values within the model, despite acknowledging (line 254) that they are poorly constrained

We agree and remove the sentence: 'By accounting for changes in accumulation rate and internal ice temperatures during deglaciation, we are able to demonstrate these were not responsible for

driving modelled grounding line retreat.'

Line 250: 'user-defined parameters...' – explicitly state what these are and what values they take

Text changed to: 'Deglacial sensitivity experiments use a range of accumulation and internal ice temperature forcings representing the potential scale of change experienced through a deglaciation (Table 1), to explore transient changes in lateral buttressing reduction (LBR) and subice shelf melt rates (SIMR) in order to isolate their relative influence on glacier thinning and retreat.'

Line 250: 'optimised...forcings' - how are they optimised?

Text modified – see above – e.g. 'optimised' not the right phrasing.

Line 257: it is not clear how progressive changes in forcing are applied – over time within a single experiment or by running a suite of experiments, each with different boundary conditions?

Text changed to: 'We therefore run a suite of individual experiments that allow us to initiate grounding line retreat by either linearly increasing SIMR or decreasing lateral buttressing over a 500 year period, with each model run applying different perturbed values for SIMR or LBR (Table 1). For combined forcing experiments (MS1-3), we alter both SIMR and lateral buttressing as above until the grounding line retreats to a near modern configuration.'

Line 265: 'a forcing perturbation is applied' - be more explicit about what this entails

Text changed to: 'All sensitivity experiments run for 15 kyr with an initial spin up period lasting 7.5 kyr for SIMR forcing and 8.5 kyr for LBR forcing, at which point the forcing perturbation relating to increased SIMR or reduced lateral buttressing is applied for the remaining modelled period.'

**Results**

In a few places, additional information is needed to explain your interpretation of the field data:

- Line 296: 'Two bedrock exposure ages ... suggest significant wet-based glacial erosion' – the reason for inferring wet-based erosion from these ages is not clearly stated

Text changed to: 'Two bedrock surface exposure ages sampled from rochés moutonnées at the highest and lowest outcrops...'

Line 298: '...displays extensive glacial erosion which suggests the ice thickness at the LGM was considerably greater than 230m' – again, more information is needed to explain why such erosion could not have been carried out by a thinner ice sheet

Text changed to: The fact that well developed landforms of glacial erosion occur at the highest outcrops at Hughes Bluff, including evidence of abundant basal sliding and plucking, indicate that the ice thickness was considerably greater than 230 m.

 Lines 311-312: '...suggest either a thin cover of cold-based ice or ice-free conditions...' – not clear how you reach this conclusion (noting that you didn't mention the potential for a thin layer of cold-based ice when discussing bedrock ages from the summit of Mt Kring)

Text changed to: In an effort to identify higher elevation glacial activity and long-term erosion history, field work was undertaken along the northern flank of David Glacier from the D'Urville Wall area (Mt. Neumayer to Cape Phillipi) (Fig. S5)

 Lines 312-313: 'High elevation bedrock samples are much younger... suggests burial by nonerosive ice' – several aspects of this sentence are unclear: which 'high elevation' samples are you referring to, where is the 'nearby' site with older bedrock ages at a similar height and which site do you suggest was buried by non-erosive ice?

We clarify in the text: Bedrock exposure ages from D'Urville Wall area (including Mt. Neumayer and Cape Phillipi) do not allow a precise estimate of the past ice surface along the northern flank of David Glacier. Supported by geomorphic evidence from Hughes Bluff, which indicates ice thicker than 230 m, and the LGM limit of ~400 masl derived from drift deposits in TNB, we suggest the past ice surface was between 300 and 649 meters higher than today in this area (Stuiver, Orombelli1990, DiNicola2009) (Fig. S2).

Line 362: in this context, ICE-6G is an ice sheet reconstruction, not a post glacial rebound model

Line 366: how do you determine the ice thickness change for each model? Do you calculate the thickness change over a specific time period, or do you calculate the difference between the minimum and maximum ice thickness at any time during the model run?

Data constrained thickness values incorrect, changed in text. Model thickness values calculated during deglacial phase.

Text changed to: 'At Mt. Kring, the average modelled thickness change throughout the deglacial phase for all models in Fig. 8A is 190 m  $\pm$  117 m, which compares well with the 173 m thickness change derived from our ice thinning chronology. In contrast, at Hughes Bluff, we capture only 171 metres of thickness change over the deglacial period and the average modelled thickness change for all models in Fig. 8B is 623 m  $\pm$  142 m.'

Line 367: figure 4b implies that the highest Holocene erratic at Mt Kring is ~170m above the present ice surface, i.e. there has been at least 170m thinning during the Holocene (not 144m). Similarly, where does the value of 181 m come from for Hughes Bluff (line 368)?

See previous comment and associated change in text.

Lines 387-390: text implies that experiments are carried out for melt rates between -2 and -10 m/yr, but such experiments are not listed in table 2 or shown in figure 9. The results presented here do not provide convincing evidence that -11 m/yr is the threshold value for triggering grounding line retreat

*Text, table and figures are consistent but maybe not clear to provide convincing evidence that 11 m/yr is threshold to trigger grounding line retreat to near modern conditions.*

Experiments are carried out for melt rates between 2 and 10 m/yr but they show very similar results: grounding line pinned to sill, upper surface above Hughes Bluff. We do not include these experiments because we would like to avoid 8 extra panels that show the similar results. We prefer to show model results where significant changes can be observed in the plots. Previously, we chose to show the lower SIMR value that initiates a change whereas it seems clearer to show the experiment with higher SIMR values.

Experiment M2 has been updated with model output corresponding to SIMR of 10 m/yr (in table, text and figure 9A).

Model M1: text on lines 389-391 implies that M1 predicts rapid grounding line retreat and that the results agree well with Mt Kring data constraints, but this is not supported by figure 9

Text changed to: 'For melt rates between 2 and 10 m/yr (exp M1, M2 respectively), grounding line retreat is initially rapid but...'

Line 397: does the 'further ice shelf debuttressing' take place after the 4% reduction in buttressing, i.e. during the same experiment, or in a completely separate experiment? Also, is a reduction by 40% the minimum value required for grounding line retreat to modern, or is this is simply an example of an experiment that showed full grounding line retreat?

We delete the word 'Further'. We confirm that the 40% reduction seems to be the minimum level of debuttressing required to retreat the grounding line to near modern if LBR on its own is being applied, but we note further down in the paragraph that it does not result in the correct final surface elevation.

Line 401: 'this simulation' - which simulation?

Text changed to: 'At Mt. Kring, modelled rapid thinning is synchronous with Hughes Bluff yet, the 40% LBR simulation (S3) results in an unrealistic final surface elevation 100s of metres below observed modern surface elevation.'

Table 3: (i) over what time period is 'modelled grounding line retreat rate' calculated? (ii) suggest listing the data-constrained thinning rates at each site, to allow comparison with modelled rates

(i) We include an \* in the table header and then define in the caption. Text included in caption: \*indicates the modelled retreat rate is calculated during the deglacial phase of experiments. The retreat rate is calculated during the deglacial phase of the experiment. (ii) We include a + in the table header for Max. thinning rates at MK and HB. Text included in caption: +Data constrained Max. thinning rate at Mt. Kring (MK) and Hughes Bluff (HB) is 0.19 m/yr and 2.06 m/yr, respectively.

**Discussion**

Line 424: take care when talking about 'matching periods of thinning' – the numerical experiments simply explore the response to an instantaneous perturbation to the boundary conditions. Your comparison could be taken to imply that Holocene change along the David Glacier was driven by a single, sudden change in local conditions. Also, it is not clear what it means for modelled retreat to 'match' the onshore thinning, are you implying that they occur at the same time?

Text changed to: Combined forcing experiments show a similar rate, magnitude and duration of onshore thinning compared with our onshore geologic records constrained...

Line 508: does the two-phase grounding line retreat result in two phases of onshore thinning?

'compares well' changed to 'aligns with phases of enhanced and reduced thinning identified in our onshore reconstructions at Mt. Kring and Hughes Bluff...'

Lines 512-514: the text implies that thinning initiated prior to grounding line retreat. However, text on lines 515-517 implies that grounding line retreat preceded onshore thinning. Which is correct?

We suggest grounding line retreat initiates onshore thinning. Text changed to: 'This synthesis suggests that beginning at 7.5 ka, the grounding line unpins from the sill at the mouth of the David Fjord, the David Glacier and proto-Nansen...'

Lines 528-530: be more explicit about the fact that you apply an instantaneous change to boundary conditions rather than applying time-evolving forcing

Text changed to: 'Modelled patterns of grounding line retreat correlate well with patterns of onshore thinning,...'

**Figures, grammar etc.**

Check for minor grammatical issues, e.g. words missing, singular/plural errors, sentences that do not make sense, use of hyphens in compound adjectives (e.g. sea-level rise, sub-ice-shelf melt)

Ensure that any new text is carefully incorporated into the existing text

Figures are very clear. Check the use of brackets when including citations in figure captions. Support the text in the Results and Discussion with more references to figures. I don't think the Drygalski Trough is labelled on any figure.

Drygalski Trough now labelled in Figure 1 and 2.

Figure 2: Aviator Glacier (AG) is not labelled on the figure, REG is not defined in the caption

Aviator Glacier (AG) and Mariner Glacier (MG) now labelled in figure. Rennick Glacier (REG) now defined in caption.

Figure 8: it is not possible to identify the source of all the lines, e.g. what is the difference between Kingslake2018 and Kingslake2018\_WDC; what is Lowry2019\_EMV? Check y-axis label on lower plot

We agree the Kingslake2018 and Kingslake2018\_WDC (WAIS divide accumulation forcing) are difficult to distinguish in Figure 8B. They are the same from -22-~-13ka and from -13ka to 0 ka, only have minor differences. We suggest to keep them as they are. We have included in the caption, Kingslake2018\_WDC uses WAIS Divide accumulation forcing and Lowry2019\_EMV is a model in which Enhanced Mantle Viscosity is applied.

**Plotting error corrected for y-axis on lower plot.**

Figures 9-11: (i) label the field sites on one of the panel A plots, (ii) consider indicating the position of the present-day grounding line and labelling the 'prominent sill' on one of the panel A plots, (iii) state which axis each of the lines in the top plots of panel B relate to and make it clear that there is no relationship between melt rates and % buttressing, (iv) why do melt rates seem to vary randomly through time in the top plots of panel B?

(i) Field sites labelled in top panel Figs 9A, 10A and 11A (ii) Grounding line and sill labelled in top panel Figs 9A, 10A and 11A, (iii) include in caption: no relationship between LBR and SIMR, (iv)

Clarified text on 265-67: In order to simulate the natural variability that might be expected in an ocean forcing record, we apply a fluctuation of up to 0.5 m/yr magnitude using a random noise generator and this variation is then added on top of the step increase in forcing that is already applied.

Further, we include the following text in caption for Figs 9, 10 & 11: For SIMR cases, we varied the SIMR perturbation within a 0.5 m/yr window at 500 year temporal frequency in order to simulate short-lived pulses of relatively warmer or cooler water mass changes.

---

## Editor Decision (ED3)

Dear authors,

Thank you for addressing the points raised in the previous round of reviews and for submitting a revised version of your manuscript.

This is an extensive study that addresses several important research questions using complementary information from the analysis of field data and numerical modelling. The manuscript is significantly improved from the previous version and the two strands of the study are now clearly linked. The inclusion of additional tables is very useful and the restructuring that has taken place in several sections has improved the flow and coherence of the text. You have included additional detail in the methods sections, but many points remain unclear, and you will see a large number of queries in relation to this section below. The results are generally clearly summarised but justification for a couple of the key points mentioned in the abstract require a little more care when interpreting and communicating the results – these are addressed in my major points below.

I know that you will be frustrated to receive another detailed review. None of the points should require additional work to be carried out and therefore my decision is 'publish subject to minor revisions', but addressing the points below will require some careful edits in terms of explaining your methods and interpreting and reporting your findings.

Pippa Whitehouse (Editor)

Major points

**Drygalski Ice Tongue** (abstract line 7, section 5.1): as far as I can tell, presence or absence of the ice tongue cannot be inferred from the flowline modelling because the ice tongue does not play a role in controlling the dynamics/thickness of the glacier, it does not provide any buttressing. To make statements about the history of the ice tongue you would need to demonstrate that the observed thinning history cannot be replicated unless the ice tongue was present from a specific time. Since the data/modelling presented in this study does not provide any insight into the history of the ice tongue, please review the relevance of any related text about local ocean conditions (e.g. lines 441-447).

**Relative timing of retreat and thinning**: the abstract (lines 8-9) states that 'simultaneous thinning along the Transantarctic Mountains occurred ~3 ka after the retreat of marine-based grounded ice'. I could not find any explicit support for this statement in the results, discussion, or conclusions. A good place to address this issue might be in section 5.3 (e.g. around lines 506-510), drawing on information presented in the results. See also comments relating to lines 512-514.

**Modelling methodology**: despite useful edits to the methods section, many aspects of the modelling remain unclear, as evidenced by the large number of minor queries below. As you address these queries, please review the order that information is provided to the reader and make sure that the following are clearly stated: which parameters or variables are fixed, which are tuned to fit modern/LGM constraints, which vary over time/space, what is the experiment duration, how/when are changes in model forcing implemented?

**Data-model comparisons at Mt Kring**: Section 3.1 states that bedrock at the summit of Mt Kring (~300 m above the ice) has an exposure age of ~550 ka and that the highest erratics are ~180m above the ice (exposure age ~7.2 ka). Text on lines 344-345 implies that you assume the site has experienced ~200 m of Holocene thinning, but text on line 434 implies that you assume the ice

surface was above the summit of Mt Kring at the LGM. Please clarify upper and lower bounds on the magnitude of Holocene thinning at this site to enable comparisons to be made with the modelling.

In various places (e.g. lines 391, 407) you state that the modelling agrees well with the data constraints at Mt Kring or modern observations (e.g. line 393). However, the final modelled ice surface at Mt Kring is several hundred metres below present in many of the experiments (this mismatch is acknowledged in relation to one of the experiments on line 401). I suspect the statements about a good fit are based on the fact that the modelled ice surface passes through the data constraints for some experiments (e.g. M3, fig. 9B; MS1-3, fig. 11B), but the continued thinning predicted by these experiments is in stark contrast with the fact that the present surface of the ice sheet is ~30 m below the lowest data point (fig. 4). Please review your assessment of the model fit to Holocene changes at Mt Kring.

Minor points

**Abstract and Introduction**

Line 14: text states that retreat and thinning is 'initiated by *interactions* between enhanced sub-ice-shelf melting and reduced lateral buttressing'. This is not supported by the results of experiments M2/M3/S2/S3 (see figures 9 and 10) which show that retreat and thinning can be initiated solely by enhanced basal melt or a reduction in buttressing. Edit text to remove the implication that interactions between processes are necessary to initiate retreat.

Lines 15-16: Figure 8 demonstrates that rapid thinning *is* captured in previous large-scale modelling efforts; perhaps clarify what you mean by 'this period' (line 16)

Lines 41-48: simplify this bullet point. Suggest starting at "Ice load reconstructions constrained by cosmogenic dating…" and say that the reconstructions are needed to model GIA, which (i) may play an important role in controlling ice sheet grounding line dynamics, and (ii) is needed to interpret gravity-based estimates of contemporary ice mass balance

Line 72: text in this paragraph repeats and expands on earlier material. Consolidate the text and review the overall structure and flow of the Introduction.

Line 82: 'evidence of a lingering ice shelf' – when?

**Methods**

Line 109: 'sufficient erosion' – sufficient for what?

Line 114: 'would have been elevated to the former ice margin prior to deposition' – text is a little ambiguous, please clarify the process described here

Line 151: equations 5 and 6 are revised versions of equation 3 – which version is used in this study? In general, try to be more specific about how the equations presented here are used in your study.

Equation 2: define all terms as soon as they are used

Line 163: '…equation *3* is modified…'

Line 172: how is the statement about mapping geomorphic features related to the equations?

Line 180: MISI does not consider the role of the ice shelf, so it is not clear what sort of feedbacks are described here. Do you seek to understand how ice shelf buttressing can modify the MISI process?

Line 180-181: how is a reduction in lateral buttressing implemented within your model?

Line 190: method used to define the onshore flowline-parallel width is unclear and not replicable

Line 195: somewhere, it would be useful to explicitly state which parameters are tuned in the modern experiments. Some information is in Table 1, but this could be better signposted in the text

Line 214: should basal melt rate be listed as a user-defined parameter on line 197? How is basal melt implemented within your model? (no mention of this in section 2.2)

Line 219: what is the modelled time period? Stated later, but reader needs to know before this point

Line 222-224: you state that you model basal stresses of 100 kPa, but then go on to say that Zoet et al. (2012) predict higher values which are consistent with your modelled values. The logic does not hang together.

Table 1: (i) units are missing for temperature and sub-ice-shelf melt rates. (ii) What is implied by a negative melt rate? (iii) Please expand on the fact that you only list 'maximum' basal melt rates. (iv) Clarify whether the values listed in the final column relate to glacial or deglacial conditions – it may be useful to document in the caption how/when the different conditions are applied in the model.

Line 230: need to explicitly state that the model does not account for isostatic deformation (along with other model limitations that only become apparent on line 412)

Lines 235-236: In the authors' rebuttal, it is explained that this text describes the approach used to ensure a stable grounding line during the first 7,500 model years. However, that is not what is implied by the text, which includes confusing information about tuning transient changes to ensure a stable LGM (how does this relate to model time?) configuration. How can applying transient changes in temperature and accumulation result in a stable configuration? How is the tuning carried out? What is 'the modelled period'?

Line 243: why are we told that accumulation was 75% of modern at 15 ka BP? How does this information relate to the forcing applied in the numerical experiments?

Line 244: 'we increase this value' – what value?

Line 246-247: it is not clear how accumulation changes over time are applied within the model

Line 247-248: 'we are able to demonstrate…' – this result (which perhaps belongs in the results) is not robust because you do not investigate the impact of varying the temperature or accumulation values within the model, despite acknowledging (line 254) that they are poorly constrained

Line 250: 'user-defined parameters…' – explicitly state what these are and what values they take

Line 250: 'optimised…forcings' – how are they optimised?

Line 257: it is not clear how progressive changes in forcing are applied – over time within a single experiment or by running a suite of experiments, each with different boundary conditions?

Line 265: 'a forcing perturbation is applied' – be more explicit about what this entails

**Results**

In a few places, additional information is needed to explain your interpretation of the field data:

- Line 296: 'Two bedrock exposure ages … suggest significant wet-based glacial erosion' – the reason for inferring wet-based erosion from these ages is not clearly stated

- Line 298: '…displays extensive glacial erosion which suggests the ice thickness at the LGM was considerably greater than 230m' – again, more information is needed to explain why such erosion could not have been carried out by a thinner ice sheet

- Lines 311-312: '…suggest either a thin cover of cold-based ice or ice-free conditions…' – not clear how you reach this conclusion (noting that you didn't mention the potential for a thin layer of cold-based ice when discussing bedrock ages from the summit of Mt Kring)

- Lines 312-313: 'High elevation bedrock samples are much younger… suggests burial by non-erosive ice' – several aspects of this sentence are unclear: which 'high elevation' samples are you referring to, where is the 'nearby' site with older bedrock ages at a similar height and which site do you suggest was buried by non-erosive ice?

Line 362: in this context, ICE-6G is an ice sheet reconstruction, not a post glacial rebound model

Line 366: how do you determine the ice thickness change for each model? Do you calculate the thickness change over a specific time period, or do you calculate the difference between the minimum and maximum ice thickness at any time during the model run?

Line 367: figure 4b implies that the highest Holocene erratic at Mt Kring is ~170m above the present ice surface, i.e. there has been at least 170m thinning during the Holocene (not 144m). Similarly, where does the value of 181 m come from for Hughes Bluff (line 368)?

Lines 387-390: text implies that experiments are carried out for melt rates between -2 and -10 m/yr, but such experiments are not listed in table 2 or shown in figure 9. The results presented here do not provide convincing evidence that -11 m/yr is the threshold value for triggering grounding line retreat

Model M1: text on lines 389-391 implies that M1 predicts rapid grounding line retreat and that the results agree well with Mt Kring data constraints, but this is not supported by figure 9

Line 397: does the 'further ice shelf debuttressing' take place after the 4% reduction in buttressing, i.e. during the same experiment, or in a completely separate experiment? Also, is a reduction by 40% the minimum value required for grounding line retreat to modern, or is this is simply an example of an experiment that showed full grounding line retreat?

Line 401: 'this simulation' – which simulation?

Table 3: (i) over what time period is 'modelled grounding line retreat rate' calculated? (ii) suggest listing the data-constrained thinning rates at each site, to allow comparison with modelled rates

**Discussion**

Line 424: take care when talking about 'matching periods of thinning' – the numerical experiments simply explore the response to an instantaneous perturbation to the boundary conditions. Your comparison could be taken to imply that Holocene change along the David Glacier was driven by a

single, sudden change in local conditions. Also, it is not clear what it means for modelled retreat to 'match' the onshore thinning, are you implying that they occur at the same time?

Line 508: does the two-phase grounding line retreat result in two phases of onshore thinning?

Lines 512-514: the text implies that thinning initiated prior to grounding line retreat. However, text on lines 515-517 implies that grounding line retreat preceded onshore thinning. Which is correct?

Lines 528-530: be more explicit about the fact that you apply an instantaneous change to boundary conditions rather than applying time-evolving forcing

**Figures, grammar etc.**

Check for minor grammatical issues, e.g. words missing, singular/plural errors, sentences that do not make sense, use of hyphens in compound adjectives (e.g. sea-level rise, sub-ice-shelf melt)

Ensure that any new text is carefully incorporated into the existing text

Figures are very clear. Check the use of brackets when including citations in figure captions. Support the text in the Results and Discussion with more references to figures. I don't think the Drygalski Trough is labelled on any figure.

Figure 2: Aviator Glacier (AG) is not labelled on the figure, REG is not defined in the caption

Figure 8: it is not possible to identify the source of all the lines, e.g. what is the difference between Kingslake2018 and Kingslake2018_WDC; what is Lowry2019_EMV? Check y-axis label on lower plot

Figures 9-11: (i) label the field sites on one of the panel A plots, (ii) consider indicating the position of the present-day grounding line and labelling the 'prominent sill' on one of the panel A plots, (iii) state which axis each of the lines in the top plots of panel B relate to and make it clear that there is no relationship between melt rates and % buttressing, (iv) why do melt rates seem to vary randomly through time in the top plots of panel B?

---

## Author Response (AR4)

Dear authors,

Thank you for resubmitting your article on "Mid-Holocene Thinning of David Glacier, Antarctica: Chronology and Controls". You have made significant improvements to the clarity of the article, including the description of the methods and the interpretation of the results.

As previously mentioned, this is an important study, and the underlying scientific results are robust. However, there remain a large number of inconsistencies or errors, and several points require clarification. These are annotated in the attached pdf along with a few grammatical issues.

I have decided to accept this article with a requirement for you to carry out 'technical corrections'. This means that the article will not return to me for review – I suspect you are very pleased to hear this. Most of the annotations identify small factual points. I have not highlighted them in order to create additional work for you, rather, they reflect my desire for you to communicate your research as clearly and accurately as possible to the reader. It is now over to you to complete that task.

Thank you for choosing to publish your research in The Cryosphere.

Pippa Whitehouse (Editor)

Dear Editor,

We appreciate your time and effort in editing this manuscript. We have completed all technical corrections and look forward to publication.

Thank you,
Jamey Stutz (lead author)